# High Rank Path Development: an approach to learning the filtration of stochastic processes

**Jiajie Tao**
Department of Mathematics
University College London
ucahjta@ucl.ac.uk

**Hao Ni**
Department of Mathematics
University College London
h.ni@ucl.ac.uk

**Chong Liu**[*]
Institute of Mathematical Sciences
ShanghaiTech University
liuchong@shanghaitech.edu.cn

## Abstract

Since the weak convergence for stochastic processes does not account for the growth of information over time which is represented by the underlying filtration, a slightly erroneous stochastic model in weak topology may cause huge loss in multi-periods decision making problems. To address such discontinuities Aldous introduced the extended weak convergence, which can fully characterise all essential properties, including the filtration, of stochastic processes; however was considered to be hard to find efficient numerical implementations. In this paper, we introduce a novel metric called High Rank PCF Distance (HRPCFD) for extended weak convergence based on the high rank path development method from rough path theory, which also defines the characteristic function for measure-valued processes. We then show that such HRPCFD admits many favourable analytic properties which allows us to design an efficient algorithm for training HRPCFD from data and construct the HRPCF-GAN by using HRPCFD as the discriminator for conditional time series generation. Our numerical experiments on both hypothesis testing and generative modelling validate the out-performance of our approach compared with several state-of-the-art methods, highlighting its potential in broad applications of synthetic time series generation and in addressing classic financial and economic challenges, such as optimal stopping or utility maximisation problems. Code is available at https://github.com/DeepIntoStreams/High-Rank-PCF-GAN.git.

## 1 Introduction

A popular criterion for measuring the differences between two stochastic processes is the weak convergence. In this framework, one views stochastic processes as path-valued random variables and then defines the convergence for their laws, which are distributions on path space. However, this viewpoint ignores the *filtration* of stochastic processes, which models the evolution of information, and therefore such loss may have negative implications in multi-period optimisation problems. For example, for the American option pricing task, even if the two underlying processes are stochastic processes with very similar laws, the corresponding price of American options can be completely different, see a toy example A.1 in Appendix A.1. To address this shortcoming of weak convergence, D. Aldous [1] introduced the notion of *extended weak convergence*. The central object in

---

[*]Corresponding author

38th Conference on Neural Information Processing Systems (NeurIPS 2024).

this methodology is the so-called prediction process, which consists of conditional distributions of the underlying process based on available information at different time beings, and therefore reflects how the associated information flow (i.e., filtration) affects the prediction of the future evolution of the underlying process as time varies. Instead of considering the laws of processes (i.e., distributions on path space) in weak convergence, one compares the laws of prediction processes, which are distributions on the *measure-valued* path space, in extended weak convergence. Since the knowledge of filtration is captured through taking conditional distributions, it was shown in [3] the topology induced by extended weak convergence, which belongs to the so-called *adapted weak topologies*[2], fully characterise essential properties of stochastic processes and endow multi-period optimisation problems with continuity, provided filtration is generated by the process itself.

While the theoretical contributions to adapted weak topologies flourish in recent years (e.g., [3], [2], [4]), the related work on numerics is still very sparse because of the very complex nature of these topologies. In this paper, we propose a novel metric called *High Rank Path Characteristic Function (HRPCFD)* which can metrise the extended weak convergence, and, more importantly, admits an efficient implementation algorithm. The core idea of this approach is built on top of the unitary feature of $\mathbb{R}^d$-valued paths ([18], [7]), which exploits the non-commutativity and the group structure of the unitary developments to encode information on order of paths. Based on the same consideration, Lou et al. [19] introduced the Path Characteristic Function (PCF) for stochastic pro-

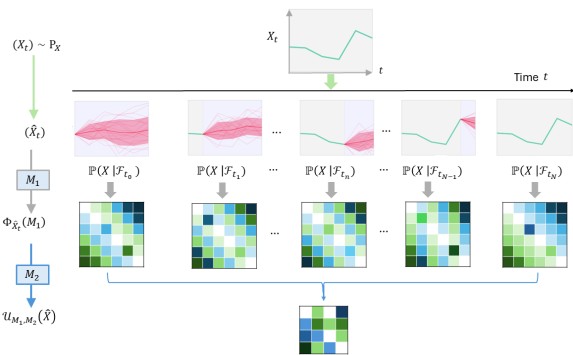

Figure 1: The high-level illustration of the high rank path development. Here the prediction process $\hat{X}_t := \mathbb{P}(X|\mathcal{F}_t)$ for all $t \in [0, T]$, $\mathbf{\Phi}_{\hat{X}_t}(M_1)$ is the PCF of the prediction process and $\mathcal{U}_{M_1,M_2}(\hat{X})$ is the high rank development of the path $t \mapsto \mathbf{\Phi}_{\hat{X}_t}(M_1)$ under the linear map $M_2$.

cesses, which induces a computable distance (namely, PCFD) to metrise the weak convergence. As extended weak convergence is defined in terms of laws of prediction processes which are measure-valued stochastic processes, the scheme of PCF remains valid in adapted weak topologies as long as one can construct a PCF of measure-valued paths. One of the main contributions of the present work is to give such a suitable notion via the so-called high rank path development (see Figure 1 for illustration); moreover, we can show that the induced distance (called HRPCFD) does not only characterise the more complicated extended weak convergence, but also inherits almost all favourable analytic properties of classical PCFD mentioned in [19]. Since the measure-valued paths take values in an infinite dimensional nonlinear space, such a generalisation of the results in [19] from $\mathbb{R}^d$-valued paths to measure-valued paths is much technically involved and therefore significantly nontrivial.

On the numerical side, we design an efficient algorithm to train HRPCFD from data and construct the HRPCF-GAN model with HRPCFD as the discriminator for time series generation. A key computational challenge in applying distances based on extended weak topology is the accurate and efficient estimation of the conditional probability measure. To address this issue, we have implemented a sequence-to-sequence regression module that effectively resolves this bottleneck. Our work is the first of its kind to apply the adapted weak topology for generative models on time series generation. Moreover, to validate the effectiveness of our approach, we conduct experiments in (1) hypothesis testing to classify different stochastic processes, and (2) conditional time series generation to predict the future time series given the past time series. Our HRPCF-GAN can be viewed as a natural generalisation of PCF-GAN [19] to the setting of extended weak convergence, so that the data generated by HRPCF-GAN possesses not only a similar law but also a similar filtration with the target model. The numerical experiments validate the out-performance of this new approach based on HRPCFD compared with several state-of-the-art GAN models for time series generation in terms of various test metrics.

---

[2]In general, any topology on the space of stochastic processes which can reflect the differences of associated filtrations can be called an adapted weak topology.

**Related work.** So far most of existing statistical and numerical methods for handling stochastic processes (e.g., [9, 16, 19]) are based on weak convergence, and the results on numerical implementation of adapted weak topologies are rather limited. The most relevant work is [21], whose theoretical foundation roots in [5]. The present paper shares a similar philosophy with [21] in the sense that both methods for defining metrics for extended weak convergence rely on the construction of a feature of the measure-valued path by transforming it into a linear space-valued path. In contrast to [21], where a measure-valued path is lifted to an infinite-dimensional Hilbert space, we reduce measure-valued paths into matrix-valued paths through unitary development which allows us to apply the techniques from [19] to design the algorithm. Another remarkable point is that in [21] one has to solve a large family of PDEs to compute the distance, which can be avoided in the numerical estimation of the HRPCFD proposed here. On the other hand, as Wasserstein distances can metrise weak convergence, the so-called causal Wasserstein distances can be used to measure adapted weak topologies. One related work is [22] which can be seen as an improved variant of the Sinkhorn divergence tailored to sequential data. Note that the discriminator (i.e., causal Wasserstein metric) used in [22] is slightly weaker than the HRPCFD, as the latter is actually equivalent to the bi-causal Wasserstein distance.

## 2 Preliminaries

### 2.1 Prediction Processes and Extended Weak Convergence

Let $I = \{0, \ldots, T\}$ and $X = (X_t)_{t \in I}$ be an $\mathbb{R}^d$–valued stochastic process defined on a filtered stochastic basis $(\Omega^X, \mathcal{F}, \mathbb{F} = (\mathcal{F}_t)_{t \in I}, \mathbb{P})$ such that $X$ is adapted to the filtration $\mathbb{F}$, i.e., $X_t$ is measurable with respect to $\mathcal{F}_t$ for all $t \in I$. We call the five–tuple $(\Omega^X, \mathcal{F}, \mathbb{F}, X, \mathbb{P})$ a *filtered process*, and denote it by $\mathbb{X}$. Throughout this paper, we will use FP to denote the space of all ($\mathbb{R}^d$–valued) filtered processes on the discrete time interval $I$, and assume that $\mathbb{F}$ is the natural filtration in the sense that for every $t \in I$, $\mathcal{F}_t = \sigma(X_0, \ldots, X_t)$.

Since each discrete time path $\boldsymbol{x} \in (\mathbb{R}^d)^{T+1}$ can be uniquely extended to a piecewise linear path on $[0, T]$ by linear interpolation, we will not distinguish the product space $(\mathbb{R}^d)^{T+1}$ and the subspace $\mathcal{X} := \{\boldsymbol{x} : [0, T] \to \mathbb{R}^d : \boldsymbol{x} \text{ is piecewise linear}\}$ of $C^{1\text{-var}}([0, T], \mathbb{R}^d)$ (the space of all continuous functions in $\mathbb{R}^d$ with bounded variation)[3]. Clearly each stochastic process $X$ can be seen as $\mathcal{X}$-valued random variable, and therefore the law of $X$, denoted by $P_X = \mathbb{P} \circ X^{-1}$, belongs to $\mathcal{P}(\mathcal{X})$, the space of probability measures on the path space $\mathcal{X}$. Recall that a sequence of filtered processes $\mathbb{X}^n = (\Omega^n, \mathcal{F}^n, \mathbb{F}^n, X^n, \mathbb{P}^n)$ converges to a limit $\mathbb{X}$ weakly or in the weak topology (in notation: $\mathbb{X}^n \xrightarrow{W} \mathbb{X}$) if the laws $P_{X^n} = \mathbb{P}^n \circ (X^n)^{-1}$ converges to $P_X$ in $\mathcal{P}(\mathcal{X})$ weakly, i.e., for all continuous and bounded functions $f \in C_b(\mathcal{X})$, it holds that $\lim_{n \to \infty} \mathbb{E}_{\mathbb{P}^n}[f(X^n)] = \mathbb{E}_{\mathbb{P}}[f(X)]$.

For each $t \in I$, we denote $\hat{X}_t := \mathbb{P}(X \in \cdot | \mathcal{F}_t)$ as the (regular) conditional distribution of $X$ given $\mathcal{F}_t$, which is a random measure taking values in $\mathcal{P}(\mathcal{X})$. We call this measure-valued process $\hat{X} = (\hat{X}_t)_{t \in I}$ the *prediction process* of the filtered process $\mathbb{X}$. By definition it is clear that the state space of $\hat{X}$ is $\mathcal{P}(\mathcal{X})^{T+1}$ and, again, by a routine linear interpolation[4], we can embed $\mathcal{P}(\mathcal{X})^{T+1}$ into $\hat{\mathcal{X}} = \{\boldsymbol{p} : [0, T] \to \mathcal{P}(\mathcal{X}) : \boldsymbol{p} \text{ is piecewise linear}\}$. Thus the law of $\hat{X}$, denoted by $P_{\hat{X}} = \mathbb{P} \circ \hat{X}^{-1}$, belongs to $\mathcal{P}(\hat{\mathcal{X}})$ (the space of probability measures on the measure-valued path space $\hat{\mathcal{X}}$), where $\hat{\mathcal{X}}$ is endowed with the product topology and $\mathcal{P}(\hat{\mathcal{X}})$ is equipped with the corresponding weak topology.

**Definition 2.1.**
- *Two filtered processes $\mathbb{X} = (\Omega^X, \mathcal{F}, \mathbb{F}, X, \mathbb{P})$ and $\mathbb{Y} = (\Omega^Y, \mathcal{G}, \mathbb{G}, Y, \mathbb{Q})$ are called synonymous if their prediction processes $\hat{X}$ and $\hat{Y}$ have the same law in $\mathcal{P}(\hat{\mathcal{X}})$, i.e., $P_{\hat{X}} = P_{\hat{Y}}$.*

- *A sequence of filtered processes $\mathbb{X}^n = (\Omega^n, \mathcal{F}^n, \mathbb{F}^n, X^n, \mathbb{P}^n)$, $n \in \mathbb{N}$ converges to another filtered process $\mathbb{X} = (\Omega^X, \mathcal{F}, \mathbb{F}, X, \mathbb{P})$ in the extended weak convergence if the law of their prediction processes $\hat{X}^n$ converges to the law of $\hat{X}$ in $\mathcal{P}(\hat{\mathcal{X}})$ weakly, i.e., for all continuous and bounded functions $\hat{f} \in C_b(\hat{\mathcal{X}})$, $\lim_{n \to \infty} \mathbb{E}_{\mathbb{P}^n}[\hat{f}(\hat{X}^n)] = \mathbb{E}_{\mathbb{P}}[\hat{f}(\hat{X})]$. In notation: $\mathbb{X}^n \xrightarrow{EW} \mathbb{X}$.*

---

[3]This means that $\mathcal{X}$ is equipped with the topology induced by the total variation norm.

[4]For $\boldsymbol{p}_{0=t_0}, \boldsymbol{p}_{t_1}, \ldots, \boldsymbol{p}_{t_N=T} \in \mathcal{P}(\mathcal{X})$ and $t \in [0, T]$, define $\boldsymbol{p}_t = \frac{t - t_i}{t_{i+1} - t_i} \boldsymbol{p}_{t_{i+1}} + \frac{t_{i+1} - t}{t_{i+1} - t_i} \boldsymbol{p}_{t_i}$.

If $\mathcal{F}_0^n$ and $\mathcal{F}_0$ are the trivial $\sigma$-algebra, then $\hat{X}_0^n = P_{X^n}$ and $\hat{X}_0 = P_X$ are laws of $X^n$ and $X$ respectively, so that $\mathbb{X}^n \xrightarrow{EW} \mathbb{X}$ certainly implies that $\mathbb{X}^n \xrightarrow{W} \mathbb{X}$. This implies that extended weak convergence is stronger than weak convergence. Moreover, the extended weak convergence induces the correct topology in multi-period decision making problems, as the next theorem (see [1, 3]) shows.

**Theorem 2.2.** *The extended weak convergence provides continuity for the value functions in multi-period optimisation problems (e.g., optimal stopping problem, utility maximisation problem), as long as the reward function is continuous and bounded.*

Admittedly, the above notions related to extended weak convergence (e.g., the spaces $\hat{\mathcal{X}}$ and $\mathcal{P}(\hat{\mathcal{X}})$, the weak convergence in $\mathcal{P}(\hat{\mathcal{X}})$ etc.) are rather abstract. Therefore, we provide some simple examples in Appendix A.1 to explain these notions in a more transparent way. We refer readers to [1] and [3] for more details on extended weak convergence.

## 2.2 Path Development and Path Characteristic Function (PCF)

In this subsection, we review some important notions and properties of $\mathbb{R}^d$-valued path development and characteristic function (PCF) for $\mathbb{R}^d$-valued stochastic processes, which will be used later to construct characteristic functions for measure-valued stochastic processes. More technical details on PCF can be found in Appendix A.2. We also refer readers to [19] and [18] for a more detailed discussion on this topic.

For $m \in \mathbb{N}$, let $\mathbb{C}^{m \times m}$ be the space of $m \times m$ complex matrices, $I_m$ denote the identity matrix in $\mathbb{C}^{m \times m}$, and $*$ be conjugate transpose. Write $U(m)$ and $\mathfrak{u}(m)$ for the Lie group of $m \times m$ unitary matrices and its Lie algebra, resp.:

$$U(m) = \{A \in \mathbb{C}^{m \times m} : A^* A = I_m\}, \quad \mathfrak{u}(m) = \{A \in \mathbb{C}^{m \times m} : A + A^* = 0\}.$$

Let $\mathcal{L}(\mathbb{R}^d, \mathfrak{u}(m))$ denote the space of linear mappings from $\mathbb{R}^d$ to $\mathfrak{u}(m)$.

**Definition 2.3.** *Let $\boldsymbol{x} \in C^{1\text{-}var}([0, T], \mathbb{R}^d)$ be a continuous path with bounded variation and $M \in \mathcal{L}(\mathbb{R}^d, \mathfrak{u}(m))$ be a linear map. The unitary feature of $\boldsymbol{x}$ under $M$ is the solution $\boldsymbol{y} : [0, T] \to U(m)$ to the following equation:*

$$d\boldsymbol{y}_t = \boldsymbol{y}_t M(d\boldsymbol{x}_t), \quad \boldsymbol{y}_0 = I_m, \tag{1}$$

*where $\boldsymbol{y}_t M(d\boldsymbol{x}_t)$ denotes the usual matrix product. We write $\mathcal{U}_M(\boldsymbol{x}) := \boldsymbol{y}_T$, i.e., the endpoint of the solution path, and by an abuse of notation, also call it the unitary feature of $\boldsymbol{x}$ (under $M$).*

The unitary feature is a special case of the *path development*, for which one may consider paths taking values in any Lie group $G$. It is easy to see that for piecewise linear path $\boldsymbol{x} = (\boldsymbol{x}_0, \ldots, \boldsymbol{x}_T) \in \mathcal{X}$, it holds $\mathcal{U}_M(\boldsymbol{x}) = \prod_{i=1}^{T} \exp(M(\Delta \boldsymbol{x}_i))$ for $\Delta \boldsymbol{x}_i = \boldsymbol{x}_i - \boldsymbol{x}_{i-1}$ and $\exp$ denotes the matrix exponential. We now use the unitary feature to define the Path Characteristic Function (PCF) for $\mathbb{R}^d$-valued stochastic processes:

**Definition 2.4.** *Let $\mathbb{X} = (\Omega^X, \mathcal{F}, \mathbb{F}, X, \mathbb{P})$ be a filtered process and $P_X$ be its law. The Path Characteristic Function (PCF) of $\mathbb{X}$ is the map $\boldsymbol{\Phi}_\mathbb{X} : \bigcup_{m \in \mathbb{N}} \mathcal{L}(\mathbb{R}^d, \mathfrak{u}(m)) \to \bigcup_{m \in \mathbb{N}} \mathbb{C}^{m \times m}$ given by*

$$\boldsymbol{\Phi}_\mathbb{X}(M) := \mathbb{E}_\mathbb{P}[\mathcal{U}_M(X)] = \int_\mathcal{X} \mathcal{U}_M(\boldsymbol{x}) P_X(d\boldsymbol{x}).$$

**Remark 2.5.** *In the present work, we only consider the discrete-time processes defined on $I = 0, \ldots, T$, and therefore the time index $t$ appeared in the stochastic process $X_t$ and its filtration $\mathcal{F}_t$ only takes values in $0, \ldots, T$. It is just a convention in the rough path community that one views a discrete time path defined on $I = 0, \ldots, T$ as a piecewise linear path defined on the continuous time interval $[0, T]$ by a routine linear interpolation, because such identification may make some formulations and computations easier (e.g., by doing so the unitary feature of a path can be formulated as the solution of an ODE on $[0, T]$).*

To distinguish $\Phi_\mathbb{X}$ from the so-called high rank PCF which will be defined in the next subsection, we also call $\Phi_\mathbb{X}$ the rank 1 PCF. The next theorem (see [19, Theorem 3.2]) justifies why $\boldsymbol{\Phi}_\mathbb{X}$ defined in Definition 2.4 is called PCF for path-valued random variables.

**Theorem 2.6** (Characteristicity of laws). *For $\mathbb{X}$ and $\mathbb{Y}$ two filtered processes, they have the same law (i.e., $P_X = P_Y$) if and only if $\mathbf{\Phi}_\mathbb{X} = \mathbf{\Phi}_\mathbb{Y}$.*

The characteristicity of PCF allows us to define a novel distance on FP which metrises the weak convergence (locally). This metric is called the PCF-based distance (PCFD), see [19, Definition 3.3]. Moreover, such PCFD possesses many nice analytic properties including boundedness ([19, Lemma 3.5]), Maximum Mean Discrepancy (MMD, [19, Proposition B.10]) among others, see [19, Section 3.2], which ensures the feasibility of using PCFD in numerical aspect.

**Remark 2.7.** *Rigorously speaking, we need to add an additional time component to every $\mathbb{R}^d$-valued process $X$ (i.e., consider $\bar{X}_t = (t, X_t^1, \ldots, X_t^d)$) to guarantee Theorem 2.6 holds true. We will always implicitly use such time-augmentation throughout the whole paper and still write $X$ instead of $\bar{X}$ for simplicity of notations.*

# 3 High Rank Path Development Embedding

We now want to construct a characteristic function for prediction processes and use it to metrise the extended weak convergence just like PCFD metrises the weak convergence. Since prediction processes are $\hat{\mathcal{X}}$-valued random variables, we first need to find a suitable notion of unitary feature/development for measure-valued paths.

## 3.1 High Rank Development of Prediction Processes

Given a filtered process $\mathbb{X} = (\Omega^X, \mathcal{F}, \mathbb{F}, X, \mathbb{P})$, remember that its prediction process $\hat{X}$ satisfies $\hat{X}_t = \mathbb{P}(X \in \cdot | \mathcal{F}_t)$ for $t \in I$. Now, for a linear operators $M \in \mathcal{L}(\mathbb{R}^d, \mathfrak{u}(n))$ for $n \in \mathbb{N}$, we take the conditional expectation of $\mathcal{U}_M$ against $\hat{X}_t = \mathbb{P}(X \in \cdot | \mathcal{F}_t)$ to obtain a $\mathbb{C}^{n \times n}$–valued stochastic process $\mathbf{\Phi}_{\hat{X}_t}(M) = \mathbb{E}_\mathbb{P}[\mathcal{U}_M(X) | \mathcal{F}_t]$, $t \in I$. Then, for any $\mathcal{M} \in \mathcal{L}(\mathbb{C}^{n \times n}, \mathfrak{u}(m))$ with some $m \in \mathbb{N}$, the unitary feature $\mathcal{U}_\mathcal{M}(t \mapsto \mathbf{\Phi}_{\hat{X}_t}(M))$ of $\mathbb{C}^{n \times n}$–valued path $(t \mapsto \mathbf{\Phi}_{\hat{X}_t}(M))$ is well defined and takes values in the unitary group $U(m)$. We call each pair $(M, \mathcal{M}) \in \mathcal{L}(\mathbb{R}^d, \mathfrak{u}(n)) \times \mathcal{L}(\mathbb{C}^{n \times n}, \mathfrak{u}(m))$ for $(n, m) \in \mathbb{N}^2$ an admissible pair of unitary representations, and the set of all admissible pairs of unitary representations is denoted by $\mathcal{A}_{\text{unitary}}$.

**Definition 3.1.** *For $(M, \mathcal{M}) \in \mathcal{A}_{\text{unitary}}$ with $M \in \mathcal{L}(\mathbb{R}^d, \mathfrak{u}(n))$, $\mathcal{M} \in \mathcal{L}(\mathbb{C}^{n \times n}, \mathfrak{u}(m))$ and $\mathbb{X} = (\Omega^X, \mathcal{F}, \mathbb{F}, X, \mathbb{P})$ a filtered process with its prediction process $\hat{X}$, we call*

$$\mathcal{U}_{M, \mathcal{M}}(\hat{X}) := \mathcal{U}_\mathcal{M}(t \mapsto \mathbf{\Phi}_{\hat{X}_t}(M)) \tag{2}$$

*the high rank development of the prediction process $\hat{X}$ under $(M, \mathcal{M})$.*

See Figure 1 for the schematic overview of the high rank development. From above we can see that the construction of $\mathcal{U}_{M, \mathcal{M}}(\hat{X})$ involves with taking finite dimensional path development in Section 2.2 *twice*: first use the PCF under $M \in \mathcal{L}(\mathbb{R}^d, \mathfrak{u}(n))$ to transform each conditional distribution $\mathbb{P}(X \in \cdot | \mathcal{F}_t)$ into a matrix $\mathbf{\Phi}_{\hat{X}_t}(M)$, and then apply the unitary feature $\mathcal{U}_\mathcal{M}(\cdot)$ to the resulting matrix-valued path $(t \mapsto \mathbf{\Phi}_{\hat{X}_t}(M))$ for $\mathcal{M} \in \mathcal{L}(\mathbb{C}^{n \times n}, \mathfrak{u}(m))$.

## 3.2 High Rank Path Characteristic Function

With the above notion of unitary feature of measure-valued paths, following Definition 2.4, we define the high rank Path Characteristic Function (HRPCF) for filtered processes.

**Definition 3.2.** *For a filtered process $\mathbb{X} = (\Omega^X, \mathcal{F}, \mathbb{F}, X, \mathbb{P}) \in$ FP, the function*

$$\mathbf{\Phi}_\mathbb{X}^2 : \mathcal{A}_{\text{unitary}} \to \bigcup_{m=1}^{\infty} \mathbb{C}^{m \times m}; (M, \mathcal{M}) \mapsto \mathbb{E}_\mathbb{P}[\mathcal{U}_{M, \mathcal{M}}(\hat{X})] = \mathbb{E}_\mathbb{P}[\mathcal{U}_\mathcal{M}(t \mapsto \mathbb{E}_\mathbb{P}[\mathcal{U}_M(X) | \mathcal{F}_t])]. \tag{3}$$

*is called the High Rank Path Characteristic Function of $\mathbb{X}$ (Abbreviation: HRPCF)[5].*

---

[5]We use the superscript "2" in $\mathbf{\Phi}_\mathbb{X}^2$ to emphasise that $\mathbf{\Phi}_\mathbb{X}^2$ is induced by taking usual path development twice.

$\mathbf{\Phi}_{\mathbb{X}}^2$ is said to be a HRPCF for $\mathbb{X}$ as it satisfies the following characteristicity of synonym for filtered processes (see Definition 2.1). For a detailed proof please check the Appendix A.

**Theorem 3.3** (Characteristicity of synonym). *Two filtered processes $\mathbb{X}$ and $\mathbb{Y}$ are synonymous if and only if they have the same high rank PCF, that is, $\mathbf{\Phi}_{\mathbb{X}}^2(M, \mathcal{M}) = \mathbf{\Phi}_{\mathbb{Y}}^2(M, \mathcal{M}), \forall (M, \mathcal{M}) \in \mathcal{A}_{\text{unitary}}$.*

### 3.3 A New Distance induced by High Rank PCF

In this subsection, we will use the second rank PCF to define a distance on FP, which can (locally) characterize the extended weak convergence, as the classical PCFD introduced in subsection 2.2 can metrise the weak topology on FP.

**Definition 3.4.** *For two filtered processes $\mathbb{X}$ and $\mathbb{Y}$, let $(\boldsymbol{M}, \boldsymbol{\mathcal{M}})$ be a random admissible pair in $\mathcal{A}_{\text{unitary}}$ with $\boldsymbol{M} \in \mathcal{L}(\mathbb{R}^d, \mathfrak{u}(n))$ for some $n$, and $\boldsymbol{\mathcal{M}} \in \mathcal{L}(\mathbb{C}^{n \times n}, \mathfrak{u}(m))$ for some $m$. The High Rank Path Characteristic Function-based distance, for short HRPCFD, between $\mathbb{X}$ and $\mathbb{Y}$ with respect to $P_{\boldsymbol{M}}$ and $P_{\boldsymbol{\mathcal{M}}}$ is defined by*

$$HRPCFD_{\boldsymbol{M}, \boldsymbol{\mathcal{M}}}^2(\mathbb{X}, \mathbb{Y}) = \int \int d_{HS}^2(\mathbf{\Phi}_{\mathbb{X}}^2(M, \mathcal{M}), \mathbf{\Phi}_{\mathbb{Y}}^2(M, \mathcal{M})) P_{\boldsymbol{M}}(dM) P_{\boldsymbol{\mathcal{M}}}(d\mathcal{M}),$$

*where $d_{HS}(\cdot, \cdot)$ denotes the Hilbert-Schmidt distance[6] on $\mathbb{C}^{m \times m}$.*

As previously mentioned in the introduction, the so-defined HRPCFD shares the same analytic properties as the classical PCF, e.g., the separation of points, boundedness and the MMD property, whose proof can be found in Appendix A. Moreover, it metrises a much stronger topology (the extended weak convergence). as shown in the next theorem.

**Theorem 3.5.** *Suppose $(\mathbb{X}^i)_{i \in \mathbb{N}}$ and $\mathbb{X}$ are filtered processes whose laws $P_{X^i}$ and $P_X$ are supported in a compact subset of $\mathcal{X}$. Then $\mathbb{X}^i \xrightarrow{EW} \mathbb{X}$ iff $\widetilde{HRPCFD}(\mathbb{X}^i, \mathbb{X}) \to 0$, where*

$$\widetilde{HRPCFD}(\mathbb{X}^i, \mathbb{X}) := \sum_{j=1}^{\infty} \frac{\min\{1, HRPCFD_{\boldsymbol{M}_j, \boldsymbol{\mathcal{M}}_j}(\mathbb{X}^i, \mathbb{X})\}}{2^j}$$

*where the sequence $(\boldsymbol{M}_j, \boldsymbol{\mathcal{M}}_j)_{j \in \mathbb{N}}$ satisfies that for any $(n, m) \in \mathbb{N}^2$ there is a $j \in \mathbb{N}$ such that $\boldsymbol{M}_j \in \mathcal{L}(\mathbb{R}^d, \mathfrak{u}(n))$ and $\boldsymbol{\mathcal{M}}_j \in \mathcal{L}(\mathbb{C}^{n \times n}, \mathfrak{u}(m))$ and $P_{\boldsymbol{M}_j}, P_{\boldsymbol{\mathcal{M}}_j}$ have full supports for all $j \in \mathbb{N}$.*

We provide a concrete example in the last paragraph of Appendix A.1 to verify the fact that HRPCFD really reflects the differences of filtrations via an explicit computation.

## 4 Methodology

In this section, let $\mathbb{X}$ and $\mathbb{Y}$ be two filtered processes with the law $P_X, P_Y \in \mathcal{P}(\mathcal{X})$, let $\mathbf{X} = (\boldsymbol{x}_i)_{i=1}^N \sim P_X$ and $\mathbf{Y} = (\boldsymbol{y}_i)_{i=1}^N \sim P_Y$ be sample paths.

### 4.1 Estimating conditional probability measure and HRPCF

A fundamental question is to estimate the conditional probability measure $\hat{X}_t = \mathbb{P}(X \in \cdot | \mathcal{F}_t)$ from the finitely many data $(\boldsymbol{x}_i)_{i=1}^N$, in particular the random variable $\mathbf{\Phi}_{\hat{X}_t}(M) = \mathbb{E}_{\mathbb{P}}[\mathcal{U}_M(X) | \mathcal{F}_t]$ for any $M \in \mathcal{L}(\mathbb{R}^d, \mathfrak{u}(n))$. We solve this problem by conducting a regression. Fix $M$ we learn a sequence-to-sequence model $F_\theta^X : \mathbb{R}^{d \times (T+1)} \to \mathbb{C}^{n \times n \times (T+1)}$, where the input and output pairs are $(\mathbf{X}_{[0,T]}, \mathcal{U}_M(\mathbf{X}_{[t,T]})_{t=0}^T)$. More specifically, we optimize the model parameters of $F_\theta^X$ by minimizing the loss function:

$$\text{RLoss}(\theta; \boldsymbol{x}, M) = \sum_{t=0}^{T} \sum_{\boldsymbol{x} \in \mathbf{X}} d_{HS}^2(F_\theta^X(\boldsymbol{x}_{[0,T]})_t, \mathcal{U}_M(\boldsymbol{x}_{[t,T]})). \tag{4}$$

It is worth noting that the choice of $F_\theta^X$ must be autoregressive models to prevent information leakage. A detailed pseudocode is shown in Algorithm 1. Then, we approximate $\mathbf{\Phi}_{\mathbb{X}}^2$ using the trained regression model $F_\theta^X$ following the Algorithm 2. We denote by $\hat{\mathbf{\Phi}}_{\mathbb{X}}^2$ the estimation of $\mathbf{\Phi}_{\mathbb{X}}^2$.

---

[6]For $A, B \in \mathbb{C}^{m \times m}$, $d_{HS}^2(A, B) = \text{tr}((A - B)(A - B)^*)$.

## 4.2 Optimizing HRPCFD

In most empirical applications as we will show in Section 5, we employ HRPCFD as a discriminator under the GAN setting. That is, we optimize the loss function $\sup_{M,\mathcal{M}} \text{HRPCFD}^2_{M,\mathcal{M}}(\mathbb{X}, \mathbb{Y})$. We would approximate the pair of random variables $(M, \mathcal{M})$ by discrete random variables $M_{K_1} = \frac{1}{K_1} \sum_{i=1}^{K_1} M_i$ and $\mathcal{M}_{K_2} = \frac{1}{K_2} \sum_{i=1}^{K_2} \mathcal{M}_i$, parametrized by $M_i \in \mathcal{L}(\mathbb{R}^d, \mathfrak{u}(n))$ and $\mathcal{M}_i \in \mathcal{L}(\mathbb{C}^{n \times n}, \mathfrak{u}(m))$, $K_1, K_2 \in \mathbb{N}$ and optimize so-called Empirical HRPCFD

$$\text{EHRPCFD}^2_{M_{K_1}, \mathcal{M}_{K_2}}(\mathbb{X}, \mathbb{Y}) = \frac{1}{K_1 K_2} \sum_{i=1}^{K_1} \sum_{j=1}^{K_2} d^2_{\text{HS}}(\hat{\boldsymbol{\Phi}}^2_{\mathbb{X}}(M_i, \mathcal{M}_j), \hat{\boldsymbol{\Phi}}^2_{\mathbb{Y}}(M_i, \mathcal{M}_j)). \quad (5)$$

In practice, the joint training on both $M_{K_1}$ and $\mathcal{M}_{K_2}$ is computationally expensive and prone to overfitting. We alleviate this problem by splitting the optimization procedure in the following three steps: 1) Optimize $(M_i)_{i=1}^{K_1}$ to maximize $\text{EPCFD}^2_{M_{K_1}}(\mathbf{X}, \mathbf{Y}) = \frac{1}{K_1} \sum_{i=1}^{K_1} d^2_{\text{HS}}(\boldsymbol{\Phi}_{\mathbf{X}}(M_i), \boldsymbol{\Phi}_{\mathbf{Y}}(M_i))$ ($\boldsymbol{\Phi}_{\mathbf{X}}(M) = \frac{1}{N} \sum_{i=1}^{n} \mathcal{U}_M(x_i)$)[19, Section 3.3] , denote by $M^*_{K_1} = (M^*_i)_{i=1}^{K_1}$ the optimized linear maps. 2) Train regression modules $F^X_{\theta_i}, F^Y_{\theta_i}$ for each $M^*_i$ using data sampled from $P_X$ and $P_Y$ respectively. 3) Optimize $(\mathcal{M}_i)_{i=1}^{K_2}$ to maximize $\text{EHRPCFD}^2_{M^*_{K_1}, \mathcal{M}_{K_2}}(\mathbb{X}, \mathbb{Y})$.

The reason behind it is natural: the optimal set $(M^*_i)_{i=1}^{K_1}$ captures the most relevant information that discriminates the distribution $P_X$ from $P_Y$. This difference is reflected in the design of higher rank expected path developments through regression models specifically trained for this purpose. Finally, the HRPCFD based on $(M^*_i)_{i=1}^{K_1}$ tends to be more significant among other choices of $(M_i)_{i=1}^{K_1}$, making it a stronger discriminator.

## 4.3 HRPCF-GAN for conditional time series generation

Following [16, 13], we consider the task of conditional time series generation to simulate the law of the future path $\mathbf{X}_{\text{future}} := \mathbf{X}_{(p,T]}$ given the past path $\mathbf{X}_{\text{past}} := \mathbf{X}_{[0,p]}$ from samples of $\mathbf{X}$. To this end, we propose the so-called HRPCF-GAN by leveraging the autoregressive generator and the trainable HRPCFD as the discriminator. See Figure 2 for the flowchart illustration.

**Conditional autoregressive generator** To simulate future time series of length $T - p$, we construct a generator $G_\theta$ based on the step-1 conditional generator $g_\theta$ following [16]. This generator, $g_\theta : \mathcal{X}_{\text{past}} \times \mathcal{Z} \to \mathbb{R}^d$, aims to produce a random variable approximating $\mathbb{P}(X_{t+1}|\mathcal{F}_t)$. By applying $g_\theta$ inductively, we can simulate future paths of arbitrary length. To address the limitation of AR-RNN generator proposed in [16], where $\mathbb{P}(X_{t+1}|\mathcal{F}_t)$ depends solely on $p$-lagged values of $X_t$, we incorporate an embedding module. This module efficiently extracts past path information into a low-dimensional latent space. The output of this embedding module, along with the noise vector, serves as the input for $g_\theta$ to generate subsequent steps in the fake time series. Further details of our proposed generator are provided in Appendix B.2.

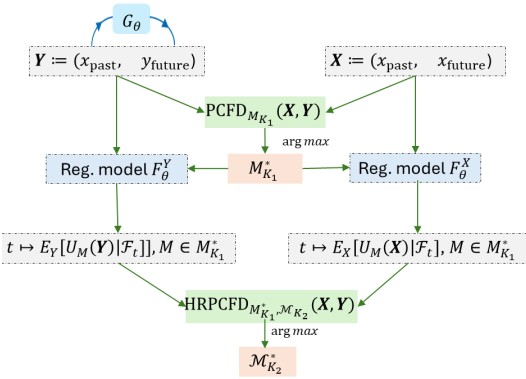

Figure 2: Flowchart of HRPCF-GAN for learning condition distribution $\mathbb{P}(X_{\text{future}}|X_{\text{past}})$.

**High Rank development discriminator** To capture the conditional law, we use the HRPCFD as the discriminator of joint law of $(\mathbf{X}_{\text{past}}, \mathbf{X}_{\text{future}})$ under true and fake measures. Here the empirical measures of $M_{K_1}$ and $\mathcal{M}_{K_2}$ are model parameters of the discriminator, which are optimized by the following maximization:

$$\max_{M_{K_1}, \mathcal{M}_{K_2}} \text{EHRPCFD}^2_{M_{K_1}, \mathcal{M}_{K_2}}(\mathbf{X}_{[0,T]}, (\mathbf{X}_{[0,p]}, G_\theta(\mathbf{X}_{[0,p]}, z))),$$

In principle, one can generate the fake data by the generator via Monte Carlo and apply the training procedure outlined in Section 4.2 for training the generative model. However, it would be computa-

tionally infeasible due to the need for recalibration of the regression module per generator update. To enhance the training efficiency for the regression module under the fake measure, we use the gradient descent method with efficient initialization obtained by the trained regression model under real data. For each generator, the corresponding regression model parameters are then updated to minimize the RLoss (Section 4.1) on a batch of newly generated samples by $G_\theta$. The detailed algorithm is described in Algorithm 3.

# 5 Numerical results

## 5.1 Hypothesis testing

To showcase the power of EHRPCFD in discriminating laws of stochastic processes, we use it as the test statistic in the permutation test. Similar experiments have been done in [21, 15]. By regarding the permutation test as a decision rule, we assess its performance via computing its *power* (probability of correctly rejecting the null hypothesis) and *type-I error* (probability of falsely rejecting the null hypothesis). Similar to [15], we compare the law of 3-dimensional Brownian motion $B$ with the set of laws of 3-dimensional fractional Brownian motion $B^H$ with Hurst parameter $H$ ranging from $[0.4, 0.6]$. Details of the methodology and implementation can be found in Appendix C.1.

**Baselines** We compare the performance of HRPCFD with other test metrics including 1) the linear and RBF signature MMDs [8, 20] and its high-rank derivative, namely High Rank signature MMDs [21]; 2) Classical vector MMDs; 3) PCFD [15, 19].

As shown in Table 1 of the test power, HRPCFD consistently outperforms other models, especially when $H$ is close to 0.5. We do see an improvement from the vanilla PCFD by considering a stronger topology. Furthermore, comparing HRPCFD and High Rank signature MMD, we observe a distinct advantage for HRPCFD. This may be due to the challenge of capturing the conditional probability measure, as High Rank signature MMD relies on linear regression for estimation, whereas we obtained a better estimation using a non-linear approach. Additional test metrics such type-I error and computational cost can be found in Appendix C.1.

| | Developments | | Signature MMDs | | | Classical MMDs | |
|---|---|---|---|---|---|---|---|
| $H$ | High Rank PCFD | PCFD | Linear | RBF | High Rank | Linear | RBF |
| 0.4 | $\mathbf{1 \pm 0}$ | $\mathbf{1 \pm 0}$ | $0.09 \pm 0.06$ | $0.97 \pm 0.03$ | $0.22 \pm 0.07$ | $0.05 \pm 0.04$ | $0.97 \pm 0.04$ |
| 0.425 | $\mathbf{1 \pm 0}$ | $\mathbf{1 \pm 0}$ | $0.1 \pm 0.05$ | $0.69 \pm 0.11$ | $0.14 \pm 0.10$ | $0.01 \pm 0.02$ | $0.58 \pm 0.10$ |
| 0.45 | $0.97 \pm 0.04$ | $\mathbf{0.99 \pm 0.02}$ | $0.04 \pm 0.04$ | $0.15 \pm 0.05$ | $0.14 \pm 0.08$ | $0.06 \pm 0.05$ | $0.24 \pm 0.08$ |
| 0.475 | $\mathbf{0.31 \pm 0.13}$ | $0.06 \pm 0.02$ | $0.01 \pm 0.02$ | $0.04 \pm 0.02$ | $0.12 \pm 0.04$ | $0.01 \pm 0.02$ | $0.02 \pm 0.02$ |
| 0.525 | $\mathbf{0.30 \pm 0.20}$ | $0.08 \pm 0.02$ | $0.05 \pm 0.02$ | $0.07 \pm 0.04$ | $0.19 \pm 0.04$ | $0.08 \pm 0.04$ | $0.09 \pm 0.04$ |
| 0.55 | $\mathbf{0.99 \pm 0.02}$ | $0.95 \pm 0.03$ | $0.13 \pm 0.05$ | $0.17 \pm 0.04$ | $0.18 \pm 0.08$ | $0.06 \pm 0.06$ | $0.19 \pm 0.11$ |
| 0.575 | $\mathbf{1 \pm 0}$ | $\mathbf{1 \pm 0}$ | $0.07 \pm 0.02$ | $0.5 \pm 0.10$ | $0.14 \pm 0.10$ | $0.10 \pm 0.10$ | $0.48 \pm 0.15$ |
| 0.6 | $\mathbf{1 \pm 0}$ | $\mathbf{1 \pm 0}$ | $0.05 \pm 0.03$ | $0.75 \pm 0.05$ | $0.22 \pm 0.05$ | $0.06 \pm 0.06$ | $0.67 \pm 0.14$ |

Table 1: Test power of the distances when $h \neq 0.5$ in the form of mean $\pm$ std over 5 runs. After careful grid search, we set optimal $\sigma = \sqrt{0.05}$ for the RBF signature MMD and classical RBF MMD, whereas $\sigma_1 = \sigma_2 = 1$ for High Rank signature MMD.

## 5.2 Generative modeling

To validate the effectiveness of our proposed HRPCF-GAN, we consider the task of learning the law of future time series conditional on its past time series.

**Dataset** We benchmark our model on both synthetic and empirical datasets. 1) multivariate fractional Brownian Motion (fBM) with Hurst parameter $H = 1/4$: this dataset exhibits non-Markovian properties and high oscillation. 2) Stock dataset: We collected the daily log return of 5 representative stocks in the U.S. market from 2010 to 2020, sourced from Yahoo Finance.

**Baseline** We compare the performance of HRPCF-GAN with well-known models for time-series generation such as RCGAN [10] and TimeGAN [23]. Furthermore, we use PCFGAN [19] as a benchmarking model to showcase the significant improvement by considering the higher rank

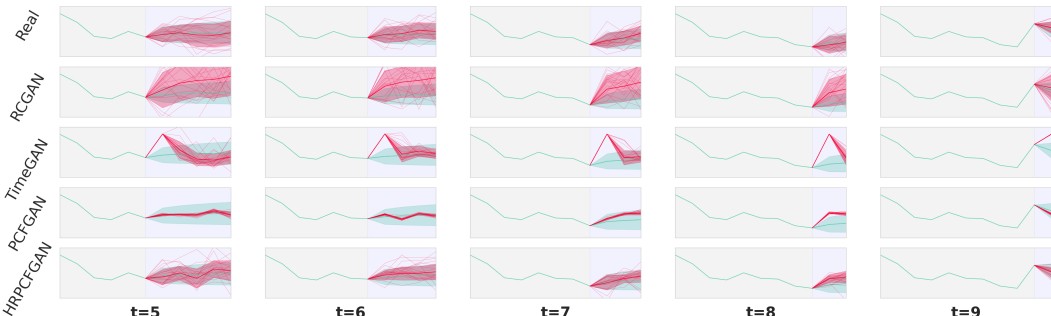

Figure 3: Sample plots of the conditional distribution $\mathbb{P}(X|\mathcal{F}_t)$ on fBM conditioned on the same past path, using both true and GAN models (arranged from top to bottom). Each column represents different $t$. The thick red /green line indicates the conditional mean of the future path estimated by model simulated samples/true models. The shaded red area presents the region of $\pm$std of model simulated samples, whereas the shaded area shown corresponds to the region of $\pm$ theoretical std.

development as the discriminator. For fairness, we use the same generator structure (LSTM-based) for all these models.

**Test metrics** To assess the fidelity, usefulness, and diversity of synthetic time series, we consider 7 test metrics, including Auto-Correlation, Cross-Correlation, Discriminative Score, Sig-$W_1$ Distance, and Conditional Expectation. For the stock dataset, we also consider a test metric based on American option pricing. A detailed definition of these test metrics can be found in Appendix C.2.

We summarize in Table 2 the performance comparison between HRPCF-GAN and benchmarking models. For both datasets, HRPCF-GAN consistently outperforms the other models. Focusing on the fBM dataset, HRPCF-GAN achieves the lowest Auto-Correlation (.082) and Cross-Correlation (0.013), which is approximately 21.9%/72.3% lower than the second-best model, indicating better performance in fitting the dynamics of the underlying process across time and feature dimensions. We also observe strong evidence in capturing the conditional probability measure as HRPCFGAN achieves the lowest Conditional Expectation score (1.693 on fBM and 0.56 on Stock). Furthermore, we observe on average an improvement of 34%/14% of HRPCF-GAN with respect to PCFGAN on fBM/Stock datasets respectively. The strong empirical results demonstrate the effectiveness of considering high rank path development to capture the filtration of stochastic processes. Finally, HRPCF-GAN attained the best estimation of an at-the-money American put option, which demonstrates its potential usage for optimal stopping problems in finance. Sample plots from all models conditioned on the same path are also shown in Figures 3, 6 and 7 for a qualitative analysis of generative quality. For additional test metrics, we refer readers to Table 5.

| Dataset | Test Metrics | RCGAN | TimeGAN | PCFGAN | HRPCF-GAN |
|---------|--------------|-------|---------|--------|-----------|
| fBM | Auto-C. | .105±.001 | .459±.003 | .125±.003 | **.082±.002** |
| | Cross-C. | .051±.001 | .092±.001 | .047±.001 | **.013±.001** |
| | Discriminative | .207±.008 | .480±.002 | .265±.006 | **.151±.006** |
| | Sig-$W_1$ | .512±.006 | .341±.011 | .199±.004 | **.169±.009** |
| | Cond. Exp. | 1.822±.023 | 2.265±.029 | 2.278±.033 | **1.693±.021** |
| Stock | Auto-C. | .239±.016 | .228±.010 | .198±.003 | **.189±.010** |
| | Cross-C. | .067±.011 | .056±.002 | .055±.004 | **.053±.005** |
| | Discriminative | .134±.058 | .020±.021 | .028±.017 | **.016±.005** |
| | Sig-$W_1$ | .013±.002 | .008±.001 | .005±.001 | **.004±.002** |
| | Cond. Exp. | .078±.003 | .079±.001 | .060±.001 | **.056±.002** |
| | Amer. Put | .546±.318 | .243±.411 | .202±.020 | **.179±.006** |

Table 2: Performance comparison of HRPCF-GAN and baselines. The best for each task is shown in bold. Each test metric is shown in the form of mean±std over 5 runs.

# 6 Conclusion and Future work

**Conclusion:** In this paper, we apply the unitary feature from rough path theory to define the CF for measure-valued paths, which further induces a distance (HRPCFD) for metrising the extended weak convergence. Theoretically, we prove the key properties of HRPCFD, such as characteristicity, uniform boundedness, etc. Additionally, the numerical experiments validate the out-performance of the approach based on HRPCFD compared with several state-of-the-art GAN models for tasks such as hypothesis testing and synthetic time series generation.

**Limitation and Future work:** The suitable choice of network architecture for generating data is crucial in the proposed HRPCF-GAN, which merits further investigation; in particular, it will be interesting to understand how the network architecture impacts the filtration structure of the generated stochastic process. Furthermore, there is room for further improvement on the estimation method of conditional expectation in terms of accuracy and training stability. Possible routes include exploring the interplay between the regression module and the generator.

**Broader impacts:** Our approach based on the extended weak convergence has the potential in many important financial and economic applications, such as optimal stopping, utility maximisation and stochastic programming. Unlike classical methods built on top of parametric stochastic differential equations, our non-parametric and data-driven method alleviates the risk of the model mis-specification, providing better solution to complex, real-world multi-period decision making problems. However, like other synthetic data generation models, it also poses risks of misuse, e.g., misrepresenting the synthetic data as real data.

## Acknowledgments and Disclosure of Funding

HN is supported by the EPSRC under the program grant EP/S026347/1 and the Alan Turing Institute under the EPSRC grant EP/N510129/1. HN extends her gratitude to Terry Lyons and Hang Lou for insightful discussions. Moreover, HN is grateful to Jing Liu for her help with Figure 1. CL is supported by the National Key Research and Development Program of China: Young Scientist Project 2023YFA1010900.

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

# A Examples and Proofs

## A.1 Examples related to extended weak convergence

**Prediction processes**  First let us give an explicit example for prediction processes of some simple filtered processes. Consider the two processes $\mathbb{X}^n = (\Omega^n, \mathcal{F}^n, \mathbb{F}^n, X^n, \mathbb{P}_n)$ and $\mathbb{X} = (\Omega, \mathcal{F}, \mathbb{F}, X, \mathbb{P})$, where

- $\Omega^n = \{\boldsymbol{x}_1^n, \boldsymbol{x}_2^n\}$, $\boldsymbol{x}_1^n = (\boldsymbol{x}_1^n(0) = 1, \boldsymbol{x}_1^n(1) = 1 + \frac{1}{n}, \boldsymbol{x}_1^n(2) = 2)$ and $\boldsymbol{x}_2^n = (\boldsymbol{x}_2^n(0) = 1, \boldsymbol{x}_2^n(1) = 1 - \frac{1}{n}, \boldsymbol{x}_2^n(2) = 0)$;
- $X_t^n(\boldsymbol{x}_i^n) = \boldsymbol{x}_i^n(t)$ for $t = 0, 1, 2$ and $i = 1, 2$ is the coordinate process on $\Omega^n$;
- $\mathbb{P}^n(\boldsymbol{x}_1^n) = \mathbb{P}^n(\boldsymbol{x}_2^n) = \frac{1}{2}$;
- $\mathbb{F}^n = (\mathcal{F}_0^n, \mathcal{F}_1^n, \mathcal{F}_2^n)$ is the natural filtration generated by $X^n$: $\mathcal{F}_0^n = \{\emptyset, \Omega^n\}$ and $\mathcal{F}_1^n = \mathcal{F}_2^n = \sigma(X_1^n, X_2^n)$ is the power set of $\Omega^n$,

and

- $\Omega = \{\boldsymbol{x}_1, \boldsymbol{x}_2\}$, $\boldsymbol{x}_1 = (\boldsymbol{x}_1(0) = 1, \boldsymbol{x}_1(1) = 1, \boldsymbol{x}_1(2) = 2)$ and $\boldsymbol{x}_2 = (\boldsymbol{x}_2(0) = 1, \boldsymbol{x}_2(1) = 1, \boldsymbol{x}_2(2) = 0)$;
- $X_t(\boldsymbol{x}_i^n) = \boldsymbol{x}_i(t)$ for $t = 0, 1, 2$ and $i = 1, 2$ is the coordinate process on $\Omega$;
- $\mathbb{P}(\boldsymbol{x}_1) = \mathbb{P}(\boldsymbol{x}_2) = \frac{1}{2}$;
- $\mathbb{F} = (\mathcal{F}_0, \mathcal{F}_1, \mathcal{F}_2)$ is the natural filtration generated by $X$: $\mathcal{F}_0 = \mathcal{F}_1 = \{\emptyset, \Omega\}$ and $\mathcal{F}_2 = \sigma(X_1, X_2)$ is the power set of $\Omega$.

We plot the sample paths of $\mathbb{X}^n$ and $\mathbb{X}$ in Fig. 4.

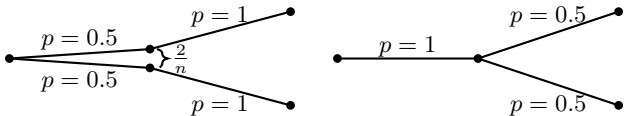

Figure 4: $\mathbb{X}^n$ (left) converges to $\mathbb{X}$ (right) weakly, but the corresponding price of American options on $\mathbb{X}^n$ cannot converge to the counterpart on $\mathbb{X}$, see Example A.1 below. Therefore the usage of slightly erroneous models in weak topology may cause significant loss in decision making problems. This example is taken from [3] and [5].

From the above, it is straightforward to check that the prediction process $\hat{X}^n$ of $\mathbb{X}^n$ is
$$\hat{X}_0^n(\boldsymbol{x}_1^n) = \hat{X}_0^n(\boldsymbol{x}_2^n) = \mathbb{P}^n(X^n \in \cdot | \mathcal{F}_0^n) = P_{X^n},$$
where $P_{X^n}$ is the law of $X^n$ under $\mathbb{P}^n$;
$$\hat{X}_1^n(\boldsymbol{x}_1^n) = \mathbb{P}^n(X^n \in \cdot | \mathcal{F}_1^n)(\boldsymbol{x}_1^n) = \delta_{\boldsymbol{x}_1^n}, \quad \hat{X}_1^n(\boldsymbol{x}_2^n) = \mathbb{P}^n(X^n \in \cdot | \mathcal{F}_1^n)(\boldsymbol{x}_2^n) = \delta_{\boldsymbol{x}_2^n},$$
where $\delta_{\boldsymbol{x}_i^n}$ ($i = 1, 2$) denotes the Dirac measure at $\boldsymbol{x}_i^n$; and
$$\hat{X}_2^n(\boldsymbol{x}_1^n) = \mathbb{P}^n(X^n \in \cdot | \mathcal{F}_2^n)(\boldsymbol{x}_1^n) = \delta_{\boldsymbol{x}_1^n}, \quad \hat{X}_2^n(\boldsymbol{x}_2^n) = \mathbb{P}^n(X^n \in \cdot | \mathcal{F}_2^n)(\boldsymbol{x}_2^n) = \delta_{\boldsymbol{x}_2^n}.$$
Consequently, it holds that the law of $\hat{X}^n$ satisfies
$$P_{\hat{X}^n} = \mathbb{P}^n(\hat{X}^n = (P_{X^n}, \delta_{\boldsymbol{x}_i^n}, \delta_{\boldsymbol{x}_i^n})) = \frac{1}{2}, \quad i = 1, 2.$$
Similarly, the prediction process $\hat{X}$ of $\mathbb{X}$ is
$$\hat{X}_0(\boldsymbol{x}_1) = \hat{X}_0(\boldsymbol{x}_2) = \mathbb{P}(X \in \cdot | \mathcal{F}_0) = P_X,$$
where $P_X$ is the law of $X$ under $\mathbb{P}$;
$$\hat{X}_1(\boldsymbol{x}_1) = \mathbb{P}(X \in \cdot | \mathcal{F}_1)(\boldsymbol{x}_1) = P_X, \quad \hat{X}_1(\boldsymbol{x}_2) = \mathbb{P}(X \in \cdot | \mathcal{F}_1)(\boldsymbol{x}_2) = P_X;$$
and
$$\hat{X}_2(\boldsymbol{x}_1) = \mathbb{P}(X \in \cdot | \mathcal{F}_2)(\boldsymbol{x}_1) = \delta_{\boldsymbol{x}_1}, \quad \hat{X}_2(\boldsymbol{x}_2) = \mathbb{P}(X \in \cdot | \mathcal{F}_2)(\boldsymbol{x}_2) = \delta_{\boldsymbol{x}_2}.$$
so that the law of $\hat{X}$ reads
$$P_{\hat{X}} = \mathbb{P}(\hat{X} = (P_X, P_X, \delta_{\boldsymbol{x}_i})) = \frac{1}{2}, \quad i = 1, 2.$$

**Test functions for extended weak convergence**   For $I = \{0, 1, \ldots, T\}$ and filtered process $\mathbb{X} \in \text{FP}$ on $I$, the typical test functions for defining the extended weak convergence have the following form:

$$\hat{f}(\hat{X}) = F(\mathbb{E}_{\mathbb{P}}[f_0(X)|\mathcal{F}_0], \ldots, \mathbb{E}[f_T(X)|\mathcal{F}_T]),$$

where $f_0, \ldots, f_T \in C_b(\mathcal{X})$ are continuous bounded functions on the path space $\mathcal{X}$ and $F \in C_b(\mathbb{R}^{T+1})$. For instance, for the filtered processes $\mathbb{X}^n$ and $\mathbb{X}$ in the above example, we have $T = 2$, and by choosing $f_0(x_0, x_1, x_2) = 1$, $f_1(x_0, x_1, x_2) = x_1 + x_2 - 2$, $f_3(x_0, x_1, x_2) = \sin(x_2 - x_1)$ and $F(y_0, y_1, y_2) = \exp(-|y_1| - y_2^2)$, in view of the facts that $\mathcal{F}_1^n = \mathcal{F}_2^n$ are the power set of $\Omega^n$ (see the last paragraph), we obtain that for each $n$,

$$
\begin{aligned}
\hat{f}(\hat{X}^n(\boldsymbol{x}_i^n)) &= \exp(-|\mathbb{E}_{\mathbb{P}^n}[X_1^n + X_2^n - 2|\mathcal{F}_1^n](\boldsymbol{x}_i^n)| - (\mathbb{E}_{\mathbb{P}^n}[\sin(X_2^n - X_1^n)|\mathcal{F}_2^n](\boldsymbol{x}_2^n))^2) \\
&= \exp(-|\boldsymbol{x}_i^n(1) + \boldsymbol{x}_i^n(2) - 2| - \sin^2(\boldsymbol{x}_i^n(2) - \boldsymbol{x}_i^n(1))) \\
&= \exp(-(1 + \frac{1}{n}) - \sin^2(1 - \frac{1}{n})), \quad i = 1, 2;
\end{aligned}
$$

and therefore $\mathbb{E}_{\mathbb{P}^n}[\hat{f}(\hat{X}^n)] = \exp(-(1 + \frac{1}{n}) - \sin^2(1 - \frac{1}{n}))$. On the other side, since $\mathcal{F}_0 = \mathcal{F}_1 = \{\emptyset, \Omega\}$ are trivial $\sigma$-algebra, for the prediction process $\hat{X}$ of $\mathbb{X}$ we have

$$
\begin{aligned}
\hat{f}(\hat{X}(\boldsymbol{x}_i)) &= \exp(-|\mathbb{E}_{\mathbb{P}}[X_1 + X_2 - 2|\mathcal{F}_1](\boldsymbol{x}_i)| - (\mathbb{E}_{\mathbb{P}}[\sin(X_2 - X_1)|\mathcal{F}_2](\boldsymbol{x}_2))^2) \\
&= \exp(-|\mathbb{E}_{\mathbb{P}}[X_1 + X_2 - 2]| - \sin^2(\boldsymbol{x}_i(2) - \boldsymbol{x}_i(1))) \\
&= \exp(-\sin^2(1)), \quad i = 1, 2
\end{aligned}
$$

as $\mathbb{E}_{\mathbb{P}}[X_1 + X_2 - 2] = 0$; and therefore

$$\mathbb{E}_{\mathbb{P}}[\hat{f}(\hat{X})] = \exp(-\sin^2(1)).$$

Clearly, as $n \to \infty$, we have $\mathbb{E}_{\mathbb{P}^n}[\hat{f}(\hat{X}^n)] = \exp(-(1 + \frac{1}{n}) - \sin^2(1 - \frac{1}{n})) \to \exp(-1 - \sin^2(1)) \neq \exp(-\sin^2(1)) = \mathbb{E}_{\mathbb{P}}[\hat{f}(\hat{X})]$, which shows that $\mathbb{X}^n$ cannot converge to $\mathbb{X}$ in the extended weak convergence according to Definition 2.1, although it is easy to see that the laws of $\mathbb{X}^n$ converges to the law of $\mathbb{X}$ in the weak topology.

In the example, while the unconditional law of the processes $\mathbb{X}^n$ converges to $\mathbb{X}$, weak convergence fails to capture a key difference between the financial models $\mathbb{X}^n$ and $\mathbb{X}$. Specifically, if an agent believes the market dynamics as in $\mathbb{X}^n$, he/she always knows the outcome of the last day in advance, granting a predictive advantage, whereas in the "fair" market $\mathbb{X}$, the agent lacks this foresight. This crucial difference in the observed information flow-"No knowledge $\Rightarrow$ Full knowledge $\Rightarrow$ Full knowledge" for $\mathbb{X}^n$ versus "No knowledge $\Rightarrow$ No knowledge $\Rightarrow$ Full knowledge" for $\mathbb{X}$—is not reflected in weak convergence alone.

*Extended Weak Topology* (EWT) is vital in this case, because it captures this difference through the conditional distributions. For markets $\mathbb{X}^n$, where the agent has full information on day 1, the conditional distribution becomes a single Dirac measure, annihilating randomness. In contrast, $\mathbb{X}$ retains genuine randomness at day 1, as reflected by a linear combination of Dirac measures. Since EWT is based on conditional distributions, it effectively measures differences in information evolution styles, ensuring continuity in multi-period decision-making as agents update their actions based on continually evolving information.

**Some important multi-periods optimisation problems**   The following multi-periods optimisation problems are very important in financial and economic applications, whose value functions are, in general, discontinuous with respect to the weak convergence, but continuous in the extended weak topology.

**Example A.1** (Optimal Stopping Problem). *Let $g : I \times \mathcal{X} \to \mathbb{R}$ be a continuous and bounded non-anticipative (i.e., for any $t \in I$ and $\boldsymbol{x} \in \mathcal{X}$, the value of $g(t, \boldsymbol{x})$ only depends on $\boldsymbol{x}_0, \ldots, \boldsymbol{x}_t$) function. For each filtered process $\mathbb{X}^n$ we set $ST_n := \{\tau : \mathbb{F}^n\text{-stopping time}\}$ be the collection of all stopping times with respect to the filtration $\mathbb{F}^n$ and similarly define ST for $\mathbb{X}$. Then the value function $v_g(\cdot)$ in the Optimal Stopping Problem (OSP) with the reward $g$ (in the context of mathematical finance, it is also called the price of American option) is defined by*

$$v_g(\mathbb{X}^n) = \sup_{\tau \in ST_n} \mathbb{E}_{\mathbb{P}^n}[g(\tau, X^n)], \quad v_g(\mathbb{X}) = \sup_{\tau \in ST} \mathbb{E}_{\mathbb{P}}[g(\tau, X)].$$

*If $\mathbb{X}^n \xrightarrow{EW} \mathbb{X}$, then $v_g(\mathbb{X}^n) \to v_g(\mathbb{X})$, whilst this continuity fails in the weak convergence: for the processes $\mathbb{X}^n$ and $\mathbb{X}$ considered as above (see Fig. 4) and the reward function $g(t, \boldsymbol{x}) := \boldsymbol{x}_t$, one has $\mathbb{X}^n \xrightarrow{W} \mathbb{X}$ but*

$$\lim_{n \to \infty} v_g(\mathbb{X}^n) \neq v_g(\mathbb{X}).$$

*Indeed, since $\mathbb{X}$ is a martingale with initial value 1, it is obvious that for any stopping time $\tau \in ST$ it always holds that $\mathbb{E}_{\mathbb{P}}[g(\tau, X)] = \mathbb{E}_{\mathbb{P}}[X_\tau] = 1$ which in turn implies that $v_g(\mathbb{X}) = 1$; on the other hand, since the filtration of $\mathbb{X}^n$ satisfies that $\mathcal{F}_1^n = \mathcal{F}_2^n$ (i.e., the agent already knows everything at day 1), it is easy to check that $\tau_\star^n = 2\mathbf{1}_{\boldsymbol{x}_1^n} + \mathbf{1}_{\boldsymbol{x}_2^n}$ is the optimal $\mathbb{F}^n$-stopping time for $v_g(\mathbb{X}^n)$ and consequently $v_g(\mathbb{X}^n) = \mathbb{E}_{\mathbb{P}^n}[X_{\tau_\star^n}^n] = \frac{1}{2} \times 2 + \frac{1}{2} \times (1 - \frac{1}{n}) = \frac{3}{2} - \frac{1}{2n}$ converges to $\frac{3}{2} \neq 0 = v_g(\mathbb{X})$.*

**Example A.2** (Utility Maximisation Problem)**.** *Let $g : \mathbb{R} \to \mathbb{R}$ be a continuous, bounded and concave utility function. For each filtered process $\mathbb{X}^n$ we set $\Lambda_n := \{\varphi = (\varphi_t)_{t=1,\dots,T} : \varphi \text{ is predictable w.r.t. } \mathbb{F}^n\}$ be the collection of all predictable strategies (i.e., $\varphi_t$ is $\mathcal{F}_{t-1}^n$-measurable for all $t = 1, \dots, T$) with respect to the filtration $\mathbb{F}^n$ and similarly define $\Lambda$ for $\mathbb{X}$. Then the value function $u_g(\cdot)$ in the utility maximisation Problem with the utility function $g$ is defined by*

$$u_g(\mathbb{X}^n) = \sup_{\varphi \in \Lambda_n} \mathbb{E}_{\mathbb{P}^n}\left[g(\int_0^T \varphi_t dX_t^n)\right], \quad u_g(\mathbb{X}) = \sup_{\varphi \in \Lambda} \mathbb{E}_{\mathbb{P}}\left[g(\int_0^T \varphi_t dX_t)\right],$$

*where $\int_0^T \varphi_t dX_t = \sum_{i=1}^T \varphi_t(X_t - X_{t-1})$ is the stochastic integral. If $\mathbb{X}^n \xrightarrow{EW} \mathbb{X}$, then $u_g(\mathbb{X}^n) \to u(\mathbb{X})$, whilst this continuity fails in the weak convergence.*

**An example of HRPCF**  We still consider the example mentioned before (see the paragraph **Prediction processes** and Fig. 4). In the previous discussions we have known that $\mathbb{X}^n \to \mathbb{X}$ in the weak convergence (i.e., the laws $P_{X^n}$ converges to the law $P_X$), but $\mathbb{X}^n$ cannot converge to $\mathbb{X}$ in the extended weak convergence. Now we show that there exists an admissible pair $(M, \mathcal{M}) \in \mathcal{A}_{\text{unitary}}$ such that

$$\lim_{n \to \infty} d_{\text{HS}}(\boldsymbol{\Phi}_{\mathbb{X}}(M, \mathcal{M}), \boldsymbol{\Phi}_{\mathbb{X}^n}(M, \mathcal{M})) \neq 0,$$

by an explicit calculation, which confirms that the HRPCFD does metrise the extended weak convergence and therefore reflect the differences of filtrations of stochastic processes.

Now we pick a linear operator $M \in \mathcal{L}(\mathbb{R}^2, \mathfrak{u}(1))$[7] which is given by $M(t, y) := y(\frac{\pi}{2}\mathrm{i}) \in \mathfrak{u}(1) \subset \mathbb{C}$, where i denotes the imaginary unit in $\mathbb{C}$. Since the prediction process $\hat{X}$ of $\mathbb{X}$ is

$$\hat{X}_0(\boldsymbol{x}_1) = \hat{X}_0(\boldsymbol{x}_2) = \mathbb{P}(X \in \cdot | \mathcal{F}_0) = P_X,$$

where $P_X$ is the law of $X$ under $\mathbb{P}$;

$$\hat{X}_1(\boldsymbol{x}_1) = \mathbb{P}(X \in \cdot | \mathcal{F}_1)(\boldsymbol{x}_1) = P_X, \quad \hat{X}_1(\boldsymbol{x}_2) = \mathbb{P}(X \in \cdot | \mathcal{F}_1^n)(\boldsymbol{x}_2) = P_X;$$

and

$$\hat{X}_2(\boldsymbol{x}_1) = \mathbb{P}(X \in \cdot | \mathcal{F}_2)(\boldsymbol{x}_1) = \delta_{\boldsymbol{x}_1}, \quad \hat{X}_2(\boldsymbol{x}_2) = \mathbb{P}(X \in \cdot | \mathcal{F}_2)(\boldsymbol{x}_2) = \delta_{\boldsymbol{x}_2},$$

we can check that

$$\mathbb{E}_{\mathbb{P}}[\mathcal{U}_M(X)|\mathcal{F}_0] = \mathbb{E}_{\mathbb{P}}[\mathcal{U}_M(X)] = \frac{1}{2}(e^{\frac{\pi}{2}\mathrm{i}} + e^{-\frac{\pi}{2}\mathrm{i}}) = 0,$$

$$\mathbb{E}_{\mathbb{P}}[\mathcal{U}_M(X)|\mathcal{F}_1] = \mathbb{E}_{\mathbb{P}}[\mathcal{U}_M(X)] = \frac{1}{2}(e^{\frac{\pi}{2}\mathrm{i}} + e^{-\frac{\pi}{2}\mathrm{i}}) = 0,$$

and

$$\mathbb{E}_{\mathbb{P}}[\mathcal{U}_M(X)|\mathcal{F}_2](\boldsymbol{x}_1) = \mathcal{U}_M(\boldsymbol{x}_1) = e^{\frac{\pi}{2}\mathrm{i}} = \mathrm{i},$$
$$\mathbb{E}_{\mathbb{P}}[\mathcal{U}_M(X)|\mathcal{F}_2](\boldsymbol{x}_2) = \mathcal{U}_M(\boldsymbol{x}_2) = e^{-\frac{\pi}{2}\mathrm{i}} = -\mathrm{i},$$

which shows that the $\mathbb{C}$-valued process $(\mathbb{E}_{\mathbb{P}}[\mathcal{U}_M(X)|\mathcal{F}_t])_{t=0,1,2}$ satisfies that

$$(\mathbb{E}_{\mathbb{P}}[\mathcal{U}_M(X)|\mathcal{F}_t](\boldsymbol{x}_1))_{t=0,1,2} = (0, 0, \mathrm{i}), \tag{6}$$

and

$$(\mathbb{E}_{\mathbb{P}}[\mathcal{U}_M(X)|\mathcal{F}_t](\boldsymbol{x}_2))_{t=0,1,2} = (0, 0, -\mathrm{i}). \tag{7}$$

---

[7]Recall that in the unitary representation of a path $\boldsymbol{x}$ we actually always consider the time-augmented version $(t, \boldsymbol{x}_t)$, so here the domain of $M$ for real valued path $\boldsymbol{x}$ is $\mathbb{R}^2$.

On the other hand, for every $n \in \mathbb{N}$, we have

$$\mathbb{P}^n[X^n \in \cdot | \mathcal{F}_0^n] = P_{X^n},$$

$$\mathbb{P}^n[X^n \in \cdot | \mathcal{F}_1^n](\boldsymbol{x}_1^n) = \mathbb{P}^n[X^n \in \cdot | \mathcal{F}_2^n](\boldsymbol{x}_1^n) = \delta_{\boldsymbol{x}_1^n},$$

and

$$\mathbb{P}^n[X^n \in \cdot | \mathcal{F}_1^n](\boldsymbol{x}_2^n) = \mathbb{P}^n[X^n \in \cdot | \mathcal{F}_2^n](\boldsymbol{x}_1^n) = \delta_{\boldsymbol{x}_2^n},$$

which provides that

$$\mathbb{E}_{\mathbb{P}^n}[\mathcal{U}_M(X^n)|\mathcal{F}_0^n] = \frac{1}{2}(e^{(1+\frac{1}{n})\frac{\pi i}{2}} + e^{-(1+\frac{1}{n})\frac{\pi i}{2}}),$$

$$\mathbb{E}_{\mathbb{P}^n}[\mathcal{U}_M(X^n)|\mathcal{F}_1^n](\boldsymbol{x}_1^n) = \mathbb{E}_{\mathbb{P}^n}[\mathcal{U}_M(X^n)|\mathcal{F}_2^n](\boldsymbol{x}_1^n) = \mathcal{U}_M(\boldsymbol{x}_1^n) = e^{(1+\frac{1}{n})\frac{\pi i}{2}},$$

and

$$\mathbb{E}_{\mathbb{P}^n}[\mathcal{U}_M(X^n)|\mathcal{F}_1^n](\boldsymbol{x}_2^n) = \mathbb{E}_{\mathbb{P}^n}[\mathcal{U}_M(X^n)|\mathcal{F}_2^n](\boldsymbol{x}_2^n) = \mathcal{U}_M(\boldsymbol{x}_2^n) = e^{-(1+\frac{1}{n})\frac{\pi i}{2}}.$$

Therefore, the $\mathbb{C}$-valued process $(\mathbb{E}_{\mathbb{P}^n}[\mathcal{U}_M(X^n)|\mathcal{F}_t^n])_{t=0,1,2}$ satisfies that

$$(\mathbb{E}_{\mathbb{P}^n}[\mathcal{U}_M(X^n)|\mathcal{F}_t^n](\boldsymbol{x}_1^n))_{t=0,1,2} = (\frac{1}{2}(e^{(1+\frac{1}{n})\frac{\pi i}{2}} + e^{-(1+\frac{1}{n})\frac{\pi i}{2}}), e^{(1+\frac{1}{n})\frac{\pi i}{2}}, e^{(1+\frac{1}{n})\frac{\pi i}{2}}), \quad (8)$$

and

$$(\mathbb{E}_{\mathbb{P}^n}[\mathcal{U}_M(X^n)|\mathcal{F}_t^n](\boldsymbol{x}_2^n))_{t=0,1,2} = (\frac{1}{2}(e^{(1+\frac{1}{n})\frac{\pi i}{2}} + e^{-(1+\frac{1}{n})\frac{\pi i}{2}}), e^{-(1+\frac{1}{n})\frac{\pi i}{2}}, e^{-(1+\frac{1}{n})\frac{\pi i}{2}}). \quad (9)$$

By viewing $\mathbb{C}$ as $\mathbb{R}^2$ and only consider the imaginary part of the above two processes $(\mathbb{E}_{\mathbb{P}}[\mathcal{U}_M(X)|\mathcal{F}_t])_{t=0,1,2}$ in (6), (7) and $(\mathbb{E}_{\mathbb{P}^n}[\mathcal{U}_M(X^n)|\mathcal{F}_t^n])_{t=0,1,2}$ in (8), (9), we may without loss of generality assume that

$$(\mathbb{E}_{\mathbb{P}}[\mathcal{U}_M(X)|\mathcal{F}_t](\boldsymbol{x}_1))_{t=0,1,2} = (0,0,1), (\mathbb{E}_{\mathbb{P}}[\mathcal{U}_M(X)|\mathcal{F}_t](\boldsymbol{x}_2))_{t=0,1,2} = (0,0,-1)$$

and

$$(\mathbb{E}_{\mathbb{P}^n}[\mathcal{U}_M(X^n)|\mathcal{F}_t^n](\boldsymbol{x}_1^n))_{t=0,1,2} = (0, \sin((1+\frac{1}{n})\frac{\pi}{2}), \sin((1+\frac{1}{n})\frac{\pi}{2})),$$

$$(\mathbb{E}_{\mathbb{P}^n}[\mathcal{U}_M(X^n)|\mathcal{F}_t^n](\boldsymbol{x}_2^n))_{t=0,1,2} = (0, -\sin((1+\frac{1}{n})\frac{\pi}{2}), -\sin((1+\frac{1}{n})\frac{\pi}{2})).$$

Now, adding the additional time component to the above real valued paths, and choosing $\mathcal{M} \in \mathcal{L}(\mathbb{R}^2, \mathfrak{u}(2))$ via

$$\mathcal{M}(\begin{bmatrix}1\\0\end{bmatrix}) = \begin{bmatrix}0 & 1\\-1 & 0\end{bmatrix}, \quad \mathcal{M}(\begin{bmatrix}0\\1\end{bmatrix}) = \begin{bmatrix}0 & i\\i & 0\end{bmatrix},$$

we can easily verify that

$$\mathcal{U}_{\mathcal{M}}((\mathbb{E}_{\mathbb{P}}[\mathcal{U}_M(X)|\mathcal{F}_t](\boldsymbol{x}_1))_{t=0,1,2}) = \exp(\mathcal{M}(\begin{bmatrix}1\\0\end{bmatrix})) \exp(\mathcal{M}(\begin{bmatrix}1\\1\end{bmatrix}))$$

$$\neq \exp(\mathcal{M}(\begin{bmatrix}1\\1\end{bmatrix})) \exp(\mathcal{M}(\begin{bmatrix}1\\0\end{bmatrix}))$$

$$= \lim_{n\to\infty} \mathcal{U}_{\mathcal{M}}((\mathbb{E}_{\mathbb{P}^n}[\mathcal{U}_M(X^n)|\mathcal{F}_t^n](\boldsymbol{x}_1^n))_{t=0,1,2}),$$

and

$$\mathcal{U}_{\mathcal{M}}((\mathbb{E}_{\mathbb{P}}[\mathcal{U}_M(X)|\mathcal{F}_t](\boldsymbol{x}_2))_{t=0,1,2}) = \exp(\mathcal{M}(\begin{bmatrix}1\\0\end{bmatrix})) \exp(\mathcal{M}(\begin{bmatrix}1\\-1\end{bmatrix}))$$

$$\neq \exp(\mathcal{M}(\begin{bmatrix}1\\-1\end{bmatrix})) \exp(\mathcal{M}(\begin{bmatrix}1\\0\end{bmatrix}))$$

$$= \lim_{n\to\infty} \mathcal{U}_{\mathcal{M}}((\mathbb{E}_{\mathbb{P}^n}[\mathcal{U}_M(X^n)|\mathcal{F}_t^n](\boldsymbol{x}_2^n))_{t=0,1,2}),$$

where exp denotes the matrix exponential on $\mathbb{C}^{2\times 2}$. From the above calculation, we can further derive that

$$
\begin{aligned}
\lim_{n\to\infty} \mathbb{E}_{\mathbb{P}^n}[\mathcal{U}_{\mathcal{M}}((\mathbb{E}_{\mathbb{P}^n}[\mathcal{U}_M(X^n)|\mathcal{F}_t^n])_{t=0,1,2})] &= \frac{1}{2}\exp(\mathcal{M}(\begin{bmatrix}1\\1\end{bmatrix}))\exp(\mathcal{M}(\begin{bmatrix}1\\0\end{bmatrix})) \\
&+ \frac{1}{2}\exp(\mathcal{M}(\begin{bmatrix}1\\-1\end{bmatrix}))\exp(\mathcal{M}(\begin{bmatrix}1\\0\end{bmatrix})) \\
&\neq \frac{1}{2}\exp(\mathcal{M}(\begin{bmatrix}1\\0\end{bmatrix}))\exp(\mathcal{M}(\begin{bmatrix}1\\1\end{bmatrix})) \\
&+ \frac{1}{2}\exp(\mathcal{M}(\begin{bmatrix}1\\0\end{bmatrix}))\exp(\mathcal{M}(\begin{bmatrix}1\\-1\end{bmatrix})) \\
&= \mathbb{E}_{\mathbb{P}}[\mathcal{U}_{\mathcal{M}}((\mathbb{E}_{\mathbb{P}}[\mathcal{U}_M(X)|\mathcal{F}_t])_{t=0,1,2})],
\end{aligned}
$$

because the matrix multiplication is non-commutative. Therefore,

$$
\lim_{n\to\infty} d_{\mathrm{HS}}(\mathbf{\Phi}_{\mathbb{X}}(M,\mathcal{M}), \mathbf{\Phi}_{\mathbb{X}^n}(M,\mathcal{M})) \neq 0,
$$

which coincides with our observation that $\mathbb{X}^n$ cannot converge to $\mathbb{X}$ for the extended weak convergence.

## A.2 A Brief Introduction to Path Characteristic Functions

In this section we will summarise some crucial properties of the Path Characteristic Functions (PCF) of $\mathbb{R}^d$-valued stochastic processes which were obtained in [19] and [18], and briefly mention its connection with the signature theory.

Recall that for $\boldsymbol{x} \in C^{1\text{-var}}([0,T],\mathbb{R}^d)$ and $M \in \mathcal{L}(\mathbb{R}^d, \mathfrak{u}(m))$, the unitary feature $\mathcal{U}_M(\boldsymbol{x})$ (also called unitary path development) of $\boldsymbol{x}$ under $M$ is defined to be $\boldsymbol{y}_T \in U(m)$ with $\boldsymbol{y}$ being the unique solution to the following linear ODE driven by $M(d\boldsymbol{x}_t)$:

$$
d\boldsymbol{y}_t = \boldsymbol{y}_t M(d\boldsymbol{x}_t), \quad \boldsymbol{y}_0 = I_m.
$$

If $\mathbb{X}$ is an $\mathbb{R}^d$-valued filtered process with sample paths in $C^{1\text{-var}}([0,T],\mathbb{R}^d)$, then its path characteristic function (PCF) is given by the expectation of the unitary feature of $X$:

$$
\mathbf{\Phi}_X(M) = \mathbb{E}_{\mathbb{P}}[\mathcal{U}_M(X)]
$$

where $M \in \mathcal{L}(\mathbb{R}^d, \mathfrak{u}(m))$, $m \in \mathbb{N}$.

It is easy to see that the PCF of stochastic processes is a natural generalisation of the classical characteristic functions for $\mathbb{R}^d$-valued random variables. Indeed, for an $\mathbb{R}^d$-valued random variable $X$, we may view it as a linear path from 0 to 1, i.e., $X_t = tX$ for $t \in [0,1]$. Then, for $m = 1$, as the 1-dimensional unitary Lie algebra $\mathfrak{u}(1)$ is simple the real vector space spanned by the imaginary unit i, we know that every linear mapping $M \in \mathcal{L}(\mathbb{R}^d, \mathfrak{u}(1))$ can be represented by

$$
M(x) = \langle x, \lambda \rangle \mathrm{i}
$$

for some $\lambda \in \mathbb{R}^d$ and $\langle \cdot, \cdot \rangle$ denotes the Eulidean inner product. In this case it holds that the unique solution $\boldsymbol{y}(\omega)$ to the ODE

$$
d\boldsymbol{y}_t(\omega) = \boldsymbol{y}_t(\omega)M(dX_t(\omega)) = \boldsymbol{y}_t(\omega)\langle X(\omega),\lambda\rangle \mathrm{i}dt, \quad \boldsymbol{y}_0(\omega) = 1
$$

is simply $\boldsymbol{y}_t(\omega) = \exp(t\langle X(\omega),\lambda\rangle \mathrm{i})$, which implies that $\mathcal{U}_M(X(\omega) = \boldsymbol{y}_1(\omega) = \exp(\langle X(\omega),\lambda\rangle \mathrm{i})$ and consequently

$$
\mathbf{\Phi}_X(M) = \int \mathcal{U}_M(X(\omega))\mathbb{P}(d\omega) = \mathbb{E}_{\mathbb{P}}[\exp(\langle X,\lambda\rangle \mathrm{i})],
$$

which is exactly the classical characteristic function of $X$ evaluated at $\lambda \in \mathbb{R}^d$.

**Connections of PCF with the Signature Theory**   Given a continuous bounded variation path $\boldsymbol{x} \in C^{1\text{-var}}([0,T],\mathbb{R}^d)$, its signature $S(\boldsymbol{x})$ (see e.g. [6]) is given by a formal series in the (dual of the) tensor algebra $T((\mathbb{R}^d)) = \prod_{n=0}^{\infty}(\mathbb{R}^d)^{\otimes n}$ over $\mathbb{R}^d$:

$$
S(\boldsymbol{x}) = (1, S_1(\boldsymbol{x}), \ldots, S_n(\boldsymbol{x}), \ldots)
$$

where $S_n(\boldsymbol{x}) = \sum_{i_1,\ldots,i_n=1}^{d} \int_{0<t_1<\ldots<t_n<1} dx_{t_1}^{i_1} \ldots dx_{t_n}^{i_n} e_{i_1} \otimes \ldots \otimes e_{i_n} \in (\mathbb{R}^d)^{\otimes n}$, where $e_1, \ldots, e_d$ are the canonical basis of $\mathbb{R}^d$ and $\otimes$ denotes the tensor product. Thanks to the universal property of the tensor algebra $T((\mathbb{R}^d))$, every linear mapping $M \in \mathcal{L}(\mathbb{R}^d, \mathfrak{u}(m))$ can be lifted to an algebra morphism $\tilde{M} : T((\mathbb{R}^d)) \to \mathbb{C}^{m\times m}$ (where $T((\mathbb{R}^d))$ is equipped with the tensor product, and $\mathbb{C}^{m\times m}$ is endowed with the matrix multiplication). It can be shown that (see [18], [19]) the unitary feature $\mathcal{U}_M(\boldsymbol{x})$ is equal to the composition of $\tilde{M}$ and the signature of $\boldsymbol{x}$, i.e., $\mathcal{U}_M(\boldsymbol{x}) = \tilde{M}(S(\boldsymbol{x}))$. Moreover, the classical signature theory (see e.g. [7]) also tells that the signature $S(\boldsymbol{x})$ belongs to the character group of $T((\mathbb{R}^d))$ with respect to a specified Hopf algebra structure. This algebraic property of signature together with the relation that $\mathcal{U}_M(\boldsymbol{x}) = \tilde{M}(S(\boldsymbol{x}))$ does not only guarantees that $\mathcal{U}_M(\boldsymbol{x}) \in U(m)$ takes values in the unitary group, but also the universality of unitary features of paths (see e.g. [19, Theorem A.8]):

**Theorem A.3.**
- *The linear functions on unitary features are stable under multiplication and complex conjugation. More precisely, for any $M_1 \in \mathcal{L}(\mathbb{R}^d, \mathfrak{u}(m_1))$, $M_2 \in \mathcal{L}(\mathbb{R}^d, \mathfrak{u}(m_2))$, $L_1 \in \mathcal{L}(\mathbb{C}^{m_1 \times m_1}, \mathbb{C})$ and $L_2 \in \mathcal{L}(\mathbb{C}^{m_2 \times m_2}, \mathbb{C})$, there exist an $M_3 \in \mathcal{L}(\mathbb{R}^d, \mathfrak{u}(m_3))$, a $L_3 \in \mathcal{L}(\mathbb{C}^{m_3 \times m_3}, \mathbb{C})$, an $M_4 \in \mathcal{L}(\mathbb{R}^d, \mathfrak{u}(m_4))$ and a $L_4 \in \mathcal{L}(\mathbb{C}^{m_4 \times m_4}, \mathbb{C})$ such that*

$$L_1(\mathcal{U}_{M_1}(\boldsymbol{x}))L_2(\mathcal{U}_{M_2}(\boldsymbol{x})) = L_3(\mathcal{U}_{M_3}(\boldsymbol{x})),$$

*and $\overline{L_1(\mathcal{U}_{M_1}(\boldsymbol{x}))} = L_4(\mathcal{U}_{M_4}(\boldsymbol{x}))$.*

- *Let $\mathcal{K} \subset C^{1\text{-}var}([0,T], \mathbb{R}^d)$ be a compact subset. For any continuous and bounded function $f : \mathcal{K} \to \mathbb{C}$ and any $\varepsilon > 0$, there is an $m_* \in \mathbb{N}$ and finitely many $M_1, \ldots, M_N \in \mathcal{L}(\mathbb{R}^d, \mathfrak{u}(m_*))$ as well as linear functionals $L_1, \ldots, L_N \in \mathcal{L}(U(m_*), \mathbb{C})$ such that*

$$\sup_{\boldsymbol{x} \in \mathcal{K}} \left| f(\boldsymbol{x}) - \sum_{i=1}^{N} L_i(\mathcal{U}_{M_i}(\boldsymbol{x})) \right| < \varepsilon.$$

Clearly, the second statement of the above theorem follows immediately from the first statement together with the Stone-Weierstrass theorem for $\mathbb{C}$-valued functions. Since the expectations of continuous bounded functions determines the distributions on $C^{1\text{-}var}([0,T], \mathbb{R}^d)$, as a corollary of the universality theorem above, we obtain the following characteristicness of PCF as mentioned in Theorem 2.6, see also [19, Theorem B.1].

**Other Properties of the unitary features and PCF** Besides the universality and the characteristicness, in [19] and [15] one can find some other nice properties of the unitary featues and PCF, which we will list below without proof:

1. Since $\mathcal{U}_M(\boldsymbol{x})$ takes values in the unitary group $U(m)$ if $M \in \mathcal{L}(\mathbb{R}^d, \mathfrak{u}(m))$, so the Hilbert-Schmidt norm of the PCF $\boldsymbol{\Phi}_M(\mathbb{X})$ of any stochastic process $\mathbb{X}$ (with continuous bounded variation sample paths) is always bounded by $\sqrt{m}$. In particular, $\boldsymbol{\Phi}_M(\mathbb{X})$ can be defined for any stochastic process with no integrability requirement.

2. The unitary feature $\mathcal{U}_M : C^{1\text{-}var}([0,T], \mathbb{R}^d) \to U(m)$ is Lipschitz continuous with respect to the bounded variation norm, see [19, Proposition B.6].

3. If the laws of stochastic processes satisfies enough integrability condition (namely their expected signatures have infinite radius of convergence, see [7] for the definition), then one can use a special subclass of linear mappings $M \in \mathcal{L}(\mathbb{R}^d, \mathfrak{u}(m))$ to determine the laws. More explicitly, for $P_X$ and $P_Y$ two laws of stochastic processes with enough integrability, $P_X = P_Y$ if and only if $\boldsymbol{\Phi}_{P_X}(M) = \boldsymbol{\Phi}_{P_Y}(M)$ for all $M \in \mathcal{L}(\mathbb{R}^d, \mathfrak{o}(m))$ such that $M$ only has possibly nonzero entries in $M_{ij}$ with $|i - j| = 1$, where $M_{ij}$ denotes the entry of $M$ at the $i$-th row and $j$-th column, and $\mathfrak{o}(m)$ is the orthogonal Lie algebra. Thanks to the significant sparsity, such $M \in \mathcal{L}(\mathbb{R}^d, \mathfrak{o}(m))$ is easy to be implemented in the numerical application. See [15] for more details.

4. The PCF induces a metric which can (locally) characterise the weak convergence of the laws of stochastic processes, which is called the PCFD (see [19, Theorem 3.8]). In fact the HRPCFD defined in the present paper can be seen as a counterpart of PCFD in the extended weak convergence.

## A.3 Proof of Theorem 3.3

In this section we prove Theorem 3.3 in a more general setting.

**Definition A.4.** *For* $(M, \mathcal{M}) \in \mathcal{A}_{unitary}$ *with* $M \in \mathcal{L}(\mathbb{R}^d, \mathfrak{u}(n))$, $\mathcal{M} \in \mathcal{L}(\mathbb{C}^{n \times n}, \mathfrak{u}(m))$ *and* $\boldsymbol{p} \in \hat{\mathcal{X}}$
*a measure-valued path, we call*

$$\mathcal{U}_{M,\mathcal{M}}(\boldsymbol{p}) := \mathcal{U}_{\mathcal{M}}(t \mapsto \boldsymbol{p}_t^M), \quad \boldsymbol{p}_t^M = \boldsymbol{\Phi}_{\boldsymbol{p}_t}(M) = \int_{\mathcal{X}} \mathcal{U}_M(\boldsymbol{x}) \boldsymbol{p}_t(d\boldsymbol{x})$$

*the high rank development of* $\boldsymbol{p}$ *under* $(M, \mathcal{M})$.

**Definition A.5.** *For* $\mu \in \mathcal{P}(\hat{\mathcal{X}})$ *a probability measure on the measure-valued path space* $\hat{\mathcal{X}}$, *the*
*function*

$$\boldsymbol{\Phi}_\mu^2 : \mathcal{A}_{unitary} \to \bigcup_{m=1}^{\infty} \mathbb{C}^{m \times m}$$

$$(M, \mathcal{M}) \mapsto \int_{\boldsymbol{p} \in \hat{\mathcal{X}}} \mathcal{U}_{M,\mathcal{M}}(\boldsymbol{p}) \mu(d\boldsymbol{p}) = \int_{\boldsymbol{p} \in \hat{\mathcal{X}}} \mathcal{U}_{\mathcal{M}}(t \mapsto \boldsymbol{p}_t^M) \mu(d\boldsymbol{p})$$

*is called the high rank path characteristic function of* $\mu$ *(Abbreviation: HRPCF).*

The next lemma is straightforward, but will be helpful for us to construct the characteristicity for laws
of measure-valued stochastic processes.

**Lemma A.6.** *Let* $\tilde{M} = (M_j)_{j=1}^{k} \in \bigoplus_{j=1}^{k} \mathcal{L}(\mathbb{R}^d, \mathfrak{u}(j))$ *for some* $k \in \mathbb{N}$, *Then, there exists an*
$M \in \mathcal{L}(\mathbb{R}^d, \mathfrak{u}(n))$ *for* $n = (1 + 2 + \ldots + k)$, *such that for any measure–valued path* $\boldsymbol{p} \in \hat{\mathcal{X}}$ *and for*
*any* $t \in [0, T]$, *one has*

$$\boldsymbol{p}_t^M = \int_{\mathcal{X}} \mathcal{U}_M(\boldsymbol{x}) \boldsymbol{p}_t(d\boldsymbol{x})$$

$$= \begin{bmatrix} \int_{\mathcal{X}} \mathcal{U}_{M_1}(\boldsymbol{x}) \boldsymbol{p}_t(d\boldsymbol{x}) & & & \\ & \int_{\mathcal{X}} \mathcal{U}_{M_2}(\boldsymbol{x}) \boldsymbol{p}_t(d\boldsymbol{x}) & & \\ & & \ddots & \\ & & & \int_{\mathcal{X}} \mathcal{U}_{M_k}(\boldsymbol{x}) \boldsymbol{p}_t(d\boldsymbol{x}) \end{bmatrix}$$

$$\in \mathbb{C}^{n \times n}.$$

*Proof.* Given an $\tilde{M} = (M_j)_{j=1}^{k} \in \bigoplus_{j=1}^{k} \mathcal{L}(\mathbb{R}^d, \mathfrak{u}(j))$, we define $M : \mathbb{R}^d \to \mathfrak{u}(n)$ for $n = 1 + 2 + \ldots + k$ via

$$M(x) = \begin{bmatrix} M_1(x) \in \mathfrak{u}(1) & & & \\ & M_2(x) \in \mathfrak{u}(2) & & \\ & & \ddots & \\ & & & M_k(x) \in \mathfrak{u}(k) \end{bmatrix} \qquad (10)$$

which is obviously a linear mapping due to the linearity of $M_1, \ldots, M_k$.
For any $\mathbb{R}^d$–valued path $\boldsymbol{x} \in \mathcal{X}$, we know that its unitary feature $\mathcal{U}_M(\boldsymbol{x})$ is the unique solution $\boldsymbol{y}$
(evaluated at time $T$) to the linear differential equation

$$d\boldsymbol{y}_t = \boldsymbol{y}_t M(d\boldsymbol{x}_t), \quad \boldsymbol{y}_0 = I_n.$$

On the other hand, let $\boldsymbol{z}_t$ be a curve in $U(n)$ defined by

$$\boldsymbol{z}_t = \begin{bmatrix} \boldsymbol{z}_1(t) \in U(1) & & & \\ & \boldsymbol{z}_2(t) \in U(2) & & \\ & & \ddots & \\ & & & \boldsymbol{z}_k(t) \in U(k) \end{bmatrix},$$

where $\boldsymbol{z}_j(t)$, $j = 1, \ldots, k$ is the unique solution to the linear differential equation

$$d\boldsymbol{y}_t = \boldsymbol{y}_t M_j(d\boldsymbol{x}_t), \quad \boldsymbol{y}_0 = I_j.$$

It is clear that $\boldsymbol{z}$ satisfies that

$$
d\boldsymbol{z}_t =
\begin{bmatrix}
d\boldsymbol{z}_1(t) & & & \\
& d\boldsymbol{z}_2(t) & & \\
& & \ddots & \\
& & & d\boldsymbol{z}_k(t)
\end{bmatrix}
$$

$$
=
\begin{bmatrix}
\boldsymbol{z}_1(t)M_1(d\boldsymbol{x}_t) & & & \\
& \boldsymbol{z}_2(t)M_2(d\boldsymbol{x}_t) & & \\
& & \ddots & \\
& & & \boldsymbol{z}_k(t)M_k(d\boldsymbol{x}_t)
\end{bmatrix}
$$

$$
=
\begin{bmatrix}
\boldsymbol{z}_1(t) & & & \\
& \boldsymbol{z}_2(t) & & \\
& & \ddots & \\
& & & \boldsymbol{z}_k(t)
\end{bmatrix}
\begin{bmatrix}
M_1(d\boldsymbol{x}_t) & & & \\
& M_2(d\boldsymbol{x}_t) & & \\
& & \ddots & \\
& & & M_k(d\boldsymbol{x}_t)
\end{bmatrix}
$$

$$
= \boldsymbol{z}_t M(d\boldsymbol{x}_t).
$$

Hence, by the uniqueness of the solution to the differential equation $d\boldsymbol{y}_t = \boldsymbol{y}_t M(d\boldsymbol{x}_t)$, and invoking that $\boldsymbol{z}_j(T) = \mathcal{U}_{M_j}(\boldsymbol{x})$ for all $j = 1, \ldots, k$, we must have

$$
\mathcal{U}_M(\boldsymbol{x}) = \boldsymbol{z}_T =
\begin{bmatrix}
\mathcal{U}_{M_1}(\boldsymbol{x}) & & & \\
& \mathcal{U}_{M_2}(\boldsymbol{x}) & & \\
& & \ddots & \\
& & & \mathcal{U}_{M_k}(\boldsymbol{x})
\end{bmatrix}.
$$

Now it follows immediately that

$$
\boldsymbol{p}_t^M = \int_{\mathcal{X}} \mathcal{U}_M(\boldsymbol{x})\boldsymbol{p}_t(d\boldsymbol{x})
$$

$$
=
\begin{bmatrix}
\int_{\mathcal{X}} \mathcal{U}_{M_1}(\boldsymbol{x})\boldsymbol{p}_t(d\boldsymbol{x}) & & & \\
& \int_{\mathcal{X}} \mathcal{U}_{M_2}(\boldsymbol{x})\boldsymbol{p}_t(d\boldsymbol{x}) & & \\
& & \ddots & \\
& & & \int_{\mathcal{X}} \mathcal{U}_{M_k}(\boldsymbol{x})\boldsymbol{p}_t(d\boldsymbol{x})
\end{bmatrix}.
$$

$\square$

Theorem 3.3 follows immediately from the next lemma by inserting $\mu = P_{\hat{X}}$ and $\nu = P_{\hat{Y}}$ for prediction processes $\hat{X}$ and $\hat{Y}$ of filtered processes $\mathbb{X}$ and $\mathbb{Y}$, respectively.

**Lemma A.7.** *Let $\mu$ and $\nu$ be two probability measures on measure–valued path space $\hat{\mathcal{X}}$ (that is, $\mu, \nu \in \mathcal{P}(\hat{\mathcal{X}})$). Then $\mu = \nu$ if and only if for every admissible pair of unitary representations $(M, \mathcal{M}) \in \mathcal{A}_{\text{unitary}}$, it holds that*

$$
\boldsymbol{\Phi}_\mu^2(M, \mathcal{M}) = \boldsymbol{\Phi}_\nu^2(M, \mathcal{M}).
$$

*Proof.* Before we start a rigorous proof, let us first give an informal proof to provide some intuition: For each measure-valued path $\boldsymbol{p} = (\boldsymbol{p}_t)_{t \in I} \in \hat{\mathcal{X}}$, we first compute the PCF $\boldsymbol{p}_t^M = \int_{\mathcal{X}} \mathcal{U}_M(\boldsymbol{x})\boldsymbol{p}_t(d\boldsymbol{x}) \in U(m)$ for every $t \in I$, where $M \in \mathcal{L}(\mathbb{R}^d, \mathfrak{u}(m))$. By doing so, the measure-valued path $\boldsymbol{p}$ is transformed to a matrix-valued path $\boldsymbol{p}^M$ in $\mathbb{C}^{m \times m}$. Thanks to the characteristic property of the PCF (see Theorem 2.6), each measure $\boldsymbol{p}_t$ is represented by its PCF $\boldsymbol{p}_t^M$, therefore we may study the matrix-valued path $\boldsymbol{p}^M$ instead of the measure-valued path $\boldsymbol{p}$. Under such identification, the distributions $\mu$ and $\nu$ on the measure-valued path space $\hat{\mathcal{X}}$ can be represented by the push-forward measure $\boldsymbol{p}_\sharp^M \mu$ and $\boldsymbol{p}_\sharp^M \nu$ respectively, which are distributions on the matrix-valued path space. In other words, showing $\mu = \nu$ is equivalent to showing that $\boldsymbol{p}_\sharp^M \mu = \boldsymbol{p}_\sharp^M \nu$. But now using the characteristic property of the PCF again, $\boldsymbol{p}_\sharp^M \mu = \boldsymbol{p}_\sharp^M \nu$ holds if and only if their PCF under linear operator $\mathcal{M} \in \mathcal{L}(\mathbb{C}^{m \times m}, \mathfrak{u}(n))$ coincide with each other, i.e., $\boldsymbol{\Phi}_{\boldsymbol{p}_\sharp^M \mu}(\mathcal{M}) = \boldsymbol{\Phi}_{\boldsymbol{p}_\sharp^M \nu}(\mathcal{M})$, and by

definition one has $\boldsymbol{\Phi}_{\boldsymbol{p}_\sharp^M \mu}(\mathcal{M}) = \boldsymbol{\Phi}_\mu^2(M, \mathcal{M})$, $\boldsymbol{\Phi}_{\boldsymbol{p}_\sharp^M \nu}(\mathcal{M}) = \boldsymbol{\Phi}_\nu^2(M, \mathcal{M})$.

Now we provide the rigorous proof of the theorem. Obviously we only need to show the "if" part.

*Step 1:* By hypothesis, for any admissible pair of unitary representations $(M, \mathcal{M}) \in \mathcal{A}_{\text{unitary}}$ with $M \in \mathcal{L}(\mathbb{R}^d, \mathfrak{u}(n))$ and $\mathcal{M} \in \mathcal{L}(\mathbb{C}^{n\times n}, \mathfrak{u}(m))$ we have

$$\boldsymbol{\Phi}_\mu^2(M, \mathcal{M}) = \int_{\boldsymbol{p} \in \hat{\mathcal{X}}} \mathcal{U}_\mathcal{M}(t \mapsto \boldsymbol{p}_t^M)\mu(d\boldsymbol{p}) = \int_{\boldsymbol{p} \in \hat{\mathcal{X}}} \mathcal{U}_\mathcal{M}(t \mapsto \boldsymbol{p}_t^M)\nu(d\boldsymbol{p}) = \boldsymbol{\Phi}_\nu^2(M, \mathcal{M}),$$

which means that $\boldsymbol{\Phi}_{(\boldsymbol{p} \mapsto \boldsymbol{p}^M)_\sharp(\mu)}(\mathcal{M}) = \boldsymbol{\Phi}_{(\boldsymbol{p} \mapsto \boldsymbol{p}^M)_\sharp(\nu)}(\mathcal{M})$, where the push-forward measures $(\boldsymbol{p} \mapsto \boldsymbol{p}^M)_\sharp(\mu)$ and $(\boldsymbol{p} \mapsto \boldsymbol{p}^M)_\sharp(\nu)$ are probability measures on the $\mathbb{C}^{n\times n}$–valued path space.

In fact, if we fix an arbitrary $n \in \mathbb{N}$ and an arbitrary $M \in \mathcal{L}(\mathbb{R}^d, \mathfrak{u}(n))$, and let $\mathcal{M} \in \mathcal{L}(\mathbb{C}^{n\times n}, \mathfrak{u}(m))$ vary over all $m \in \mathbb{N}$, we actually have the above equality $\boldsymbol{\Phi}_{(\boldsymbol{p} \mapsto \boldsymbol{p}^M)_\sharp(\mu)}(\mathcal{M}) = \boldsymbol{\Phi}_{(\boldsymbol{p} \mapsto \boldsymbol{p}^M)_\sharp(\nu)}(\mathcal{M})$ for all $\mathcal{M} \in \mathcal{L}(\mathbb{R}^n, \mathfrak{u}(m))$, $m \in \mathbb{N}$. Therefore, by applying the characteristicity of PCF of measures on finite dimensional vector space valued path spaces, see Theorem 2.6, we obtain that $(\boldsymbol{p} \mapsto \boldsymbol{p}^M)_\sharp(\mu) = (\boldsymbol{p} \mapsto \boldsymbol{p}^M)_\sharp(\nu)$ for any $n \in \mathbb{N}$ and any $M \in \mathcal{L}(\mathbb{R}^d, \mathfrak{u}(n))$.

*Step 2:* Fix an $k \in \mathbb{N}$ and a sequence of operators $\tilde{M} = (M_j)_{j=1}^k \in \bigoplus_{j=1}^k \mathcal{L}(\mathbb{R}^d, \mathfrak{u}(j))$. Let $n = 1 + 2 + \ldots + k$. By Lemma A.6 above, there exists an $M \in \mathcal{L}(\mathbb{R}^d, \mathfrak{u}(n))$ such that for any $\boldsymbol{p} \in \hat{\mathcal{X}}$, one has

$$\boldsymbol{p}_t^M = \int_\mathcal{X} \mathcal{U}_M(\boldsymbol{x})\boldsymbol{p}_t(d\boldsymbol{x})$$
$$= \begin{bmatrix} \int_\mathcal{X} \mathcal{U}_{M_1}(\boldsymbol{x})\boldsymbol{p}_t(d\boldsymbol{x}) & & & \\ & \int_\mathcal{X} \mathcal{U}_{M_2}(\boldsymbol{x})\boldsymbol{p}_t(d\boldsymbol{x}) & & \\ & & \ddots & \\ & & & \int_\mathcal{X} \mathcal{U}_{M_k}(\boldsymbol{x})\boldsymbol{p}_t(d\boldsymbol{x}) \end{bmatrix}.$$

Now we take an arbitrary partition $\{t_1 < t_2 < \ldots < t_N\}$ of the time interval $[0, T]$. Since we have shown in Step 1 that $(\boldsymbol{p} \mapsto \boldsymbol{p}^M)_\sharp(\mu) = (\boldsymbol{p} \mapsto \boldsymbol{p}^M)_\sharp(\nu)$, that is, the law of $\mathbb{C}^{n\times n}$–valued stochastic process $\boldsymbol{p}_t^M = \int_\mathcal{X} \mathcal{U}_M(\boldsymbol{x})\boldsymbol{p}_t(d\boldsymbol{x})$, $t \in [0, T]$ under $\mu \in \mathcal{P}(\hat{\mathcal{X}})$ coincides with its law under $\nu \in \mathcal{P}(\hat{\mathcal{X}})$, we indeed have that the distributions of their marginals at $t_1, \ldots, t_N$ are same, that is,

$$(\boldsymbol{p} \mapsto (\boldsymbol{p}_{t_1}^M, \ldots, \boldsymbol{p}_{t_N}^M))_\sharp \mu = (\boldsymbol{p} \mapsto (\boldsymbol{p}_{t_1}^M, \ldots, \boldsymbol{p}_{t_N}^M))_\sharp \nu \in \mathcal{P}((\mathbb{C}^{n\times n})^N).$$

Now, for each $i = 1, \ldots, N$ and $j = 1, \ldots, k$, we pick arbitrary linear functions $\boldsymbol{L}_j(i) \in \mathcal{L}(\mathbb{C}^{j\times j}, \mathbb{R})$ and continuous and bounded functions $g_i \in C_b(\mathbb{R})$, and use them to define a function $\tilde{g}_i : \mathbb{C}^{n\times n} \to \mathbb{R}$ for $i = 1, \ldots, N$ such that for any matrix $A \in \mathbb{C}^{n\times n}$ (recall that $n = 1 + 2 + \ldots + k$) written in the form

$$A = \begin{bmatrix} A_1 \in \mathbb{C}^{1\times 1} & \star & \star & \star \\ \star & A_2 \in \mathbb{C}^{2\times 2} & \star & \star \\ \star & \star & \ddots & \star \\ \star & \star & \star & A_k \in \mathbb{C}^{k\times k} \end{bmatrix},$$

it holds that

$$\tilde{g}_i(A) = g_i \circ \left( \sum_{j=1}^k \boldsymbol{L}_j(i) \circ A_i \right).$$

Obviously each function $\tilde{g}_i$ is continuous and bounded.

Let $\tilde{g} : (\mathbb{C}^{n\times n})^N \to \mathbb{R}$ be the continuous and bounded function such that $\tilde{g}(A^1, \ldots, A^N) = \prod_{i=1}^N \tilde{g}_i(A^i)$ for every sequence $\bar{A} = (A^1, \ldots, A^N) \in (\mathbb{C}^{n\times n})^N$. From the equality $(\boldsymbol{p} \mapsto (\boldsymbol{p}_{t_1}^M, \ldots, \boldsymbol{p}_{t_N}^M))_\sharp \mu = (\boldsymbol{p} \mapsto (\boldsymbol{p}_{t_1}^M, \ldots, \boldsymbol{p}_{t_N}^M))_\sharp \nu \in \mathcal{P}((\mathbb{C}^{n\times n})^N)$ it follows that

$$\int \tilde{g}(\bar{A})(\boldsymbol{p} \mapsto (\boldsymbol{p}_{t_1}^M, \ldots, \boldsymbol{p}_{t_N}^M))_\sharp \mu(d\bar{A}) = \int \tilde{g}(\bar{A})(\boldsymbol{p} \mapsto (\boldsymbol{p}_{t_1}^M, \ldots, \boldsymbol{p}_{t_N}^M))_\sharp \nu(d\bar{A}),$$

which can be reformulated as

$$\int_{\hat{\mathcal{X}}} \prod_{i=1}^{N} g_i \left( \sum_{j=1}^{k} \mathbb{E}_{\boldsymbol{p}_{t_i}}[\boldsymbol{L}_j(i) \circ \mathcal{U}_{M_j}] \right) \mu(d\boldsymbol{p}) = \tag{11}$$

$$\int_{\hat{\mathcal{X}}} \prod_{i=1}^{N} g_i \left( \sum_{j=1}^{k} \mathbb{E}_{\boldsymbol{p}_{t_i}}[\boldsymbol{L}_j(i) \circ \mathcal{U}_{M_j}] \right) \nu(d\boldsymbol{p})$$

where $\mathbb{E}_{\boldsymbol{p}_t}[\boldsymbol{L}_j(i) \circ \mathcal{U}_{M_j}] = \int_{\boldsymbol{x} \in \mathcal{X}} \boldsymbol{L}_j(i) \circ \mathcal{U}_{M_j}(\boldsymbol{x})\boldsymbol{p}_t(d\boldsymbol{x})$.

*Step 3:* It is a well known fact (see e.g. [7]) that the vector space generated by all real-valued linear functionals of unitary representations on the path space $\mathcal{X}$, namely

$$\mathcal{C} = \text{span}\{L \circ \mathcal{U}_M : \mathcal{X} \to \mathbb{R} : L \in \mathcal{L}(\mathbb{C}^{j \times j}, \mathbb{R}), M \in \mathcal{L}(\mathbb{R}^d, \mathfrak{u}(j)), j \in \mathbb{N}\},$$

is a sub-algebra in the space $C_b(\mathcal{X})$ of continuous and bounded (real-valued) functions on $\mathcal{X}$ which separates the points. Moreover, by picking $M_0 : \mathbb{R}^d \to \mathfrak{u}(1)$ to be the trivial representation (i.e., $M_0(x) = 0 \in \mathbb{C}$ for all $x \in \mathbb{R}^d$) we see that for any path $\boldsymbol{x} \in \mathcal{X}$, $M_0(\boldsymbol{x}) = 1 \in \mathbb{R}$. Therefore, by the Giles' Theorem ([8, Theorem 9]) it follows that the set $\mathcal{C}$ is dense in $C_b(\mathcal{X})$ related to the so called strict topology[8].
Now, fix arbitrary continuous and bounded functions $f_i \in C_b(\mathcal{X})$, $i = 1, \dots, N$. From the density of $\mathcal{C}$ in $C_b(\mathcal{X})$ one can find a sequence of unitary representations $\tilde{M}^{(k)} = (M_j^{(k)})_{j=1}^{k} \in \bigoplus_{j=1}^{k} \mathcal{L}(\mathbb{R}^d, \mathfrak{u}(j))$, $k \in \mathbb{N}$ together with a sequence of linear operators $(\boldsymbol{L}^{(k)}(i))_{k \in \mathbb{N}}$, $i = 1, \dots, N$ with each $\boldsymbol{L}^k(i) = (\boldsymbol{L}_j^{(k)}(i))_{j=1}^{k} \in \bigoplus_{j=1}^{k} \mathcal{L}(\mathbb{C}^{j \times j}, \mathbb{R})$ such that for every $i = 1, \dots, N$ it holds that

$$f_i = \lim_{k \to \infty} \sum_{j=1}^{k} \boldsymbol{L}_j^{(k)}(i) \circ \mathcal{U}_{M_j^{(k)}}, \tag{12}$$

where the convergence happens in the strict topology. Furthermore, since every probability measure $\boldsymbol{p}_{t_i} \in \mathcal{P}(\mathcal{X})$ $(i = 1, \dots, N)$ belongs to the topological dual of $C_b(\mathcal{X})$ equipped with the strict topology by the Giles' theorem, invoking the relation (12) we actually obtain that for every $i = 1, \dots, N$,

$$\mathbb{E}_{\boldsymbol{p}_{t_i}}[f_i] = \int_{\mathcal{X}} f_i(\boldsymbol{x})\boldsymbol{p}_{t_i}(d\boldsymbol{x}) = \lim_{k \to \infty} \sum_{j=1}^{k} \mathbb{E}_{\boldsymbol{p}_{t_i}}[\boldsymbol{L}_j^{(k)}(i) \circ \mathcal{U}_{M_j^{(k)}}].$$

Then, as a consequence of the result (11) obtained in Step 2, we can apply the bounded convergence theorem to get that

$$\int_{\hat{\mathcal{X}}} \prod_{i=1}^{N} g_i(\mathbb{E}_{\boldsymbol{p}_{t_i}}[f_i])\mu(d\boldsymbol{p}) = \lim_{k \to \infty} \int_{\hat{\mathcal{X}}} \prod_{i=1}^{N} g_i \left( \sum_{j=1}^{k} \mathbb{E}_{\boldsymbol{p}_{t_i}}[\boldsymbol{L}_j^{(k)}(i) \circ \mathcal{U}_{M_j^{(k)}}] \right) \mu(d\boldsymbol{p})$$

$$= \lim_{k \to \infty} \int_{\hat{\mathcal{X}}} \prod_{i=1}^{N} g_i \left( \sum_{j=1}^{k} \mathbb{E}_{\boldsymbol{p}_{t_i}}[\boldsymbol{L}_j^{(k)}(i) \circ \mathcal{U}_{M_j^{(k)}}] \right) \nu(d\boldsymbol{p})$$

$$= \int_{\hat{\mathcal{X}}} \prod_{i=1}^{N} g_i(\mathbb{E}_{\boldsymbol{p}_{t_i}}[f_i])\nu(d\boldsymbol{p}). \tag{13}$$

On the other hand, by the Urysohn's lemma, for any $i = 1, \dots, N$, any positive number $R_i > 0$, the indicator function $1_{[-R_i, R_i]}$ can be pointwise approximated by a sequence of $[0, 1]$–valued continuous functions $(g_i^\ell)_{\ell \in \mathbb{N}}$. Hence, by replacing the functions $g_i$ by $g_i^\ell$ in (13) and then letting $\ell \to \infty$, using the bounded convergence theorem we can derive that

$$\int_{\hat{\mathcal{X}}} \prod_{i=1}^{N} 1_{[-R_i, R_i]}(\mathbb{E}_{\boldsymbol{p}_{t_i}}[f_i])\mu(d\boldsymbol{p}) = \int_{\hat{\mathcal{X}}} \prod_{i=1}^{N} 1_{[-R_i, R_i]}(\mathbb{E}_{\boldsymbol{p}_{t_i}}[f_i])\nu(d\boldsymbol{p})$$

---

[8]For the definition of the strict topology, see e.g. [8, Definition 8].

or, equivalently,

$$\int_{\hat{\mathcal{X}}} \prod_{i=1}^{N} 1_{\theta_{f_i}^{-1}([-R_i,R_i])}(\boldsymbol{p}_{t_i})\mu(d\boldsymbol{p}) = \int_{\hat{\mathcal{X}}} \prod_{i=1}^{N} 1_{\theta_{f_i}^{-1}([-R_i,R_i])}(\boldsymbol{p}_{t_i})\nu(d\boldsymbol{p}) \tag{14}$$

where $\theta_{f_i}(\boldsymbol{p}_{t_i}) := \mathbb{E}_{\boldsymbol{p}_{t_i}}[f_i]$ denotes the evaluation map of $f_i \in C_b(\mathcal{X})$ against the measure $\boldsymbol{p}_{t_i}$.

*Step 4:* By the very definition of weak topology on $\mathcal{P}(\mathcal{X})$, its Borel $\sigma$–algebra is generated by the sets of the form that $\theta_f^{-1}([-R,R])$ for $f \in C_b(\mathcal{X})$ and $R > 0$. Consequently, the Borel $\sigma$–algebra on the product space $\mathcal{P}(\mathcal{X})^N$ is generated by the measurable rectangles of the form that $\prod_{i=1}^{N} \theta_{f_i}^{-1}([-R_i,R_i])$ for $f_i \in C_b(\mathcal{X})$ and $R_i > 0$. From Eq. (14) we know that

$$\mu((\boldsymbol{p}_{t_1},\ldots,\boldsymbol{p}_{t_N}) \in \prod_{i=1}^{N} \theta_{f_i}^{-1}([-R_i,R_i])) = \nu((\boldsymbol{p}_{t_1},\ldots,\boldsymbol{p}_{t_N}) \in \prod_{i=1}^{N} \theta_{f_i}^{-1}([-R_i,R_i]))$$

for all such measurable rectangles. Since the above equation holds for any partition $\{t_1 < \ldots < t_N\}$ of $[0,T]$ and the laws of (continuous) stochastic processes are uniquely determined by their marginals on finitely many time points, we can conclude that $\mu = \nu$ in $\mathcal{P}(\hat{\mathcal{X}})$ by a routine application of the monotone class theorem. $\qquad\square$

Now, for filtered processes $\mathbb{X}$ and $\mathbb{Y}$, we note that the associated prediction processes $\hat{X}$ and $\hat{Y}$ are stochastic processes taking values in $\mathcal{P}(\mathcal{X})$ which can be viewed as $\hat{\mathcal{X}}$-valued random variable, which in turn implies that their laws $P_{\hat{X}}$ and $P_{\hat{Y}}$ are elements in $\mathcal{P}(\hat{\mathcal{X}})$. Hence, inserting $\mu = P_{\hat{X}}$ and $\nu = P_{\hat{Y}}$ into the above Lemma A.7 we can easily deduce Theorem 3.3.

### A.4 Properties of HRPCFD

In this section we will mainly prove the properties recorded in section 3.3.

First let us prove the property of HRPCFD on the separation of laws of prediction processes. To achieve this we need the following useful continuity lemma.

**Lemma A.8.** *For any fixed $\mu \in \mathcal{P}(\hat{\mathcal{X}})$, any fixed $n$ and $m$, the mapping*

$$(M,\mathcal{M}) \in \mathcal{L}(\mathbb{R}^d, \mathfrak{u}(n)) \times \mathcal{L}(\mathbb{C}^{n \times n}, \mathfrak{u}(m)) \mapsto \boldsymbol{\Phi}_\mu^2(M,\mathcal{M}) \in \mathbb{C}^{m \times m}$$

*is continuous for the operator norm topology on $\mathcal{L}(\mathbb{R}^d, \mathfrak{u}(n)) \times \mathcal{L}(\mathbb{C}^{n \times n}, \mathfrak{u}(m))$ and the Hilbert–Schmidt norm topology on $\mathbb{C}^{m \times m}$.*

*Proof.* For admissible pairs $(M,\mathcal{M})$ and $(M',\mathcal{M}')$ from $\mathcal{L}(\mathbb{R}^d, \mathfrak{u}(n)) \times \mathcal{L}(\mathbb{C}^{n \times n}, \mathfrak{u}(m))$, by the definition of HRPCF we have

$$\|\boldsymbol{\Phi}_\mu^2(M,\mathcal{M}) - \boldsymbol{\Phi}_\mu^2(M',\mathcal{M}')\|_{\mathrm{HS}} = \left\| \int_{\boldsymbol{p}\in\hat{\mathcal{X}}} \mathcal{U}_\mathcal{M}(t \mapsto \boldsymbol{p}_t^M)\mu(d\boldsymbol{p}) - \int_{\boldsymbol{p}\in\hat{\mathcal{X}}} \mathcal{U}_{\mathcal{M}'}(t \mapsto \boldsymbol{p}_t^{M'})\mu(d\boldsymbol{p}) \right\|_{\mathrm{HS}}$$

$$\leq \int_{\boldsymbol{p}\in\hat{\mathcal{X}}} \left\| \mathcal{U}_\mathcal{M}(t \mapsto \boldsymbol{p}_t^M) - \mathcal{U}_{\mathcal{M}'}(t \mapsto \boldsymbol{p}_t^{M'}) \right\|_{\mathrm{HS}} \mu(d\boldsymbol{p})$$

$$\leq \int_{\boldsymbol{p}\in\hat{\mathcal{X}}} \left\| \mathcal{U}_\mathcal{M}(t \mapsto \boldsymbol{p}_t^M) - \mathcal{U}_\mathcal{M}(t \mapsto \boldsymbol{p}_t^{M'}) \right\|_{\mathrm{HS}} \mu(d\boldsymbol{p})$$

$$+ \int_{\boldsymbol{p}\in\hat{\mathcal{X}}} \left\| \mathcal{U}_\mathcal{M}(t \mapsto \boldsymbol{p}_t^{M'}) - \mathcal{U}_{\mathcal{M}'}(t \mapsto \boldsymbol{p}_t^{M'}) \right\|_{\mathrm{HS}} \mu(d\boldsymbol{p}). \tag{15}$$

Let us first estimate the first integrand on the right hand side of (15). By [19, Proposition B.6] we know that for each measure–valued path $\boldsymbol{p} \in \hat{\mathcal{X}}$, one has

$$\|\mathcal{U}_\mathcal{M}(t \mapsto \boldsymbol{p}_t^M) - \mathcal{U}_\mathcal{M}(t \mapsto \boldsymbol{p}_t^{M'})\|_{\mathrm{HS}} \leq \|\mathcal{M}\|_{\mathrm{op}}\|\boldsymbol{p}^M - \boldsymbol{p}^{M'}\|_{\text{1-var}},$$

where $\boldsymbol{p}_t^M = \Phi_{\boldsymbol{p}_t}(M) = \int_{\mathcal{X}} \mathcal{U}_M(\boldsymbol{x})\boldsymbol{p}_t(d\boldsymbol{x})$ and $\boldsymbol{p}_t^{M'} = \Phi_{\boldsymbol{p}_t}(M') = \int_{\mathcal{X}} \mathcal{U}_{M'}(\boldsymbol{x})\boldsymbol{p}_t(d\boldsymbol{x})$ are $\mathbb{C}^{n \times n}$–valued paths. Since $\boldsymbol{p} \in \hat{\mathcal{X}}$ is piecewise linear, these $\mathbb{C}^{n \times n}$–valued paths $\boldsymbol{p}^M = (t \mapsto \Phi_{\boldsymbol{p}_t}(M))$ and

$\boldsymbol{p}^{M'} = (t \mapsto \Phi_{\boldsymbol{p}_t}(M'))$ are also piecewise linear, say, they are linear on time subintervals $[t_i, t_{i+1}]$ for $i = 0, \ldots, N-1$. Then we indeed have

$$\|\boldsymbol{p}^M - \boldsymbol{p}^{M'}\|_{\text{1-var}} = \sum_{i=0}^{N-1} \|(\boldsymbol{p}^M - \boldsymbol{p}^{M'})_{t_i, t_{i+1}}\|_{\text{HS}} \leq 2\sum_{i=0}^{N} \|\boldsymbol{p}_{t_i}^M - \boldsymbol{p}_{t_i}^{M'}\|_{\text{HS}},$$

whence the estimates

$$\|\mathcal{U}_{\mathcal{M}}(t \mapsto \boldsymbol{p}_t^M) - \mathcal{U}_{\mathcal{M}}(t \mapsto \boldsymbol{p}_t^{M'})\|_{\text{HS}} \lesssim \|\mathcal{M}\|_{\text{op}} \sum_{i=0}^{N} \|\boldsymbol{p}_{t_i}^M - \boldsymbol{p}_{t_i}^{M'}\|_{\text{HS}}. \tag{16}$$

Now we note that for each $i = 0, \ldots, N$, we have

$$\boldsymbol{p}_{t_i}^M - \boldsymbol{p}_{t_i}^{M'} = \int_{\boldsymbol{x} \in \mathcal{X}} \mathcal{U}_M(\boldsymbol{x})\boldsymbol{p}_{t_i}(d\boldsymbol{x}) - \int_{\boldsymbol{x} \in \mathcal{X}} \mathcal{U}_{M'}(\boldsymbol{x})\boldsymbol{p}_{t_i}(d\boldsymbol{x}).$$

Recalling that for each $\mathbb{R}^d$–valued path $\boldsymbol{x} \in \mathcal{X}$, one has $\mathcal{U}_M(\boldsymbol{x}) = \boldsymbol{y}_T^{M,\boldsymbol{x}}$ and $\mathcal{U}_{M'}(\boldsymbol{x}) = \boldsymbol{y}_T^{M',\boldsymbol{x}}$, where $\boldsymbol{y}^{M,\boldsymbol{x}}$ and $\boldsymbol{y}^{M',\boldsymbol{x}}$ are the unique solutions to the linear ODEs

$$d\boldsymbol{y}_t^{M,\boldsymbol{x}} = \boldsymbol{y}_t^{M,\boldsymbol{x}} M(d\boldsymbol{x}_t), \quad \boldsymbol{y}_0^{M,\boldsymbol{x}} = I_n$$

and

$$d\boldsymbol{y}_t^{M',\boldsymbol{x}} = \boldsymbol{y}_t^{M',\boldsymbol{x}} M'(d\boldsymbol{x}_t), \quad \boldsymbol{y}_0^{M',\boldsymbol{x}} = I_n$$

respectively, by the continuity of the flow of ODE (see e.g. [11, Theorem 3.15]), we obtain that for any $\boldsymbol{x} \in \mathcal{X}$,

$$\|\mathcal{U}_M(\boldsymbol{x}) - \mathcal{U}_{M'}(\boldsymbol{x})\|_{\text{HS}} \leq C(n, \|\boldsymbol{x}\|_{\text{1-var}})\|M - M'\|_{\text{op}}.$$

In particular, if $\|M' - M\|_{\text{op}} \to 0$, then for all $\boldsymbol{x} \in \mathcal{X}$ we have $\|\mathcal{U}_M(\boldsymbol{x}) - \mathcal{U}_{M'}(\boldsymbol{x})\|_{\text{HS}} \to 0$. Then because $\mathcal{U}_M$ and $\mathcal{U}_{M'}$ are unitary representations taking values in the compact group $U(n)$, by the dominated convergence theorem we have $\|\boldsymbol{p}_{t_i}^M - \boldsymbol{p}_{t_i}^{M'}\|_{\text{HS}} \to 0$ as $\|M' - M\|_{\text{op}} \to 0$ for any $i = 0, \ldots, N$. As a result, in view of (16) we obtain that

$$\|M' - M\|_{\text{op}} \to 0 \Rightarrow \|\mathcal{U}_{\mathcal{M}}(t \mapsto \boldsymbol{p}_t^M) - \mathcal{U}_{\mathcal{M}}(t \mapsto \boldsymbol{p}_t^{M'})\|_{\text{HS}} \to 0$$

for all $\boldsymbol{p} \in \hat{\mathcal{X}}$. Then, as $\mathcal{U}_{\mathcal{M}}$ is a unitary representation taking values in the compact group $U(m)$, by the dominated convergence theorem again we have

$$\|M' - M\|_{\text{op}} \to 0 \Rightarrow \int_{\boldsymbol{p} \in \hat{\mathcal{X}}} \left\|\mathcal{U}_{\mathcal{M}}(t \mapsto \boldsymbol{p}_t^M) - \mathcal{U}_{\mathcal{M}}(t \mapsto \boldsymbol{p}_t^{M'})\right\|_{\text{HS}} \mu(d\boldsymbol{p}) \to 0. \tag{17}$$

Next we turn to bound the second integrand in (15), namely $\|\mathcal{U}_{\mathcal{M}}(t \mapsto \boldsymbol{p}_t^{M'}) - \mathcal{U}_{\mathcal{M}'}(t \mapsto \boldsymbol{p}_t^{M'})\|_{\text{HS}}$. Again, invoking that $\mathcal{U}_{\mathcal{M}}(t \mapsto \boldsymbol{p}_t^{M'}) = \boldsymbol{y}_T^{\mathcal{M},\boldsymbol{p}^{M'}}$ and $\mathcal{U}_{\mathcal{M}'}(t \mapsto \boldsymbol{p}_t^{M'}) = \boldsymbol{y}_T^{\mathcal{M}',\boldsymbol{p}^{M'}}$ are the unique solutions (evaluated at $T$) to the linear ODEs

$$d\boldsymbol{y}_t^{\mathcal{M},\boldsymbol{p}^{M'}} = \boldsymbol{y}_t^{\mathcal{M},\boldsymbol{p}^{M'}} \mathcal{M}(d\boldsymbol{p}_t^{M'}), \quad \boldsymbol{y}_0^{\mathcal{M},\boldsymbol{p}^{M'}} = I_m$$

and

$$d\boldsymbol{y}_t^{\mathcal{M}',\boldsymbol{p}^{M'}} = \boldsymbol{y}_t^{\mathcal{M}',\boldsymbol{p}^{M'}} \mathcal{M}'(d\boldsymbol{p}_t^{M'}), \quad \boldsymbol{y}_0^{\mathcal{M}',\boldsymbol{p}^{M'}} = I_m$$

respectively, by the continuity of the flow of ODE, we obtain that for each $\boldsymbol{p} \in \hat{\mathcal{X}}$,

$$\|\mathcal{U}_{\mathcal{M}}(t \mapsto \boldsymbol{p}_t^{M'}) - \mathcal{U}_{\mathcal{M}'}(t \mapsto \boldsymbol{p}_t^{M'})\|_{\text{HS}} \leq C(m, \|\boldsymbol{p}^{M'}\|_{\text{1-var}})\|\mathcal{M} - \mathcal{M}'\|_{\text{op}}.$$

Since $\mathcal{U}_{M'}$ takes values in the compact group $U(n)$, it is easy to see that for piecewise linear path $\boldsymbol{p}_t^{M'} = \int \mathcal{U}_{M'}(\boldsymbol{x})\boldsymbol{p}_t(d\boldsymbol{x})$ it holds that $\sup_{M' \in \mathcal{L}(\mathbb{R}^d, \mathfrak{u}(n))} \|\boldsymbol{p}^{M'}\|_{\text{1-var}} < \infty$, which implies that for any $\boldsymbol{p} \in \hat{\mathcal{X}}$ and for any $M' \in \mathcal{L}(\mathbb{R}^d, \mathfrak{u}(n))$,

$$\|\mathcal{M}' - \mathcal{M}\|_{\text{op}} \to 0 \Rightarrow \|\mathcal{U}_{\mathcal{M}}(t \mapsto \boldsymbol{p}_t^{M'}) - \mathcal{U}_{\mathcal{M}'}(t \mapsto \boldsymbol{p}_t^{M'})\|_{\text{HS}} \to 0.$$

Again, since $\mathcal{U}_{\mathcal{M}}$ and $\mathcal{U}_{\mathcal{M}'}$ are unitary features with values in compact group $U(m)$, by the dominated convergence theorem we must have

$$\int_{\boldsymbol{p} \in \hat{\mathcal{X}}} \left\|\mathcal{U}_{\mathcal{M}}(t \mapsto \boldsymbol{p}_t^{M'}) - \mathcal{U}_{\mathcal{M}'}(t \mapsto \boldsymbol{p}_t^{M'})\right\|_{\text{HS}} \mu(d\boldsymbol{p}) \to 0 \tag{18}$$

as long as $\|\mathcal{M}' - \mathcal{M}\|_{\text{op}} \to 0$. Now, combining (18), (17) and (15) we can conclude that
$$\|\boldsymbol{\Phi}_\mu^2(M, \mathcal{M}) - \boldsymbol{\Phi}_\mu^2(M', \mathcal{M}')\|_{\text{HS}} \to 0$$
as long as $\|M' - M\|_{\text{op}} \to 0, \|\mathcal{M}' - \mathcal{M}\|_{\text{op}} \to 0$, which is the desired continuity claim. $\qquad\square$

Now we are able to prove the first property of HRPCFD.

**Theorem A.9** (Separation of points). *Let $\mu, \nu \in \mathcal{P}(\hat{\mathcal{X}})$ be two distributions on measure–valued path space such that $\mu \neq \nu$. Then there exists a pair of integers $(n, m) \in \mathbb{N}^2$ such that for any $P_M \in \mathcal{P}(\mathcal{L}(\mathbb{R}^d, \mathfrak{u}(n)))$ with full support and any $P_{\mathcal{M}} \in \mathcal{P}(\mathcal{L}(\mathbb{C}^{n \times n}, \mathfrak{u}(m)))$ with full support, one has*
$$HRPCFD_{M, \mathcal{M}}(\mu, \nu) > 0.$$
*In particular, for filtered processes $\mathbb{X}$ and $\mathbb{Y}$, if they are not synonymous, then with $\mu = P_{\hat{X}}$ and $\nu = P_{\hat{Y}}$ there exists a pair of integers $(n, m) \in \mathbb{N}^2$ such that for any $P_M \in \mathcal{P}(\mathcal{L}(\mathbb{R}^d, \mathfrak{u}(n)))$ with full support and any $P_{\mathcal{M}} \in \mathcal{P}(\mathcal{L}(\mathbb{C}^{n \times n}, \mathfrak{u}(m)))$ with full support, one has*
$$HRPCFD_{M, \mathcal{M}}(\mathbb{X}, \mathbb{Y}) > 0.$$

*Proof.* Thanks to Lemma A.7, if $\mu \neq \nu$, then there must exist an admissible pair of unitary representations $(M_0, \mathcal{M}_0) \in \mathcal{A}_{\text{unitary}}$ with $M_0 \in \mathcal{L}(\mathbb{R}^d, \mathfrak{u}(n))$ and $\mathcal{M}_0 \in \mathcal{L}(\mathbb{C}^{n \times n}, \mathfrak{u}(m))$ such that
$$\boldsymbol{\Phi}_\mu^2(M_0, \mathcal{M}_0) \neq \boldsymbol{\Phi}_\nu^2(M_0, \mathcal{M}_0).$$
Then, by the continuity result proved in Lemma A.8, there exists a $\delta > 0$ such that for all $M \in \mathcal{L}(\mathbb{R}^d, \mathfrak{u}(n))$ and all $\mathcal{M} \in \mathcal{L}(\mathbb{C}^{n \times n}, \mathfrak{u}(m))$ with $\|M_0 - M\|_{\text{op}} \leq \delta$ and $\|\mathcal{M}_0 - \mathcal{M}\|_{\text{op}} \leq \delta$, it holds that
$$d_{\text{HS}}(\boldsymbol{\Phi}_\mu^2(M, \mathcal{M}), \boldsymbol{\Phi}_\nu^2(M, \mathcal{M})) > 0.$$
Let $B(M_0, \delta) \subset \mathcal{L}(\mathbb{R}^d, \mathfrak{u}(n))$ and $B(\mathcal{M}_0, \delta) \subset \mathcal{L}(\mathbb{C}^{n \times n}, \mathfrak{u}(m))$ denote the ball centered at $M_0$ and $\mathcal{M}_0$ with radius $\delta$ (with respect to the operator norms) respectively. Then if $P_M$ and $P_{\mathcal{M}}$ have full supports, we have $P_M(B(M_0, \delta)) > 0$ and $P_{\mathcal{M}}(B(\mathcal{M}_0, \delta)) > 0$, which implies that
$$\begin{aligned}
\text{HRPCFD}_{M, \mathcal{M}}^2(\mu, \nu) &= \int \int d_{\text{HS}}^2(\boldsymbol{\Phi}_\mu^2(M, \mathcal{M}), \boldsymbol{\Phi}_\nu^2(M, \mathcal{M})) P_M(dM) P_{\mathcal{M}}(d\mathcal{M}) \\
&\geq \int_{B(\mathcal{M}_0, \delta)} \int_{B(M_0, \delta)} d_{\text{HS}}^2(\boldsymbol{\Phi}_\mu^2(M, \mathcal{M}), \boldsymbol{\Phi}_\nu^2(M, \mathcal{M})) P_M(dM) P_{\mathcal{M}}(d\mathcal{M}) \\
&> 0,
\end{aligned}$$
as claimed. $\qquad\square$

The boundedness of HRPCFD is easy to show by using the same arguments as in the proof of [19, Lemma 3.5] for PCFD.

**Lemma A.10.** *Let $\mu, \nu \in \mathcal{P}(\hat{\mathcal{X}})$ be two distributions on measure–valued path space. Then for any given integers $(n, m) \in \mathbb{N}^2$, for any $P_M \in \mathcal{P}(\mathcal{L}(\mathbb{R}^d, \mathfrak{u}(n)))$ and any $P_{\mathcal{M}} \in \mathcal{P}(\mathcal{L}(\mathbb{C}^{n \times n}, \mathfrak{u}(m)))$, one has*
$$HRPCFD_{M, \mathcal{M}}(\mu, \nu) \leq 2\sqrt{m}.$$

*Proof.* By the triangle inequality, we have
$$\begin{aligned}
\text{HRPCFD}_{M, \mathcal{M}}^2(\mu, \nu) &= \int \int d_{\text{HS}}^2(\boldsymbol{\Phi}_\mu^2(M, \mathcal{M}), \boldsymbol{\Phi}_\nu^2(M, \mathcal{M})) P_M(dM) P_{\mathcal{M}}(d\mathcal{M}) \\
&\leq \int \int (\|\boldsymbol{\Phi}_\mu^2(M, \mathcal{M})\|_{\text{HS}} + \|\boldsymbol{\Phi}_\nu^2(M, \mathcal{M})\|_{\text{HS}})^2 P_M(dM) P_{\mathcal{M}}(d\mathcal{M}).
\end{aligned}$$
Since $\boldsymbol{\Phi}_\mu^2(M, \mathcal{M}) = \int_{\boldsymbol{p} \in \hat{\mathcal{X}}} \mathcal{U}_{\mathcal{M}}(t \mapsto \boldsymbol{p}_t^M) \mu(d\boldsymbol{p})$ and $\mathcal{U}_{\mathcal{M}}$ takes values in $U(m)$ such that $\|\mathcal{U}_{\mathcal{M}}\|_{\text{HS}} = \sqrt{\text{tr}(\mathcal{U}_{\mathcal{M}} \mathcal{U}_{\mathcal{M}}^*)} = \sqrt{\text{tr}(I_m)} = \sqrt{m}$, we indeed have
$$\|\boldsymbol{\Phi}_\mu^2(M, \mathcal{M})\|_{\text{HS}} \leq \left( \int_{\boldsymbol{p} \in \hat{\mathcal{X}}} \|\mathcal{U}_{\mathcal{M}}(t \mapsto \boldsymbol{p}_t^M)\|_{\text{HS}}^2 \mu(d\boldsymbol{p}) \right)^{\frac{1}{2}} \leq \sqrt{m}.$$
Similarly, it holds that $\|\boldsymbol{\Phi}_\nu^2(M, \mathcal{M})\|_{\text{HS}} \leq \sqrt{m}$. Combining all above together we can deduce that $\text{HRPCFD}_{M, \mathcal{M}}^2(\mu, \nu) \leq 4m$. $\qquad\square$

Just like the classical PCFD (cf. [19, Proposition B.10]) we can also show that the HRPCFD is a specific Maximum Mean Discrepancy (MMD). For the definition of MMD, we refer readers to [19, Definition B.9].

**Proposition A.11.** *For any* $(n, m) \in \mathbb{N}^2$, *any* $P_{\boldsymbol{M}} \in \mathcal{P}(\mathcal{L}(\mathbb{R}^d, \mathfrak{u}(n)))$ *and any* $P_{\boldsymbol{\mathcal{M}}} \in \mathcal{P}(\mathcal{L}(\mathbb{C}^{n \times n}, \mathfrak{u}(m)))$, *the HRPCFD with respect to* $P_{\boldsymbol{M}}$ *and* $P_{\boldsymbol{\mathcal{M}}}$ *is an MMD with the kernel function* $\hat{\kappa} : \hat{\mathcal{X}} \times \hat{\mathcal{X}} \to \mathbb{R}$ *given by*

$$\hat{\kappa}(\boldsymbol{p}, \tilde{\boldsymbol{p}}) = \mathbb{E}_{P_{\boldsymbol{M}} \otimes P_{\boldsymbol{\mathcal{M}}}}[\langle \mathcal{U}_{\mathcal{M}}(t \mapsto \boldsymbol{p}_t^M), \mathcal{U}_{\mathcal{M}}(t \mapsto \tilde{\boldsymbol{p}}_t^M) \rangle_{\mathrm{HS}}]$$

$$= \mathbb{E}_{P_{\boldsymbol{M}} \otimes P_{\boldsymbol{\mathcal{M}}}}[tr(\mathcal{U}_{\mathcal{M}}(\boldsymbol{p}^M \star (\tilde{\boldsymbol{p}}^M)^{-1}))]$$

*where* $\star$ *denotes the concatenation operator on paths and* $(\tilde{\boldsymbol{p}}^M)^{-1}$ *denotes the reverse of the path* $t \mapsto \tilde{\boldsymbol{p}}_t^M$.

*Proof.* For $\mu, \nu \in \mathcal{P}(\hat{\mathcal{X}})$, it is easy to deduce that

$$\mathrm{HRPCFD}_{M, \mathcal{M}}^2(\mu, \nu) = \int \int \|\boldsymbol{\Phi}_\mu^2(M, \mathcal{M}) - \boldsymbol{\Phi}_\nu^2(M, \mathcal{M})\|_{\mathrm{HS}}^2 P_{\boldsymbol{M}}(dM) P_{\boldsymbol{\mathcal{M}}}(d\mathcal{M})$$

$$= \mathbb{E}_{P_{\boldsymbol{M}} \otimes P_{\boldsymbol{\mathcal{M}}}}[\|\boldsymbol{\Phi}_\mu^2(M, \mathcal{M})\|_{\mathrm{HS}}^2] + \mathbb{E}_{P_{\boldsymbol{M}} \otimes P_{\boldsymbol{\mathcal{M}}}}[\|\boldsymbol{\Phi}_\nu^2(M, \mathcal{M})\|_{\mathrm{HS}}^2]$$

$$- 2\mathbb{E}_{P_{\boldsymbol{M}} \otimes P_{\boldsymbol{\mathcal{M}}}}[\langle \boldsymbol{\Phi}_\mu^2(M, \mathcal{M}), \boldsymbol{\Phi}_\nu^2(M, \mathcal{M}) \rangle_{\mathrm{HS}}].$$

Moreover, note that

$$\mathbb{E}_{P_{\boldsymbol{M}} \otimes P_{\boldsymbol{\mathcal{M}}}}[\langle \boldsymbol{\Phi}_\mu^2(M, \mathcal{M}), \boldsymbol{\Phi}_\nu^2(M, \mathcal{M}) \rangle_{\mathrm{HS}}] = \int \langle \int \mathcal{U}_{\mathcal{M}}(\boldsymbol{p}^M) \mu(d\boldsymbol{p}), \int \mathcal{U}_{\mathcal{M}}(\tilde{\boldsymbol{p}}^M) \nu(d\tilde{\boldsymbol{p}}) \rangle_{\mathrm{HS}} d(\mathbb{P}_{\boldsymbol{M}} \otimes \mathbb{P}_{\boldsymbol{\mathcal{M}}})$$

$$= \int \int \langle \mathcal{U}_{\mathcal{M}}(\boldsymbol{p}), \mathcal{U}_{\mathcal{M}}(\tilde{\boldsymbol{p}}^M) \rangle_{\mathrm{HS}} \mu(d\boldsymbol{p}) \otimes \nu(d\tilde{\boldsymbol{p}}) d(P_{\boldsymbol{M}} \otimes P_{\boldsymbol{\mathcal{M}}})$$

$$= \int \left( \int \langle \mathcal{U}_{\mathcal{M}}(\boldsymbol{p}), \mathcal{U}_{\mathcal{M}}(\tilde{\boldsymbol{p}}^M) \rangle_{\mathrm{HS}} d(P_{\boldsymbol{M}} \otimes P_{\boldsymbol{\mathcal{M}}}) \right) \mu(d\boldsymbol{p}) \otimes \nu(d\tilde{\boldsymbol{p}}),$$

where we used the Fubini's theorem in the last equality. Therefore we actually obtain that for the kernel

$$\hat{\kappa}(\boldsymbol{p}, \tilde{\boldsymbol{p}}) = \mathbb{E}_{\mathbb{P}_{\boldsymbol{M}} \otimes \mathbb{P}_{\boldsymbol{\mathcal{M}}}}[\langle \mathcal{U}_{\mathcal{M}}(t \mapsto \boldsymbol{p}_t^M), \mathcal{U}_{\mathcal{M}}(t \mapsto \tilde{\boldsymbol{p}}_t^M) \rangle_{\mathrm{HS}}]$$

it holds that

$$\mathrm{HRPCFD}_{M, \mathcal{M}}^2(\mu, \nu) = \int \hat{\kappa}(\boldsymbol{p}, \tilde{\boldsymbol{p}}) \mu(d\boldsymbol{p}) \otimes \mu(d\tilde{\boldsymbol{p}}) + \int \hat{\kappa}(\boldsymbol{p}, \tilde{\boldsymbol{p}}) \nu(d\boldsymbol{p}) \otimes \nu(d\tilde{\boldsymbol{p}}) - 2\int \hat{\kappa}(\boldsymbol{p}, \tilde{\boldsymbol{p}}) \mu(d\boldsymbol{p}) \otimes \nu(d\tilde{\boldsymbol{p}}),$$

which implies that $\mathrm{HRPCFD}_{M, \mathcal{M}}$ is a MMD with the kernel function $\hat{\kappa}$.

The last claim is obvious: for any $\boldsymbol{p}, \tilde{\boldsymbol{p}} \in \hat{\mathcal{X}}$, one has, due to the fact that every $A \in U(m)$ satisfies $A^{-1} = A^*$, that

$$\kappa(\boldsymbol{p}, \tilde{\boldsymbol{p}}) = \mathbb{E}_{P_{\boldsymbol{M}} \otimes P_{\boldsymbol{\mathcal{M}}}}[\langle \mathcal{U}_{\mathcal{M}}(t \mapsto \boldsymbol{p}_t^M), \mathcal{U}_{\mathcal{M}}(t \mapsto \tilde{\boldsymbol{p}}_t^M) \rangle_{\mathrm{HS}}]$$

$$= \mathbb{E}_{P_{\boldsymbol{M}} \otimes P_{\boldsymbol{\mathcal{M}}}}[\mathrm{tr}(\mathcal{U}_{\mathcal{M}}(t \mapsto \boldsymbol{p}_t^M) \mathcal{U}_{\mathcal{M}}(t \mapsto \tilde{\boldsymbol{p}}_t^M)^*)]$$

$$= \mathbb{E}_{P_{\boldsymbol{M}} \otimes P_{\boldsymbol{\mathcal{M}}}}[\mathrm{tr}(\mathcal{U}_{\mathcal{M}}(t \mapsto \boldsymbol{p}_t^M) \mathcal{U}_{\mathcal{M}}(t \mapsto \tilde{\boldsymbol{p}}_t^M)^{-1})]$$

$$= \mathbb{E}_{P_{\boldsymbol{M}} \otimes P_{\boldsymbol{\mathcal{M}}}}[\mathrm{tr}(\mathcal{U}_{\mathcal{M}}(\boldsymbol{p}^M \star (\tilde{\boldsymbol{p}}^M)^{-1}))],$$

where we used the multiplicative property of the unitary features for $\mathbb{C}^{n \times n}$–valued paths $\boldsymbol{p}^M$ and $\tilde{\boldsymbol{p}}^M$, see also [19, Lemma A.5]. $\qquad\square$

Now we will construct a metric from HRPCFD which can characterise the extended weak convergence on precompact subset of FP.

**Lemma A.12.** *Suppose that* $\{(P_{\boldsymbol{M}_n}, P_{\boldsymbol{\mathcal{M}}_m}) \in \mathcal{P}(\mathcal{L}(\mathbb{R}^d, \mathfrak{u}(n))) \times \mathcal{P}(\mathcal{L}(\mathbb{C}^{n \times n}, \mathfrak{u}(m))) : n \in \mathbb{N}, m \in \mathbb{N}\}$ *is a double sequence of distributions with full supports. After a re-numeration we label them as a sequence* $((P_{\boldsymbol{M}_j}, P_{\boldsymbol{\mathcal{M}}_j}))_{j \in \mathbb{N}}$ *such that each* $(\boldsymbol{M}_j, \boldsymbol{\mathcal{M}}_j)$ *is a random admissibe pair in* $\mathcal{A}_{unitary}$. *Then the following defines a metric on* $\mathcal{P}(\hat{\mathcal{X}})$:

$$\widetilde{HRPCFD}(\mu, \nu) = \sum_{j=1}^{\infty} \frac{\min\{1, HRPCFD_{\boldsymbol{M}_j, \boldsymbol{\mathcal{M}}_j}(\mu, \nu)\}}{2^j}.$$

*Proof.* The symmetry and the triangle inequality are easy to check. We only need to show that $\widetilde{\mathrm{HRPCFD}}(\mu, \nu) = 0$ if and only $\mu = \nu$. The "if" part is trivial. Now suppose that $\widetilde{\mathrm{HRPCFD}}(\mu, \nu) = 0$ holds but $\mu \neq \nu$. Then by Theorem A.9 we know that there exists a pair of integers $(n, m) \in \mathbb{N}^2$ such that for any $P_{\boldsymbol{M}} \in \mathcal{P}(\mathcal{L}(\mathbb{R}^d, \mathfrak{u}(n)))$ and any $P_{\boldsymbol{\mathcal{M}}} \in \mathcal{P}(\mathcal{L}(\mathbb{C}^{n \times n}, \mathfrak{u}(m)))$ with full supports, it holds that $\mathrm{HRPCFD}_{\boldsymbol{M}, \boldsymbol{\mathcal{M}}}(\mu, \nu) > 0$. So, let us pick some $j \in \mathbb{N}$ such that $(\boldsymbol{M}_j, \boldsymbol{\mathcal{M}}_j) \in \mathcal{P}(\mathcal{L}(\mathbb{R}^d, \mathfrak{u}(n))) \times \mathcal{P}(\mathcal{L}(\mathbb{C}^{n \times n}, \mathfrak{u}(m)))$, we must have $\mathrm{HRPCFD}_{\boldsymbol{M}_j, \boldsymbol{\mathcal{M}}_j}(\mu, \nu) > 0$, which implies that $\widetilde{\mathrm{HRPCFD}}(\mu, \nu) \geq \frac{\min\{1, \mathrm{HRPCFD}_{\boldsymbol{M}_j, \boldsymbol{\mathcal{M}}_j}(\mu, \nu)\}}{2^j} > 0$, a contradiction. Hence we obtain that the so–defined $\widetilde{\mathrm{HRPCFD}}(\mu, \nu)$ is really a metric. $\qquad\square$

**Theorem A.13.** *Fix a sequence of random admissible pairs $(\boldsymbol{M}_j, \boldsymbol{\mathcal{M}}_j)_{j \in \mathbb{N}} \subset \mathcal{A}_{\text{unitary}}$ such that for every $(n, m) \in \mathbb{N}^2$ there exists a $j \in \mathbb{N}$ with $\boldsymbol{M}_j \in \mathcal{L}(\mathbb{R}^d, \mathfrak{u}(n))$ and $\boldsymbol{\mathcal{M}}_j \in \mathcal{L}(\mathbb{C}^{n \times n}, \mathfrak{u}(m))$ and their distributions $P_{\boldsymbol{M}_j} \in \mathcal{P}(\mathcal{L}(\mathbb{R}^d, \mathfrak{u}(n)))$ and $P_{\boldsymbol{\mathcal{M}}_j} \in \mathcal{P}(\mathcal{L}(\mathbb{C}^{n \times n}, \mathfrak{u}(m)))$ are fully supported. Let $\widetilde{\mathrm{HRPCFD}}$ be the metric defined as in Lemma A.12 via this sequence $(\boldsymbol{M}_j, \boldsymbol{\mathcal{M}}_j)_{j \in \mathbb{N}}$.*

1. *Let $\mathcal{K} \subset FP$ be a compact subset in the space $FP$ of filtered processes equipped with the topology induced by extended weak convergence. Then, for every sequence of filtered processes $(\mathbb{X}^k = (\Omega^k, \mathcal{F}^k, \mathbb{F}^k, X^k, \mathbb{P}^k))_{k \in \mathbb{N}} \subset \mathcal{K}$ and $\mathbb{X} = (\Omega, \mathcal{F}, \mathbb{F}, X, \mathbb{P}) \in FP$, we have*

$$\mathbb{X}^k \xrightarrow{EW} \mathbb{X} \iff \widetilde{\mathrm{HRPCFD}}(\mathbb{X}^k, \mathbb{X}) \to 0$$

*as $k \to \infty$.*

2. *Let $K \subset \mathcal{X}$ be a compact subset. Let $FP(K)$ be the space of all filtered processes taking values in $K$. Then, for every sequence of filtered processes $(\mathbb{X}^k = (\Omega^k, \mathcal{F}^k, \mathbb{F}^k, X^k, \mathbb{P}^k))_{k \in \mathbb{N}} \subset FP(K)$ and $\mathbb{X} = (\Omega, \mathcal{F}, \mathbb{F}, X, \mathbb{P}) \in FP(K)$, we have*

$$\mathbb{X}^k \xrightarrow{EW} \mathbb{X} \iff \widetilde{\mathrm{HRPCFD}}(\mathbb{X}^k, \mathbb{X}) \to 0$$

*as $k \to \infty$.*

*Proof.* 1. First suppose that $\mathbb{X}^k \xrightarrow{EW} \mathbb{X}$ for a sequence $(\mathbb{X}^k)_{k \in \mathbb{N}} \subset \mathcal{K}$ and $\mathbb{X} \in \mathcal{K}$. Clearly, for any sequence of piecewise linear measure-valued paths $\boldsymbol{p}^k$, $k \in \mathbb{N}$ and $\boldsymbol{p}$ (which are linear on each subinterval $[i, i+1]$, $i = 0, \ldots, T-1$) we have $\boldsymbol{p}^k \to \boldsymbol{p}$ with respect to the product topology on $\hat{\mathcal{X}}$ implies that for each fixed unitary representation $M \in \mathcal{L}(\mathbb{R}^d, \mathfrak{u}(n))$, the $\mathbb{C}^{n \times n}$–valued paths $(\boldsymbol{p}^k)^M = (\int \mathcal{U}_M(\boldsymbol{x}) \boldsymbol{p}_t^k(d\boldsymbol{x}))_{t \in [0,T]}$ converges to $\boldsymbol{p}^M = (\int \mathcal{U}_M(\boldsymbol{x}) \boldsymbol{p}_t(d\boldsymbol{x}))_{t \in [0,T]}$ with respect to the total variation norm as $k \to \infty$. Then, for every fixed unitary representation $\mathcal{M} \in \mathcal{L}(\mathbb{C}^{n \times n}, \mathfrak{u}(m))$, by the continuity of unitary feature map $\mathcal{U}_{\mathcal{M}}$ relative to the total variation norm (see e.g. [19, Proposition B.6]), we have $\mathcal{U}_{\mathcal{M}}(t \mapsto (\boldsymbol{p}^k)_t^M) \to \mathcal{U}_{\mathcal{M}}(t \mapsto \boldsymbol{p}_t^M)$ in $\mathbb{C}^{m \times m}$ (relative to the Hilbert–Schmidt norm) as $k \to \infty$. Hence, we actually have shown that the function $\boldsymbol{p} \in \hat{\mathcal{X}} \mapsto \mathcal{U}_{\mathcal{M}}(t \mapsto \boldsymbol{p}_t^M) \in \mathbb{C}^{m \times m}$ is continuous and bounded for the product topology on $\hat{\mathcal{X}}$. Now, as $\mathbb{X}^k \xrightarrow{EW} \mathbb{X}$ means that $P_{\hat{X}^k} \to P_{\hat{X}}$ weakly in $\mathcal{P}(\hat{\mathcal{X}})$ as $k \to \infty$, we indeed have for all $(M, \mathcal{M}) \in \mathcal{A}_{\text{unitary}}$,

$$\lim_{k \to \infty} \int \mathcal{U}_{\mathcal{M}}(t \mapsto \boldsymbol{p}_t^M) P_{\hat{X}^k}(d\boldsymbol{p}) = \int \mathcal{U}_{\mathcal{M}}(t \mapsto \boldsymbol{p}_t^M) P_{\hat{X}}(d\boldsymbol{p}),$$

that is, $\lim_{k \to \infty} \|\boldsymbol{\Phi}_{\mathbb{X}^k}^2(M, \mathcal{M}) - \boldsymbol{\Phi}_{\mathbb{X}}^2(M, \mathcal{M})\|_{\mathrm{HS}} = 0$. This observation together with the boundedness of the HRPCFD (see Lemma A.10), allows us to apply the dominated convergence theorem to derive that for every $j \in \mathbb{N}$ one has

$$\mathrm{HRPCFD}_{\boldsymbol{M}_j, \boldsymbol{\mathcal{M}}_j}^2(\mathbb{X}^k, \mathbb{X})$$

$$= \int \int d_{\mathrm{HS}}^2(\boldsymbol{\Phi}_{\mathbb{X}^k}^2(M, \mathcal{M}), \boldsymbol{\Phi}_{\mathbb{X}}^2(M, \mathcal{M})) P_{\boldsymbol{M}_j}(dM) P_{\boldsymbol{\mathcal{M}}_j}(d\mathcal{M}) \to 0$$

as $k \to \infty$. Consequently, we can conclude that $\widetilde{\mathrm{HRPCFD}}(\mathbb{X}^k, \mathbb{X}) \to 0$ as $k \to \infty$. Conversely, suppose that $(\mathbb{X}^k)_{k \in \mathbb{N}}$ is a sequence of filtered processes in $\mathcal{K}$ and $\mathbb{X} \in FP$ such that $\lim_{k \to \infty} \widetilde{\mathrm{HRPCFD}}(\mathbb{X}^k, \mathbb{X}) = 0$. Since $\mathcal{K}$ is compact, there is a subsequence of $(\mathbb{X}^k)_{k \in \mathbb{N}}$

(without loss of generality, assume this subsequence is the sequence itself) converging to a limit $\mathbb{Y} = (\Omega^Y, \mathcal{G}, \mathbb{G}, Y, \mathbb{Q}) \in \mathcal{K}$ in the extended weak topology. From the previous argument we know that $\lim_{k\to\infty} \widetilde{\mathrm{HRPCFD}}(\mathbb{X}^k, \mathbb{Y}) = 0$. Therefore, we actually obtain that $\widetilde{\mathrm{HRPCFD}}(\mathbb{X}, \mathbb{Y}) = 0$. In view of Theorem A.9, the equality $\widetilde{\mathrm{HRPCFD}}(\mathbb{X}, \mathbb{Y}) = 0$ means that $P_{\hat{X}} = P_{\hat{Y}}$, i.e., $\mathbb{X}$ and $\mathbb{Y}$ are synonymous. The above reasoning reveals that any accumulation point $\mathbb{Y}$ of the sequence $(\mathbb{X}^k)_{k\in\mathbb{N}}$ in the extended weak convergence coincides with $\mathbb{X}$. As a consequence, we have $\mathbb{X}^k \to \mathbb{X}$ in the extended weak topology as $k \to \infty$.

2. By [4, Theorem 1.7] the subspace $\mathrm{FP}(K)$ is precompact in FP for the extended weak topology, if $K \subset \mathcal{X}$ is compact. Hence the claim follows immediately from the result contained in the statement 1 with $\mathcal{K} = \overline{\mathrm{FP}(K)}$ (the closure of $\mathrm{FP}(K)$ with respect to the extended weak convergence).

$\square$

# B  Methodology and algorithm

## B.1  Estimating the conditional probability measure

---
**Algorithm 1** Training algorithm seq-to-seq regression model
---
**Input:** $X$ - real data; $B$ - batch size; $\eta_r$ - learning rate for the regression module; $d$ - path feature dimension; $l$ - lie degree; $M \in \mathbb{R}^{d\times\dim \mathfrak{u}_l}$; $T$ - path length; $\eta_r$ - learning rate.
1:  $F_\theta^X \leftarrow$ initialize
2:  **for** $i \in (1, \dots, \mathrm{iter}_r)$ **do**
3:      Sample $\boldsymbol{x}$ from $\boldsymbol{X}$ of size $B$
4:      $\mathcal{U}_{j,M}(t) \leftarrow \mathcal{U}_M(\boldsymbol{x}_{j,[t,T]})$ with $t \in \{0, \dots, T\}, j \in \{1, \dots, B\}$
5:      $\mathrm{RLoss}(\theta; \boldsymbol{x}, M) \leftarrow \frac{1}{B(T+1)} \sum_{t=0}^{T} \sum_{j=1}^{B} ||F_\theta^X(\boldsymbol{x}_{j,[0,T]})_t - \mathcal{U}_{j,M}(t)||_{HS}^2$
6:      $\theta \leftarrow \theta - \eta_r \cdot \nabla_\theta(\mathrm{RLoss}(\theta; \boldsymbol{x}, M))$
7:  **end for**
8:  **return** $F_{\theta^*}^X$ $\qquad\qquad\qquad\qquad\qquad\qquad\qquad\qquad$ ▷ Return the optimal model
---

---
**Algorithm 2** Sampling algorithm to approximate $\boldsymbol{\Phi}_{\mathbb{X}}^2$
---
**Input:** $F_\theta^X$ - regression module; $\boldsymbol{X} = (\boldsymbol{x}_j)_{j=1}^N$ - data sampled from distribution $P_X$; $\eta_r$ - learning rate for the regression module; $d$ - path feature dimension; $n, m$ - Lie degrees; $M \in \mathbb{R}^{d\times\dim \mathfrak{u}_l}$ $\mathcal{M} \in \mathbb{R}^{\dim \mathfrak{u}_{(n)}\times\dim \mathfrak{u}_{(m)}}$; $T$ - path length.
1:  $\hat{\boldsymbol{p}}^{\hat{X},M} \leftarrow$ zero matrix of length $N \times (T+1)$
2:  **for** $t \in (0, \dots, T)$ **do**
3:      **for** $j \in (1, \dots, N)$ **do**
4:          $\mathcal{U}_{j,M,\mathrm{past}}(t) \leftarrow \mathcal{U}_M(\boldsymbol{x}_{j,[0,t]})$
5:          $\hat{\mathcal{U}}_{j,M,\mathrm{future}}(t) \leftarrow F_\theta^X(\boldsymbol{x}_{j,[0,T]})_t$
6:          $\hat{\boldsymbol{p}}_{j,t}^{\hat{X},M} \leftarrow \mathcal{U}_{j,M,\mathrm{past}}(t) * \hat{\mathcal{U}}_{j,M,\mathrm{future}}(t)$
7:      **end for**
8:  **end for**
9:  $\hat{\boldsymbol{\Phi}}_{\mathbb{X}}^2(M, \mathcal{M}) \leftarrow \frac{1}{N} \sum_{j=1}^N \mathcal{U}_\mathcal{M}(\hat{\boldsymbol{p}}_j^{\hat{X},M})$
10: **return** $\hat{\boldsymbol{\Phi}}_{\mathbb{X}}^2(M, \mathcal{M})$
---

## B.2  HRPCF-GAN

In this section, we provide the mathematical formulation of HRPCF-GAN for conditional time series generation. Let $X := (X_t)_{t=1}^T$ denote a $\mathbb{R}^d$-valued time series of length $T$ with its distribution $P_X$. Suppose that we have i.i.d. samples $\mathbf{X} = (x_i)_i$ from $P_X$. We are interested in generating synthetic

future paths to approximate the conditional distribution of the future path $X_{\text{future}} := X_{(p,T]}$ given the past path $X_{\text{future}} := X_{[0,p]}$. For ease of notations, let $\mathcal{X}_{\text{past}} := \mathbb{R}^{d \times p}$ and $\mathcal{X}_{\text{future}} = \mathbb{R}^{d \times (T-p)}$ denote the space of the past path and future path, respectively.

**Conditional generator**    One step generator $g_\theta : \mathcal{X}_{\text{past}} \times \mathcal{Z} \to \mathbb{R}^d$, which maps $(x_{\text{past}}, z_t)$ to samples of the next time step via the following formula:

$$\begin{cases} h = [F_{\theta_e}(x_{\text{past}})]_p \\ o = F_{\theta_a}(h, z) \end{cases}$$

$F_{\theta_e} : \mathcal{X}_{\text{past}} \to \mathcal{H}$ is the sequence-to-sequence embedding module to extract the key information of the path up to time $t$ and $F_{\theta_a}$ exhibits the autoregressive generator architecture. We denote by where $\theta = (\theta_e, \theta_a)$ the generator's parameter.

We then apply one step generator $g_\theta$ in a rolling window basis to generate future time series of length $T - p$. More specifically, $G_\theta : (x_{[0:p]}, (z_t)_{t=p+1}^T) \mapsto (o_t)_{t=p+1}^T$, where we first set $o_{0:p} = x_{0:p}$ and for every $t \geq p$, $o_{t+1} = g_\theta(o_{t-p:t}, z_t)$.

In the following, we summarise the training algorithm for HRPCF-GAN in Algorithm 3.

---

**Algorithm 3** Training algorithm for HRPCF-GAN

---

**Input:** $p$ - past path length; $T$ - total path length; $d$ - path feature dimension; $\boldsymbol{X}$ - real data; $n$ - lie degree for EPCFD; $K_1$ - number of linear maps; $\boldsymbol{M} \in \mathbb{R}^{K_1 \times d \times \dim \mathfrak{u}_{(n)}}$; $m$ - lie degree for EHRPCFD; $K_2$ - number of linear maps for EHRPCFD; $\mathcal{M} \in \mathbb{R}^{K_2 \times \dim \mathfrak{u}_{(n)} \times \dim \mathfrak{u}_{(m)}}$; $G_\theta$ - generator; $B$ - batch size; $z$ - noise dimension; $\text{iter}_r$ frequency of regression module fine-tuning; $\eta_g, \eta_d$ - generator and discriminator learning rates.

1: # Vanilla PCFGAN training
2: **while** $\theta, \boldsymbol{M}$ not converge **do**
3:     Sample $z \sim \mathcal{N}^{z \times (T-q)}(0, 1)$ of size $B$, sample $x$ from $\boldsymbol{X}$ of size $B$
4:     $\tilde{\boldsymbol{x}}_{(p,T]} \leftarrow G_\theta(\boldsymbol{x}_{[0,p]}, z)$
5:     $\text{Loss}(\theta, \boldsymbol{M}; \boldsymbol{x}, z) \leftarrow \text{EPCFD}_{\boldsymbol{M}}^2(\boldsymbol{x}_{[0,T]}, (\boldsymbol{x}_{[0,p]}, \tilde{\boldsymbol{x}}_{(p,T]}))$
6:     $\boldsymbol{M} \leftarrow \boldsymbol{M} - \eta_d \cdot \nabla_{\boldsymbol{M}}(-\text{Loss}(\theta, \mathcal{M}; \boldsymbol{x}, z))$            ▷ Maximize the loss
7:     $\theta \leftarrow \theta - \eta_g \cdot \nabla_\theta(\text{Loss}(\theta, \mathcal{M}; \boldsymbol{x}, z))$            ▷ Minimize the loss
8: **end while**
9: # Regression training for real measure
10: **for** $i \in (1, \ldots, K_1)$ **do**
11:     $F_{\iota_i}^{\text{real}} \leftarrow$ initialize
12:     Train $F_{\iota_i}^{\text{real}}$ using $\boldsymbol{X}$ as described in Algorithm 1
13:     $F_{\eta_i}^{\text{fake}} \leftarrow F_{\iota_i}^{\text{real}}$            ▷ Set as the initialization
14: **end for**
15: # High-Rank PCF-GAN training
16: **while** $\theta, \mathcal{M}$ not converge **do**
17:     Sample $z \sim \mathcal{N}^{z \times (T-q)}(0, 1)$ of size $B$, sample $x$ from $\boldsymbol{X}$ of size $B$
18:     $\tilde{\boldsymbol{x}}_{(p,T]} \leftarrow G_\theta(\boldsymbol{x}_{[0,p]}, z)$
19:     **for** $i \in (1, \ldots, K_1)$ **do**
20:         **for** $t \in (0, \ldots, T)$ **do**
21:             $\hat{\boldsymbol{p}}_{i,t}^{\text{real}, M_i} \leftarrow \mathcal{U}_{M_i}(\boldsymbol{x}_{[0,t]}) * F_{\iota_i}^{\text{real}}(\boldsymbol{x}_{[0,T]})$
22:             $\hat{\boldsymbol{p}}_{i,t}^{\text{fake}, M_i} \leftarrow \mathcal{U}_{M_i}(\boldsymbol{x}_{[0,t]}) * F_{\iota_i}^{\text{fake}}((\boldsymbol{x}_{[0,p]}, \tilde{\boldsymbol{x}}_{(p,T]}))$
23:         **end for**
24:     **end for**
25:     $\text{Loss}(\theta, \mathcal{M}; \boldsymbol{x}, z, \boldsymbol{M}) \leftarrow \text{EHRPCFD}_{\boldsymbol{M}, \mathcal{M}}^2(\boldsymbol{x}_{[0,T]}, (\boldsymbol{x}_{[0,p]}, \tilde{\boldsymbol{x}}_{(p,T]}))$   ▷ Use Algorithm 2 and
        Equation (5) with $\boldsymbol{p}_{i,t}^{\text{real}, M_i}$ and $\boldsymbol{p}_{i,t}^{\text{fake}, M_i}$
26:     $\mathcal{M} \leftarrow \mathcal{M} - \eta_d \cdot \nabla_{\mathcal{M}}(-\text{Loss}(\theta, \mathcal{M}; \boldsymbol{x}, z, \boldsymbol{M}))$            ▷ Maximize the loss
27:     $\theta \leftarrow \theta - \eta_g \cdot \nabla_\theta(\text{Loss}(\theta, \mathcal{M}; \boldsymbol{x}, z, \boldsymbol{M}))$
28:     Do the following every $\text{iter}_r$ iterations:
29:     $\tilde{\boldsymbol{X}} \leftarrow (\boldsymbol{x}_{[0,p]}, G_\theta(\boldsymbol{X}_{[0,p]}, z))$
30:     Train $F_{\eta_i}^{\text{fake}}$ using $\tilde{\boldsymbol{X}}$ as described in Algorithm 1 for every $i \in (1, \ldots, K_1)$
31: **end while**

---

## B.3 Hypothesis testing for stochastic processes

We provide the following two algorithms for training ERHPCFD for the permutation test and computing the test power/Type 1 error of the permutation test, respectively.

---

**Algorithm 4** Training algorithm for the permutation test

---

**Input:** $\mathbf{X}$ - samples from distribution $\mu$; $\mathbf{Y}$ - samples from distribution $\nu$; $m > 0$ - sample size of $\mathbf{X}$; $n > 0$ - sample size of $\mathbf{Y}$; $n$ - lie degree for EPCFD; $K_1$ - number of linear maps; $M \in \mathbb{R}^{K_1 \times d \times \dim \mathfrak{u}_{(n)}}$; $m$ - lie degree for EHRPCFD; $K_2$ - number of linear maps for EHRPCFD; $\mathcal{M} \in \mathbb{R}^{K_2 \times \dim \mathfrak{u}_{(n)} \times \dim \mathfrak{u}_{(m)}}$; $B$ - batch size; $\eta$ - learning rate; iter$_1$, iter$_2$ - number of iterations.

1: # Vanilla PCFD optimization
2: **for** iter $\in (1, \ldots,$ iter$_1)$ **do**
3:      sample $\boldsymbol{x}, \boldsymbol{y}$ from $\boldsymbol{X}, \boldsymbol{Y}$ of size $B$
4:      Loss$(\boldsymbol{M}; \boldsymbol{x}, \boldsymbol{y}) \leftarrow$ EPCFD$^2_{\boldsymbol{M}}(\boldsymbol{x}_{[0,T]}, \boldsymbol{y}_{[0,T]})$
5:      $\boldsymbol{M} \leftarrow \boldsymbol{M} - \eta \cdot \nabla_{\boldsymbol{M}}(-$Loss$(\boldsymbol{M}; \boldsymbol{x}, \boldsymbol{y}))$          ▷ Maximize the loss
6: **end for**
7: # Regression training for real measure
8: **for** $i \in (1, \ldots, K_1)$ **do**
9:      $F^{\mathrm{X}}_{\iota_i}, F^{\mathrm{Y}}_{\iota_i} \leftarrow$ initialize
10:      Train $F^{\mathrm{X}}_{\iota_i}$ using $\boldsymbol{X}$ and $\boldsymbol{M}_i$ as described in Algorithm 1
11:      Train $F^{\mathrm{Y}}_{\iota_i}$ using $\boldsymbol{Y}$ and $\boldsymbol{M}_i$ as described in Algorithm 1
12: **end for**
13: # High Rank PCFD optimization
14: **for** iter $\in (1, \ldots,$ iter$_2)$ **do**
15:      sample $\boldsymbol{x}, \boldsymbol{y}$ from $\boldsymbol{X}, \boldsymbol{Y}$ of size $B$
16:      **for** $i \in (1, \ldots, K_1)$ **do**
17:          **for** $t \in (0, \ldots, T)$ **do**
18:              $\hat{\boldsymbol{p}}^{\mathrm{X}, M_i}_{i,t} \leftarrow \mathcal{U}_{M_i}(\boldsymbol{x}_{[0,t]}) * F^{\mathrm{X}}_{\iota_i}(\boldsymbol{x}_{[0,T]})$
19:              $\hat{\boldsymbol{p}}^{\mathrm{Y}, M_i}_{i,t} \leftarrow \mathcal{U}_{M_i}(\boldsymbol{y}_{[0,t]}) * F^{\mathrm{Y}}_{\iota_i}(\boldsymbol{y}_{[0,T]})$
20:          **end for**
21:      **end for**
22:      Loss$(\mathcal{M}; \boldsymbol{x}, \boldsymbol{y}, \boldsymbol{M}) \leftarrow$ EHRPCFD$^2_{\boldsymbol{M}, \mathcal{M}}(\boldsymbol{x}_{[0,T]}, \boldsymbol{y}_{[0,T]})$ ▷ Use Algorithm 2 and Equation (5)
     with $\boldsymbol{p}^{\mathrm{X}, M_i}_{i,t}$ and $\boldsymbol{p}^{\mathrm{Y}, M_i}_{i,t}$
23:      $\mathcal{M} \leftarrow \mathcal{M} - \eta_d \cdot \nabla_{\mathcal{M}}(-$Loss$(\theta, \mathcal{M}; \boldsymbol{x}, \boldsymbol{y}, \boldsymbol{M}))$          ▷ Maximize the loss
24: **end for**
25: **return** $\boldsymbol{M}, \mathcal{M}$          ▷ Return learnt parameters

---

**Algorithm 5** Estimating the test power/Type-I error of the permutation test

---

**Input:** $\alpha \in (0, 1)$ - significance level; $N > 0$ - # of experiments; $M > 0$ - # of permutations; $\mathbf{X}$ - samples from distribution $\mu$; $\mathbf{Y}$ - samples from distribution $\nu$; $m > 0$ - sample size of $\mathbf{X}$; $n > 0$ - sample size of $\mathbf{Y}$; $T$ - test statistic function; $H_0 \in \{1, 0\}$ - whether the null hypothesis is true or false ($H_0 = 1$ if $\mu = \nu$; otherwise $H_0 = 0$)

1: $\mathbf{Z} \leftarrow \text{Concatenate}(\mathbf{X}, \mathbf{Y})$
2: num_rejections $\leftarrow 0$
3: $i \leftarrow 1$
4: **while** $i \leq N$ **do**
5: $\quad \mathcal{T} \leftarrow \text{EmptyList}$
6: $\quad j \leftarrow 1$
7: $\quad$ **while** $j \leq M$ **do**
8: $\quad\quad \sigma \sim \text{Permutation}(\{1, 2, \cdots, m + n\})$
9: $\quad\quad T_\sigma \leftarrow T(\{\mathbf{Z}_{\sigma(1)}, \mathbf{Z}_{\sigma(2)}, \cdots, \mathbf{Z}_{\sigma(m)}\}, \{\mathbf{Z}_{\sigma(m+1)}, \cdots, \mathbf{Z}_{\sigma(m+n)}\})$
10: $\quad\quad \mathcal{T}.\text{append}(T_\sigma)$
11: $\quad\quad j \leftarrow j + 1$
12: $\quad$ **end while**
13: $\quad$ **if** $T(\mathbf{X}, \mathbf{Y}) > (1 - \alpha)\%$ quantile of $\mathcal{T}$ **then**
14: $\quad\quad$ num_rejections $\leftarrow$ num_rejections $+ 1$
15: $\quad$ **end if**
16: $\quad i \leftarrow i + 1$
17: **end while**
18: ratio $\leftarrow$ num_rejections $/ N$
19: **if** $H_0$ **then**
20: $\quad$ Type_I_error $\leftarrow$ ratio
21: $\quad$ **return** Type_I_error
22: **else**
23: $\quad$ test_power $\leftarrow$ ratio
24: $\quad$ **return** test_power
25: **end if**

---

## C  Numerical results

**Code**   The code is written in Python 3.10.8 and Pytorch 1.11.0. The supplementary code is available at `https://github.com/DeepIntoStreams/High-Rank-PCF-GAN.git` for ensuring full reproducibility. The experiments were performed on a computational system running Ubuntu 22.04.2 LTS, comprising five Quadro RTX 8000 GPUs with 48GB of memory each. The experiments are run on single GPU and the training time ranges from 30 minutes to 4 hours.

### C.1  Hypothesis testing

**Description**   The permutation test is a statistical method used to decide whether two measures $\mu, \nu$ are the same. The null hypothesis states $H_0 : \mu = \nu$ whereas the alternative hypothesis $H_1 : \mu \neq \nu$. Given a test metric $T$ and sample data $\boldsymbol{X} = \{\boldsymbol{x}_1, \ldots, \boldsymbol{x}_n\}$, $\boldsymbol{Y} = \{\boldsymbol{y}_1, \ldots, \boldsymbol{y}_m\}$ from $\mu$ and $\nu$ respectively. We construct the following distribution

$$\mathcal{T} := \left\{ T(\mathbf{Z}_{\sigma(1):\sigma(n)}, \mathbf{Z}_{\sigma(m+1):\sigma(n+m)}) \mid \sigma \in \Sigma_{n+m} \right\}.$$

where $\mathbf{Z} = (\boldsymbol{X}, \boldsymbol{Y})$ and $\Sigma_{n+m}$ is the permutation group of $n + m$ elements. Given the significance level $\alpha$, we reject the null hypothesis if $T(\boldsymbol{X}, \boldsymbol{Y}) > (1 - \alpha)\%$ quantile of $\mathcal{T}$.

**Methodology**   For each $H$, we sample the training dataset $\mathcal{D}_{\text{train}} = (B_{\text{train}}, B_{\text{train}}^H)$ and optimize the set $(\boldsymbol{M}_{K_1}, \mathcal{M}_{K_2})$ to maximize EHRPCFD[2] between the pair of measures, a detailed procedure can be found in Algorithm 4. Then, we sample two independent sets $\mathcal{D}_{\text{test}}^{H_0} = (B_{\text{test}}^H, \tilde{B}_{\text{test}}^H)$, $\mathcal{D}_{\text{test}}^{H_1} = (B_{\text{test}}, B_{\text{test}}^H)$ and calculate the power and type-I error accordingly. We refer to Algorithm 5 for the computation of test metrics.

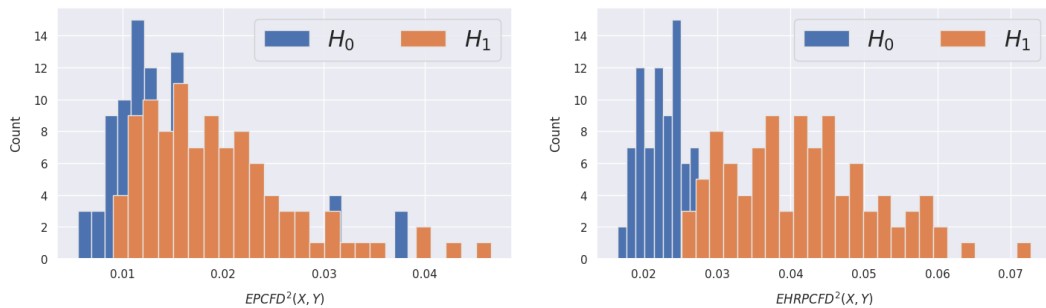

Figure 5: Distributions of EPCFD (left) and EHRPCFD (right) under $H_0$ and $H_1$ with Hurst parameter $H = 0.475$. The distribution consists of 100 runs under both hypotheses. For EPCFD, fix $K_1 = 8$ and $n = 5$. For High Rank PCFD fix $K_1 = 1$, $n = 3$, $K_2 = 10$, $m = 13$.

**Implementation details**   We provide the full details of the implementation of the numerical experiment in Section 5.1. Adopting the notation in Algorithm 4, we fix $n = 3$, $m = 13$, $K_1 = 1$ and $K_2 = 10$, these values are chosen via hyper-parameter fine-tuning. The regression model consists of a 2-layer LSTM module. The model's parameter is optimized using Adam optimizer with learning rates $0.001$ (for regression) and $0.02$ (for EHRPCFD).

**Additional numerical results**   We provide comprehensive tables for summarising the Type-I error and the computational time involved in Section 5.1.

| | Developments | | Signature MMDs | | | Classical MMDs | |
|---|---|---|---|---|---|---|---|
| $H$ | High Rank PCFD | PCFD | Linear | RBF | High Rank | Linear | RBF |
| 0.4 | $0.04 \pm 0.04$ | $0.04 \pm 0.04$ | $0.06 \pm 0.05$ | $0.04 \pm 0.04$ | $0.09 \pm 0.08$ | $0.07 \pm 0.06$ | $0.04 \pm 0.04$ |
| 0.425 | $0.07 \pm 0.04$ | $0.08 \pm 0.07$ | $0.03 \pm 0.03$ | $0.04 \pm 0.04$ | $0.14 \pm 0.05$ | $0.01 \pm 0.02$ | $0.03 \pm 0.02$ |
| 0.45 | $0.06 \pm 0.05$ | $0.08 \pm 0.02$ | $0.05 \pm 0.04$ | $0.02 \pm 0.03$ | $0.10 \pm 0.07$ | $0.04 \pm 0.04$ | $0.06 \pm 0.06$ |
| 0.475 | $0.04 \pm 0.04$ | $0.02 \pm 0.04$ | $0.05 \pm 0.04$ | $0.07 \pm 0.06$ | $0.12 \pm 0.07$ | $0.05 \pm 0.04$ | $0.03 \pm 0.04$ |
| 0.525 | $0.07 \pm 0.04$ | $0.09 \pm 0.07$ | $0.03 \pm 0.03$ | $0.04 \pm 0.02$ | $0.13 \pm 0.02$ | $0.02 \pm 0.02$ | $0.01 \pm 0.02$ |
| 0.55 | $0.05 \pm 0.03$ | $0.03 \pm 0.04$ | $0.07 \pm 0.06$ | $0.05 \pm 0.05$ | $0.17 \pm 0.10$ | $0.05 \pm 0.04$ | $0.02 \pm 0.02$ |
| 0.575 | $0.06 \pm 0.04$ | $0.04 \pm 0.04$ | $0.02 \pm 0.03$ | $0.06 \pm 0.07$ | $0.12 \pm 0.07$ | $0.06 \pm 0.02$ | $0.06 \pm 0.05$ |
| 0.6 | $0.10 \pm 0.07$ | $0.06 \pm 0.04$ | $0.05 \pm 0.04$ | $0.05 \pm 0.04$ | $0.09 \pm 0.04$ | $0.06 \pm 0.05$ | $0.06 \pm 0.05$ |

Table 3: Type-I error of the distances when $H \neq 0.5$ in the form of mean $\pm$ std over 5 runs. For PCFD, fix $K_1 = 8$ and $n = 5$. For High Rank PCFD fix $K_1 = 1$, $n = 3$, $K_2 = 10$, $m = 13$. For the RBF signature MMD and classical RBF MMD, fix $2\sigma^2 = 0.1$. For High Rank signature MMD, fix $\sigma_1 = \sigma_2 = 1$.

| | Developments | | Signature MMDs | | | Classical MMDs | |
|---|---|---|---|---|---|---|---|
| | High Rank PCFD | PCFD | Linear | RBF | High Rank | Linear | RBF |
| $m = n$ | Inference time (seconds) | | | | | | |
| 10 | $3.17 \pm 0.02$ | $2.58 \pm 0.01$ | $95.12 \pm 0.21$ | $122.9 \pm 0.32$ | $1214.76 \pm 12.41$ | $0.13 \pm 0.01$ | $0.28 \pm 0.03$ |
| 50 | $32.32 \pm 1.53$ | $23.14 \pm 1.04$ | $402.69 \pm 0.23$ | $533.43 \pm 0.33$ | $-$ | $0.56 \pm 0.04$ | $1.17 \pm 0.16$ |
| 100 | $111.25 \pm 2.92$ | $89.13 \pm 2.19$ | $1329.73 \pm 0.57$ | $1760.18 \pm 0.29$ | $-$ | $1.16 \pm 0.04$ | $2.64 \pm 0.11$ |
| Mini-batch size | Training time (seconds over 500 iterations) | | | | | | |
| 1024 | $695.18 \pm 8.57$ | $73.51 \pm 6.21$ | $-$ | $-$ | $-$ | $-$ | $-$ |

Table 4: Inference time of the permutation test across different sample sizes ($m = n$) and the training time of High Rank PCFD and PCFD before conducting the permutation test. The result is in the form of mean $\pm$ std over 5 runs. For PCFD, fix $K_1 = 8$ and $n = 5$. For High Rank PCFD fix $K_1 = 1$, $n = 3$, $K_2 = 10$, $m = 13$. For the RBF signature MMD and classical RBF MMD, fix $2\sigma^2 = 0.1$. Here fix $h = 0.45$.

## C.2 Generative modeling

**Datasets construction**  (1) 3-**dimensional fractional Brownian motion:** we simulate samples using the publicly available Python package fbm. The total length of each sample is 11 (counting a fixed initial point). The training and test data consists of two independent sampled sets of size 10000. (2) **Stock:** we select 5 representative stocks in the U.S. market, namely, Apple, Lockheed Martin, J.P. Morgan, Amazon, and P& G, and collect the daily return data from 2010 to 2020. The data collection is done using the Python package yfinance. We then construct the dataset using a rolling-window basis with length 10 (two weeks in real time) and stride 2. Finally, we split the dataset into training and test sets with a ratio of 0.8.

**Baseline**  We compare the performance of HRPCF-GAN with well-known models for time-series generation such as RCGAN [10] and TimeGAN [23]. Furthermore, we use PCFGAN [19] as a benchmarking model to showcase the significant improvement by considering the higher rank development as the discriminator. For fairness, we use the same generator structure (LSTM-based) for all these models.

**Conditional Generator**  The generator design is described in Appendix B.2. In particular, we choose $F_{\theta_e}$ and $F_{\theta_\alpha}$ to be two independent 2-layer LSTM modules. The first module takes the past path and encodes the necessary information to the latent space. The final hidden and cell state will be used as the input for the second LSTM module and the latent noise vector to produce the output distribution of the next time step. Also, we use the auto-regressive to simulate the future path recursively.

**Implementation details**  The training procedure is described in Algorithm 3, we adopt the same notation in this section. For both datasets, we set $T = 10$ and $p = 5$. We use the development layers on the unitary matrix [18] to calculate the PCFD distance, in particular, we fix $K_1 = 5$, $n = 5$, $K_2 = 10$, $m = 13$ for the discriminator design, these are obtained via hyper-parameter tuning. The regression model consists of a 2-layer LSTM module. Finally, we use the ADAM optimizer [12], to train both the generator and discriminator with learning rates 0.0001 and 0.002 respectively. We fine-tune the regression every 500 generator optimization iterations. To improve the training stability of GAN, we employed three techniques. Firstly, we applied a constant exponential decay rate of 0.97 to the learning rate for every 500 generator training iterations. Secondly, we clipped the norm of gradients in both generator and discriminator to 10.
All benchmarking models are trained with 15000 training iterations. For HRPCF-GAN, we trained the vanilla PCF-GAN with 10000 iterations then we switched to HRPCF discriminator and trained the model for a further 5000 iterations.

**Evaluation metrics**  We list here the test metrics we used for generative model assessment.

- Marginal score [16]: the average of Wasserstein distance of the marginal distribution between real and fake data across each dimension.
- Auto-correlation score [16]: the $l_1$ norm of the difference in the ACF between real and fake data

$$ACF(X, Y) := \sum_{\tau=1}^{T} \sum_{i=1}^{d} \left\| \hat{C}(\tau; X^{(i)}) - \hat{C}(\tau; Y^{(i)}) \right\|,$$

  where $\hat{C}(\tau; X)$ is the empirical auto-correlation estimator of $X_t$ and $X_{t+\tau}$.
- Cross-correlation score [16]: the $l_1$ norm of the difference in the correlation between real and fake data across each feature dimension.

$$Corr(X, Y) = \sum_{s,t=1}^{T} \sum_{i,j=1}^{d} \left\| \rho(X_s^{(i)}, X_t^{(j)}) - \rho(Y_s^{(i)}, Y_t^{(j)}) \right\|,$$

  where $\rho$ is the empirical correlation estimator.
- Discriminative score [23]: we train a post-hoc classifier to distinguish real data from fake data. Lower the score (absolute difference between classification accuracy and 0.5), meaning inability to classify, indicate better performance of the generative model.

- Predictive score [23]: we train a sequence-to-sequence model to predict the latter part of a time series given the first part, using generated data and real data, resp. The trained models are then tested on the real data resp. The lower loss (|TSTR-TRTR|) means the better resemblance of synthetic data to real data for the predictive task.

- $\text{Sig}W_1$ score [16]: by embedding the time series to the signature space, we can approximate the $W_1$ distance by the $l_2$ norm of the signature of the real and fake data.

$$\text{Sig}_{W_1}(X, Y) = ||\mathbb{E}_X[\text{Sig}(X_{[0,T]})] - \mathbb{E}_Y[\text{Sig}(Y_{[0,T]})]||_{l_2},$$

  where Sig denotes the signature transform of a path.

- Conditional expectation score: we estimate the conditional expectation of the future path on the fake measure via Monte Carlo and compute the averaged pairwise $l_2$ norm between real data.

- Outgoing Nearest Neighbour Distance score [14]: the ONND calculates for each example of real data the distance between the nearest generated data. This score tests the model's capability to capture the diversity of the target distribution.

- American put option score: we use Least-Square Monte Carlo method [17] to price an at-the-money American put option using both real and generated data. We set the strike date $T = 5$ days and risk-free rate $r = 0.01$. The score is computed as the average of $l_1$ differences of the estimated price across each stock.

For each test metric, a lower value indicates better model performance. We provide the results on the additional metrics in Table 5.

| Dataset | Test Metrics | RCGAN | TimeGAN | PCFGAN | HRPCF-GAN |
|---|---|---|---|---|---|
| fBM | Marginal | .010±.000 | .041±.000 | .007±.000 | **.005±.000** |
| | Predictive | .456±.004 | .686±.013 | .474±.003 | **.446±.002** |
| | ONND | **.622±.002** | .632±.002 | .654±.002 | **.622±.002** |
| Stock | Marginal (1+) | 1.181±.144 | .626±.137 | .476±.146 | **.272±.122** |
| | Predictive | .010±.000 | **.009±.000** | **.009±.000** | **.009±.000** |
| | ONND | .017±.001 | .017±.000 | **.016±.000** | **.016±.000** |

Table 5: Performance comparison of High Rank PCF-GAN and baselines. The best for each task is shown in bold. Each test metric is shown in the form of mean±std over 5 runs.

In all the numerical experiments of GAN training, we used a moderate matrix order ($l \leq 30$) to achieve satisfactory results. Specifically, our experiments were conducted on a single GPU, with the training time for HRPCF-GAN ranging from 30 minutes to 4 hours. Although HRPCF-GAN takes longer to train compared to other baselines, the total training time is kept at a manageable level, while the HRPCF-GAN consistently delivers better performance. We summarize the computation time of each of the models over 100 training iterations in Table 6.

| Training Time (s) | TimeGAN | RCGAN | PCF-GAN | HRPCF-GAN |
|---|---|---|---|---|
| fBM | $11.21 \pm 0.28$ | $5.98 \pm 0.35$ | $15.63 \pm 1.31$ | $31.96 \pm 2.92$ |
| Stock | $12.48 \pm 0.31$ | $7.39 \pm 0.65$ | $17.33 \pm 1.36$ | $34.52 \pm 2.72$ |

Table 6: Time measurement over 100 training iterations. The experiments are done using a single Quadro RTX 8000 GPU; each experiment is repeated 5 times, with the mean and standard deviation recorded.

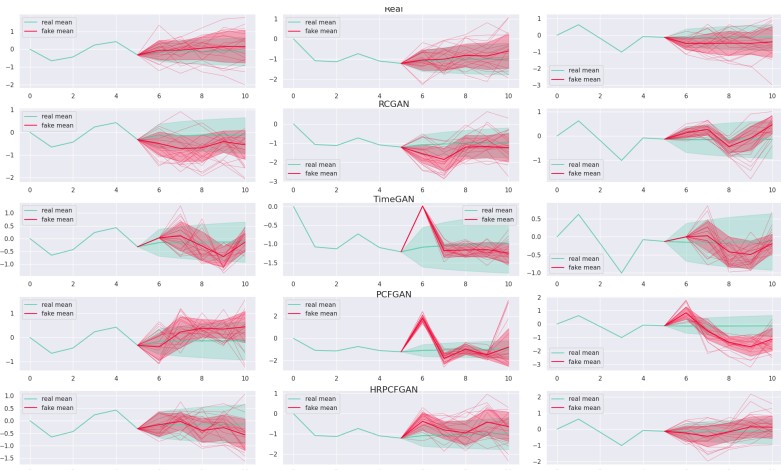

Figure 6: Sample plots from all models on fractional Brownian Motion conditioned on the same past path. The thick red line indicates the conditional mean of future estimated by fake samples, whereas the shaded red area presents the region of ±std. The thick green line corresponds to the theoretical value for the future expectation and the shaded area shown corresponds to the region of ± theoretical std.

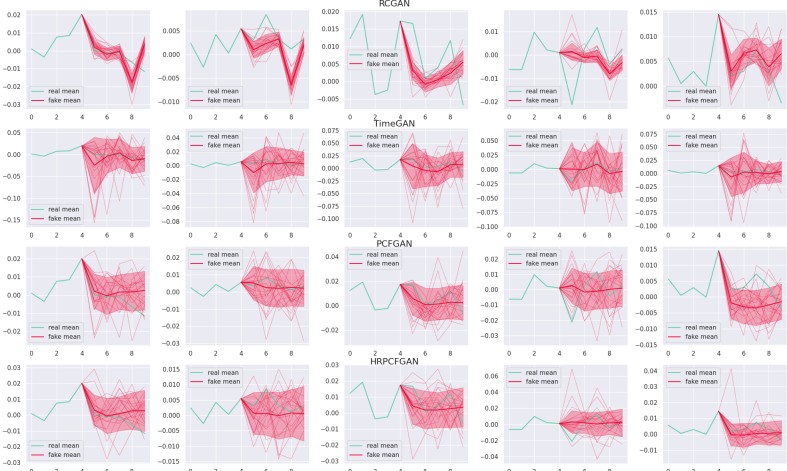

Figure 7: Sample plots from all models on Stock dataset conditioned on the same past path. The thick red line indicates the conditional mean of future estimated by fake samples, whereas the shaded red area presents the region of ±std.

