# OpenReview forum: "High Rank Path Development: an approach to learning the filtration of stochastic processes"
_NeurIPS.cc/2024/Conference — NeurIPS 2024 poster_

### Official Review · Reviewer_Yt11 · 2024-07-07

**Soundness:** 3
**Presentation:** 3
**Contribution:** 3
**Rating:** 6
**Confidence:** 1

**Summary:**

The paper addresses the issue of weak convergence of stochastic processes, whereby evolving information is generally unaccounted for. This can lead to discontinuities when applying these processes to multi-period decision-making problems. Prior work has proposed the concept of extended weak convergence, as introduced by Aldous (1981), but practical numerical implementations have been challenging. To address this, the authors introduce a novel metric called High Rank PCF Distance (HRPCFD) which is shown to overcome computational issues encountered in previous attempts. The paper then demonstrates the utility of HRPCFD via experiments on hypothesis testing and generative modelling of time series data.

**Strengths:**

Unfortunately this paper lies well outside my area of expertise and I am unable to review it effectively. The mathematical framework around extended weak convergence is not an area I’m familiar with, and I consequently found it challenging to grasp the nuances of the problem statement, the significance of the proposed HRPCFD metric, and the potential implications for applications in finance and economics.

So as not to negatively impact the paper’s chances of acceptance, I have defaulted to a mid-range score in my review, which reflects my assessment that the paper could nevertheless still be made more accessible for readers who are less familiar with the domain.

**Weaknesses:**

See above.

**Questions:**

See above.

---

> ### Author Rebuttal · Authors · 2024-08-05
>
> We address the reviewer's comments and questions in detail as below.
>
> ---
>
> **Comments:** *Unfortunately this paper lies well outside my area of expertise and I am unable to review it effectively. The mathematical framework around extended weak convergence is not an area I’m familiar with, and I consequently found it challenging to grasp the nuances of the problem statement, the significance of the proposed HRPCFD metric, and the potential implications for applications in finance and economics.*
>
> **Answer:**
>
> We appreciate the reviewer's kindness and positive score for our work. In the following, we would like to provide further explanation for our work, which hopefully would facilitate the reviewer's understanding.
>
> 1. *Importance and scope of our work*. Time series data, such as the stock price process, is ubiquitous and often exhibits randomness in finance and economics. Accurate modelling for the law of random time series (stochastic processes) plays an important role in decision marking. For example, training high-fidelity generative models to simulate synthetic time series that mimic future stock price movements can aid in developing more effective trading strategies and provide better estimators for assessing portfolio risk, with significant implications in finance and economics.
>
>     To tackle the challenge of time series modelling, a key question lies in defining a metric that quantifies the difference between two stochastic processes. We will explain the importance of the extended weak topology (EWT) in defining such a metric in the next bullet point. In this paper, we proposed the HRPCFD metric to capture EWT and design a computationally efficient algorithm to a computationally efficient algorithm that enables its direct application to enhance the discriminator of GANs to develop high-quality generative models for synthetic time series data.
>
> 2. *Why is EWT important?* We understand the difficulty for readers to grasp the meaning of EWT at first glance. To facilitate the understanding, we would like to use the toy example (Example A.1 in the paper) to illustrate the key concept of EWT and its usefulness. In the example, while the unconditional law of the processes $\mathbb{X}^n$ converges to $\mathbb{X}$, weak convergence fails to capture a key difference between the financial models $\mathbb{X}^n$ and $\mathbb{X}$. Specifically, if an agent believes the market dynamics as in $\mathbb{X}^n$, he/she always knows the outcome of the last day in advance, granting a predictive advantage, whereas in the “fair” market $\mathbb{X}$, the agent lacks this foresight. This crucial difference in the observed information flow-"$\text{No knowledge} \Rightarrow \text{Full knowledge} \Rightarrow \text{Full knowledge}$" for $\mathbb{X}^n$ versus "$\text{No knowledge} \Rightarrow \text{No knowledge} \Rightarrow \text{Full knowledge}$" for $\mathbb{X}$—is not reflected in weak convergence alone.
>
>     EWT is vital because it captures this difference through the conditional distributions. For markets $\mathbb{X}^n$, where the agent has full information on day 1, the conditional distribution becomes a single Dirac measure, annihilating randomness. In contrast, $\mathbb{X}$ retains genuine randomness at day 1, as reflected by a linear combination of Dirac measures. Since EWT is based on conditional distributions, it effectively measures differences in information evolution styles, ensuring continuity in multi-period decision-making as agents update their actions based on continually evolving information.
>
> 3. *Significance of the HRPCFD metric.*  In machine learning, many popular metrics on stochastic processes, (e.g., the Wasserstein distance, the PCFD metric, the signature MMD) based on weak topology, fail to ensure that the closeness of the conditional law (filtration) of two processes. To this end, we propose a novel HRPCFD metric to tackle this issue and prove that it can characterise EWT. Therefore, by using *HRPCFD metric* as the discriminator,  we are able to detect those differences of filtrations between various stochastic processes which are hard to be measured by those classical tools based on the weak convergence, and this observation was verified by our Hypothesis testing in Section 5.1.
>
>     Besides, our proposed HRPCFD is much more computationally efficient than the existing HR-SigMMD, which can theoretically characterise EWT. This enables the feasibility of using the HRPCFD as the discriminator of GANs for synthetic time series in practice.
>
>     The resulting *HRPCF-GAN* demonstrates the consistent outperformance of the strong and popular GAN model baselines specifically for time series generation, such as TimeGAN and RCGAN (see Section 5.2). The comprehensive evaluation of model performance using 8 different metrics indicates that the synthetic time series generated by our proposed HRPCF-GAN shares not only a similar law but also a similar filtration information with real-time series. These promising results highlight the potential of HRPCF-GAN for synthetic time series generation and its applications across various domains, including finance and economics.
>
> ---
>
> **Comments:** *So as not to negatively impact the paper’s chances of acceptance, I have defaulted to a mid-range score in my review, which reflects my assessment that the paper could nevertheless still be made more accessible for readers who are less familiar with the domain.*
>
> **Answer:** We will try our best to explain the essential ideas behind these abstract terms  and simplified lots of presentations in the revised version to make sure that the mathematical content is accessible for readers without any prior specialised knowledge on this topic. For instance, we will add a very detailed explanation after Example A.1. (as above) to show why the EWT is important in the multi-periods optimisation problems.

---

> > ### Comment · Reviewer_Yt11 · 2024-08-11
> > **Acknowledgement**
> >
> > Thank you for the additional context and explanation of the paper's contributions.

---

> > > ### Author Response · Authors · 2024-08-11
> > >
> > > You're welcome. If you have any further questions, please feel free to let us know. We are happy to address them as soon as possible. Many thanks.

---

### Official Review · Reviewer_9Q6u · 2024-07-08

**Soundness:** 4
**Presentation:** 4
**Contribution:** 4
**Rating:** 9
**Confidence:** 4

**Summary:**

Time series is ubiquitous in machine learning. They are modeled as stochastic processes and therefore notions of distance between stochastic processes and more generally convergence of stochastic processes are fundamental ideas. Weak convergence of probability measures occupies a central position in this area, but for many settings, like optimal stopping, or the one studied in this paper, it is not sufficient. Extended weak convergence, defined via weak convergence of prediction processes, is the right notion. This paper introduces a metric on the space of filtered processes that metrizes the topology of extended weak convergence, proposes statistical procedures to compute it, and then tests these ideas on GANs for time series.

**Strengths:**

This is a paper on a very nice topic and I learnt a lot while reading it. Some of the ideas are very abstract and the paper does a good job of organizing the topics and defining everything precisely so that a meticulous reader is not left confused. It is welcoming to see a rigorous paper in ML conferences. The ideas introduced in the paper are novel and the proofs of all the claim are carefully done, although I can't claim to have read every section in appendix in detail.

**Weaknesses:**

Although as mentioned above the paper defines everything clearly, the exposition on PCF and HRPCF could be improved. It took me quite some time after re-reading the paper multiple times and some other referenced paper to develop an intuition for these concepts, even though I have some background on probabilistic notions like weak convergence and extended weak convergence. I understand this is difficult to do well in a conference paper with page limits, but I think having a more detailed appendix on PCF and HRPCF would help.

I should mention here that I didn't find the experiments super convincing, but I am viewing this paper as a theoretical contribution, and thus any experiments it has as an added bonus and not a weakness.

**Questions:**

None

---

> ### Author Rebuttal · Authors · 2024-08-05
>
> We are pleased to know that the reviewer enjoyed reading our paper. We address the reviewer's comments as follows.
>
> **Comments:** *Although as mentioned above the paper defines everything clearly, the exposition on PCF and HRPCF could be improved.* *It took me quite some time after re-reading the paper multiple times and some other referenced paper to develop an intuition for these concepts*, *even though I have some background on probabilistic notions like weak convergence and extended weak convergence.* *I understand this is difficult to do well in a conference paper with page limits, but I think having a more detailed appendix on PCF and HRPCF would help.*
>
> **Answer:** We will add two additional sections in the appendix to provide a self-contained survey on the essential properties of extended weak convergence and PCF, and explain the construction of the HRPCF in a more transparent way.
>
> **Comments:** *I should mention here that I didn't find the experiments super convincing, but I am viewing this paper as a theoretical contribution, and thus any experiments it has as an added bonus and not a weakness.*
>
> **Answer:** We acknowledge that there is further room for improvement in numerical experiments. However, we would like to highlight that the proposed HRPCFD is significantly more computationally efficient than the existing high-rank signature MMD counterpart. This is empirically illustrated in Table 4 of Appendix C.1, which compares the inference times for the hypothesis testing example. Besides, the proposed HRPCFGAN overcomes the computational bottleneck of the HRSigMMD to enable its applications in generative modelling. We benchmark the proposed HRPCFGAN with strong GAN baselines specifically for time series generation, such as timeGAN and RCGAN, and show consistent performance improvement in 8 different test metrics to provide a comprehensive profile for the quality of synthetic time series data generation. The promising numerical results indicate the potential applications of HRPCFGAN on more challenging complex empirical datasets, which merits future research.

---

> > ### Comment · Reviewer_9Q6u · 2024-08-11
> >
> > Thank you for the reply. I have re-read the paper and the author response, and have increased my score and confidence to reflect my positive evaluation of the paper.

---

> > > ### Author Response · Authors · 2024-08-11
> > >
> > > Thank you very much!

---

### Official Review · Reviewer_gzqb · 2024-07-11

**Soundness:** 3
**Presentation:** 3
**Contribution:** 3
**Rating:** 7
**Confidence:** 4

**Summary:**

The paper constructs a computationally-implementable metric which metrizes an "extended" weak convergence for stochastic processes, which more plausibly accounts for the convergence of the process with respect to their filtrations.
The result can apparently more effectively account for similarities between controlled processes, at least in the class of linearly-interpolated stochastic paths considered here.

**Strengths:**

The method seems to construct high-rank analogues of classic tools, such as the characteristic function, such that the prediction processes arising from projection onto the filtration at each moment can be quantified and divergence between them metrized, without density evaluations, using empirical measures.

The classical SDE reasoning seems sound. I confess that I am less familiar with the rough path theory component, but there are no obvious red flags in the material.

**Weaknesses:**

The paper is long;
The results seem to be an improvement both theoretically and empirically over the main antecedents

* [18] Hang Lou, Siran Li, and Hao Ni. PCF-GAN: generating sequential data via the characteristic  function of measures on the path space. Advances in Neural Information Processing Systems,  36, 2023.
* [19] Cristopher Salvi, Thomas Cass, James Foster, Terry Lyons, and Weixin Yang. The signature 392 kernel is the solution of a goursat pde. SIAM Journal on Mathematics of Data Science, 3(3):873–899, January 2021.
* [20] Cristopher Salvi, Maud Lemercier, Chong Liu, Blanka Hovarth, Theodoros Damoulas, and  Terry Lyons. Higher order kernel mean embeddings to capture filtrations of stochastic processes.  Advances in Neural Information Processing Systems, 34:16635–16647, 2021

But it is not clear whether the increment is "important" in practice; Is the increased performance "worth" the implementation effort and/or computational cost? The answer is probably problem-dependent.

**Questions:**

l101: The prediction process seems to be introduced on a fixed finite set of times — $I = {0, \dots, T }$ and $X = (X_t)_{t\in I} $ — and yet we are concerned with continuously-indexed processes, so I would more naturally assume I is the interval $I=[0,T]$, and in fact we discuss linear interpolation in l109. Is this a notational confusion? What is the index $t$ of the filtration $\mathcal{F}_t$?

Title: I'm not sure about the grammar of the paper's title. "High Rank Path Development: an approach of learning the filtration of stochastic processes" -> "High Rank Path Development: an approach _to_ learning the filtration of stochastic processes"?

**Limitations:**

No particular issues noted.

---

> ### Author Rebuttal · Authors · 2024-08-05
>
> We address the reviewer's comments and questions in detail as below.
>
> ---
>
> **Comments:** *The results seem to be an improvement both theoretically and empirically over the main antecedents*
>
> *[18] Hang Lou, Siran Li, and Hao Ni. PCF-GAN: generating sequential data via the characteristic function of measures on the path space. Advances in Neural Information Processing Systems, 36, 2023.*
>
> *[19] Cristopher Salvi, Thomas Cass, James Foster, Terry Lyons, and Weixin Yang. The signature kernel is the solution of a Goursat PDE. SIAM Journal on Mathematics of Data Science, 3(3):873–899, January 2021.*
>
> *[20] Cristopher Salvi, Maud Lemercier, Chong Liu, Blanka Hovarth, Theodoros Damoulas, and Terry Lyons. Higher order kernel mean embeddings to capture filtrations of stochastic processes. Advances in Neural Information Processing Systems, 34:16635–16647, 2021.*
>
> *But it is not clear whether the increment is "important" in practice; Is the increased performance "worth" the implementation effort and/or computational cost? The answer is probably problem-dependent.*
>
> **Answer:**
> We agree with the referee that the answer is problem-dependent. We expect that our method would demonstrate more significant strength in solving multi-period optimization problems, such as optimal stopping and utility maximization, as well as in conditional time series tasks where the extended weak topology is crucial.
>
> However, the empirical performance is hard to estimate prior to experiments due to various factors regarding the dataset, models and training procedure (e.g., data size, model hyper-parameters, optimizer, etc). Therefore, it is very challenging to have such a cost-effective analysis before conducting the experiments. For practical applications, typically, given the computational budget, one would try a suite of machine learning methods and choose the best one among them. Given the moderate additional computational cost and robustness of our proposed HRPCF metric, we believe that the HRPCF metric and its corresponding GAN models are well-suited for practical applications.
>
> 1. *Consistent performance improvement \& moderate computational cost*. Our numerical experiments verified that our approach has a significantly better performance than several state-of-the-art methods (based on the weak convergence) in hypothesis testing and generative modelling. The computation complexity of HRPCFD is significantly reduced compared to the existing high rank methodologies such as HR-SigMMD, which is illustrated in Table 4 of the appendix. In comparison with popular GAN models (e.g., PCFGAN, TimeGAN and RCGAN), our HRPCF-GAN takes a longer time, but at the same magnitude. For example, in our numerical examples, the training time of HRPCF-GAN is kept at a manageable level across different datasets ranging from 30 minutes to 4 hours.
>
> 2. *Open source codes*. The open-source codes for our proposed method will significantly reduce the implementation effort for its empirical applications. We will make the code repository publicly available upon publication. By doing so, it is readily available for reuse and adaptation by the research community.  In particular, our HRPCFGAN codes can be flexibly applied to general time series data without the need for re-implementation from scratch.
>
> 3. *Robustness of HRPCFD metrics*. We would like to emphasize that our approach based on the extended weak convergence is **robust in any case**, as our Example A.1 shows: it can happen that sometimes weak topology provides a completely wrong criterion, whilst the extended weak convergence always works correctly, which was proved rigorously in many literature (see e.g., [1]).
>
> On a related note, it is interesting to investigate the sufficient condition under which stochastic processes such that the induced extended weak topology is strictly stronger than weak topology. To our best knowledge, so far, there is no easy criterion to judge in which circumstance the weak convergence coincides with the extended weak convergence (and even when they are equivalent, it is very hard to get a quantitative comparison). This study will be instrumental for the theoretical underpinnings of the performance gain of the EWT-based distance and provide guidance on empirical applications.
>
> ---
>
> **Question:** *The prediction process seems to be introduced on a fixed finite set of times--$I = 0,\ldots,T$ and $X = (X_t)_{t \in I}$--and* *yet we are concerned with continuously-indexed processes, so I would more naturally assume $I$ is the interval $I = [0,T]$*, *and in fact we discuss linear interpolation in l109. Is this a notational confusion? What is the index $t$ of the filtration $\mathcal{F}_t$ ?*
>
> **Answer:** In the present work, we only consider the discrete-time processes defined on $I= 0, \ldots, T$, and therefore the time index $t$ appeared in the filtration $\mathcal{F}_t$ only takes values in $0,\ldots,T$. It is just a convention in the rough path community that one views a discrete time path defined on $I= 0, \ldots, T$ as a piecewise linear path defined on the continuous time interval $[0,T]$ by a routine linear interpolation, because such identification may make some formulations easier (e.g., by doing so the unitary feature of a path can be formulated as the solution of an ODE on $[0,T]$). We will leave a remark in the revised version to explain these notations explicitly to avoid confusion.
>
> ---
>
> **Question:** *I'm not sure about the grammar of the paper's title. "High Rank Path Development: an approach of learning the filtration* *of stochastic processes" -> "High Rank Path Development: an approach to learning the filtration of stochastic processes"?*
>
> **Answer:** Thanks for this suggestion and we will change the title accordingly.
>
> ---
>
>
> [1] Julio Backhoff-Veraguas, Daniel Bartl, Mathias Beiglboeck, and Manu Eder. All adapted topologies are equal. Probability Theory and Related Fields. 178(3), 2020.

---

> > ### Comment · Reviewer_gzqb · 2024-08-11
> >
> > Thank you for your  response to my review. I don't think there is any need to shift my recommended score; I remain positively disposed towards a good paper, and I thank the authors for sharing it with us.

---

> > > ### Author Response · Authors · 2024-08-12
> > >
> > > You are welcome. Thank you very much for your positive feedback on our paper!

---

### Official Review · Reviewer_zBhD · 2024-07-12

**Soundness:** 3
**Presentation:** 3
**Contribution:** 1
**Rating:** 4
**Confidence:** 4

**Summary:**

This paper proposes High Rank Path method, motivated by the extended convergence notion and the rough path theory, to generate (conditioned) time-series data. A new metric HRPCFD is introduced, and experiments are conducted for Brownian motion, GANs with applications in finance.

**Strengths:**

The paper is rigorously written, which introduces a new metric HRPCFD on the path-valued processes based on various ideas from probability theory -- extended convergence, rough path, signature... I checked most proofs, and they are correct.

**Weaknesses:**

Weakness and comments:

(1) The paper may be too heavy for the Neurips audience (though I enjoyed reading it). It seems to be more suitable for a rigorous mathematical or statistical journal (e.g., Annals of Statistics).

(2) Many proofs of the results (e.g., Thm 3.3) are purely measure-theoretical, and I think the authors may shrink some proofs to keep the idea concise.

(3) The authors may want to explain why the proposed HRPCFD outperforms others (e.g., signature...) Is there any possible theoretical guarantee?

(4) The authors may have a discussion on the computational efforts of the proposed method (e.g., computational complexity and running time). The path-space optimization (or signature-type methods) often suffer from computational efficiency.

**Questions:**

See weakness.

---

> ### Author Rebuttal · Authors · 2024-08-05
>
> We thank the reviewer for the helpful comments. We address each question in detail as follows.
>
> ---
>
> **Answer to Q(1) / Why this work fits NeurIPS**
>
> Our paper addresses the crucial problem of defining the computationally feasible metric on the law of stochastic processes to capture the extended weak topology, which is ubiquitous for measuring the sensitivity w.r.t. models in decision making problems.  This has significant implications in machine learning, statistics and probability, particularly in hypothesis testing and generative models for time series data. As such, our work aligns well with the scope of NeurIPS.
>
> Besides, we would like to highlight substantial numerical contributions of our paper, which might be overlooked. In this paper, we design an efficient algorithm for training HRPCFD from data and construct the HRPCF-GAN by using HRPCFD as the discriminator for conditional time series generation. Numerical results show the promise of our proposed HRPCF-GAN for empirical time series data. We will make the code repository publicly available upon publication, which would be a useful tool for generative models for synthetic time series generation. The concrete algorithms and description of the corresponding methodology along with the open-source code repository, should have their own interests in the machine learning community.
>
> Our work balances the theoretical underpinnings of the HRPCFD metric with the practical implementation of the HRPCF-GAN algorithm for synthetic time series generation. Given the diversity and interdisciplinary nature of the NeurIPS audience, we believe that the NeurIPS conference is a suitable avenue for this work.
>
> ---
>
> **Answer to Q(2):** Now we adopt the reviewer's suggestion to add an ``informal proof'' for this theorem by emphasising the core idea and the main steps of the proof in an intuitive way so that the readers can easily understand the whole picture of the proof.
>
> ---
>
> **Answer to Q(3):**
> By the definition of the extended weak convergence (EWC) we can see that it induces a genuinely stronger topology than the weak convergence (i.e., **the EWC always implies weak convergence** while the converse fails in general, see Example A.1). Moreover, the EWC induces the coarsest topology for which the value function in any decision making problem is continuous.  Since the **HRPCFD metrises the EWC**,  it is natural to see an outperformance of HRPCFD in hypothesis testing and generative modeling than any other metrics based on the weak convergence, e.g., PCFD, signature MMD.
>
> ---
>
>
>
> **Answer to Q(4):**
> We summarize the computation complexity and running time of our proposed HRPCF method as follows. We will add Table 1 and a discussion on the computational complexity in the revised version.
>
> 1. *Inference time*. As we use the same generator architecture across different GAN models for a fair comparison, the inference time of the PCF-GAN is the same as that of other baseline models.
>
> 2. *Computational complexity \& Training time*. Let $d$ and $T$ denote the feature dimension and time dimension of the target time series data. The training algorithm of HRPCF-GAN can be divided into three parts: 1) Vanilla PCF-GAN training, 2) Regression training, and 3) HRPCF-GAN training. For each part, the training time complexity per epoch is linear in both $d$ and $T$ when keeping the hyper-parameters the same. However, the evaluation of EPCFD/EHRPCFD might be costly when using a very large matrix order $l$ of the Lie algebra. It could alleviated by employing the scaling-and-squaring method for an efficient computation for matrix exponential. Detailed complexity of this operation is discussed in [1].
>
> In all the numerical experiments of GAN training, we used a moderate matrix order ($l \leq 30$) to achieve satisfactory results. Specifically, our experiments were conducted on a single GPU, with the training time for HRPCF-GAN ranging from 30 minutes to 4 hours. Although HRPCF-GAN takes longer to train compared to other baselines, the total training time is kept at a manageable level, while the HRPCF-GAN consistently delivers better performance. We summarize the computation time of each of the models over 100 training iterations in Table 1.
>
> | Training Time (s) | TimeGAN           | RCGAN            | PCF-GAN           | HRPCF-GAN         |
> |-------------------|------------------|------------------|------------------|------------------|
> | fBM               | 11.21 ± 0.28      | 5.98 ± 0.35      | 15.63 ± 1.31      | 31.96 ± 2.92      |
> | Stock             | 12.48 ± 0.31      | 7.39 ± 0.65      | 17.33 ± 1.36      | 34.52 ± 2.72      |
> | RV                | 12.75 ± 0.41      | 6.37 ± 0.59      | 12.94 ± 1.21      | 32.77 ± 2.05      |
>
> *Table 1: Time measurement over 100 training iterations. The experiments are done using a single Quadro RTX 8000 GPU; each experiment is repeated 5 times, with the mean and standard deviation recorded.*
>
> 3. *Significant computation reduction of HRPCFD over HR-SigMMD*. The dimension of PCF is independent with $d$, and the same applies to HRPCF. However, in contrast, SigMMD's dimension grows geometrically with $d$, leading to dimensionality issues. The kernel trick can be used to alleviate the curse of dimensionality issue for the signature-based method. However, it is worth noting that the training time of SigMMD computed via the signature kernel is quadratic in terms of the sample size $n$ and time dimension $T$. In comparison, the computation complexity of HRPCFD is linear w.r.t $n$ and $T$.
>
> The significant computational advantage of PCF-based methods over signature-based methods is demonstrated in Table 4 of the appendix.
>
> [1] F. Longstaff and E. Schwartz. A new scaling and squaring algorithm for the matrix exponential. SIAM Journal on Matrix Analysis and Applications, 31(3):970–989, 2009.

---

> > ### Comment · Reviewer_zBhD · 2024-08-11
> >
> > I would thank the authors for the response, and will raise the score to 4.

---

> > > ### Author Response · Authors · 2024-08-11
> > >
> > > Thank you very much for raising your score! We noticed that the updated score is still negative. Please let us know your concerns and questions, and we will address them as soon as possible. Thanks in advance for your time.

---

> > ### Comment · Reviewer_gzqb · 2024-08-11
> > **What is the purpose of NeurIPS?**
> >
> > I possibly share reviewer zBhD's concerns about the scope of NeurIPS. Is this paper within scope for the conference? My feeling is that the conference has increasingly broad and increasingly unclear scope, and we may wish to prevent NeurIPS "eating" all the other conferences and journals. On the other hand, I am not sure that policing this boundary is responsibility of the authors of any given paper. To my mind this paper "does enough work" to make some abstract but useful mathematical quantity more computationally available, so it is probably within scope of the conference as it stands. Does the reviewer feel similarly? Or do they see the question of the _heaviness_ of this paper differently?

---

> > > ### Author Response · Authors · 2024-08-13
> > >
> > > Thank you very much for your thoughtful comments. You raised a valid point about the scope of NeurIPS, which is an important open question. The interpretation of its scope is subjective, and we respect the different views that individuals may hold.
> > >
> > > We appreciate your positive feedback regarding our paper being within the conference's scope. In our opinion,  channeling mathematical insights into innovative ML algorithms and investigating their theoretical underpinnings is crucial for advancing machine learning (ML) research. We have made our best efforts to ensure the rigor of our paper by introducing key mathematical tools while still making it accessible to a general ML audience in terms of its computational and application aspects.
> > >
> > > We welcome and are open to any suggestions for further improving our presentation of the paper from reviewers. We firmly believe that our paper would be a valuable contribution to the NeurIPS community.

---

### Author Rebuttal · Authors · 2024-08-05

We deeply appreciate all the reviewers for their helpful comments and constructive suggestions. We are pleased that all the reviewers find our work sound and well-presented. In the following, we provide detailed responses to the questions raised by each reviewer individually.

---

### Decision · Program_Chairs · 2024-09-25

**Decision:**

Accept (poster)

**Comment:**

Most of this paper's reviews are positive.

The main concern of the sole negative review - which comes from a reviewer who is a seasoned expert and said that they checked the paper's proofs - is that (a) the paper borders on too mathematical for NeurIPS, and (b) the emphasis is too much on technical aspects, many of which are already reasonably-well-understood to the experts, and too little on novel broad insights.

In discussion, however, that reviewer said they are not against the paper and are happy for it to be accepted if the other reviewers find it valuable and insightful. My understanding is that this is precisely what the other reviewers have said. I therefore recommend acceptance on basis of reviewer consensus.